# SCALING ATOMISTIC PROTEIN BINDER DESIGN WITH GENERATIVE PRETRAINING AND TEST-TIME COMPUTE

Kieran Didi[1,2,*]     Zuobai Zhang[1,3,4,*]     Guoqing Zhou[1,*]     Danny Reidenbach[1,*]
Zhonglin Cao[1,*]     Sooyoung Cha[8,9,*]     Tomas Geffner[1]     Christian Dallago[1]
Jian Tang[3,5,6]     Michael M. Bronstein[2,7]     Martin Steinegger[8,9,10,11]
Emine Kucukbenli[1,◇]     Arash Vahdat[1,◇]     Karsten Kreis[1,†]

[1]NVIDIA  [2]University of Oxford  [3]Mila - Québec AI Institute  [4]Université de Montréal  [5]HEC Montréal
[6]CIFAR AI Chair       [7]AITHYRA          [8]School of Biological Sciences, Seoul National University
[9]Interdisciplinary Program in Bioinformatics, Seoul National University  [10]Institute of Molecular Biology
and Genetics, Seoul National University       [11]Artificial Intelligence Institute, Seoul National University

*Project page:* https://research.nvidia.com/labs/genair/proteina-complexa/

## ABSTRACT

Protein interaction modeling is central to protein design, which has been transformed by machine learning with applications in drug discovery and beyond. In this landscape, structure-based de novo binder design is cast as either conditional generative modeling or sequence optimization via structure predictors ("hallucination"). We argue that this is a false dichotomy and propose *Proteína-Complexa*, a novel fully atomistic binder generation method unifying both paradigms. We extend recent flow-based latent protein generation architectures and leverage the domain-domain interactions of monomeric computationally predicted protein structures to construct *Teddymer*, a new large-scale dataset of synthetic binder-target pairs for pretraining. Combined with high-quality experimental multimers, this enables training a strong base model. We then perform inference-time optimization with this generative prior, unifying the strengths of previously distinct generative and hallucination methods. Proteína-Complexa sets a new state of the art in computational binder design benchmarks: it delivers markedly higher in-silico success rates than existing generative approaches, and our novel test-time optimization strategies greatly outperform previous hallucination methods under normalized compute budgets. We also demonstrate interface hydrogen bond optimization, fold class-guided binder generation, and extensions to small molecule targets and enzyme design tasks, again surpassing prior methods. Code, models and new data will be publicly released.

## 1 INTRODUCTION

Designing binding proteins is a central challenge in computational biology. As protein interactions are mediated by structure, most binder design methods adopt a structure-centric view. Advances in machine learning for protein structure prediction (Jumper et al., 2021; Abramson et al., 2024) and structure generation (Watson et al., 2023; Ingraham et al., 2023) now enable increasingly accurate de novo binder design in silico (Bennett et al., 2023). Modern AI-based approaches fall into two classes: *generative methods*, such as RFDiffusion (Watson et al., 2023), treat binder design as conditional generation, training on binder-target complex structures and producing new candidates for unseen targets; *hallucination methods*, exemplified by BindCraft (Pacesa et al., 2025), use the confidence and alignment scores of structure predictors to assess interfaces and optimize binder sequences via gradient feedback.

This dichotomy contrasts with large generation systems in language and image modeling, where a pretrained base model is combined with adaptive inference-time compute scaling and reasoning in a single framework (Wei et al., 2022; Snell et al., 2025; Ma et al., 2025). By analogy, current binder design methods resemble either pure training-time optimization (generative) or pure inference-time optimization without generative prior (hallucination). Inspired by inference-time scaling in language and vision, we argue this split is false and introduce *Proteína-Complexa* (hereafter Complexa), to our knowl-

---

*Core contributor.       ◇Equal advising.       †Project lead.

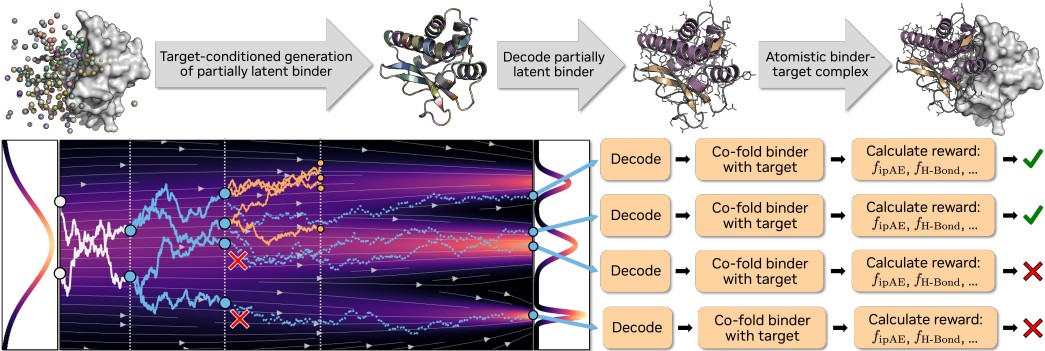

Figure 1: *(Top)* **Proteína-Complexa's** target-conditioned generation process. *(Bottom)* Scaling test-time compute, we use Complexa's generative prior for more efficient optimization than previous hallucination methods (Sec. 3.4). We depict beam search, which steers stochastic generation toward high-quality binders, guided by structure prediction models' interface scores or hydrogen bond energies. Intermediate candidate states (blue) are scored via rollouts (blue, dotted), promising candidates are kept, and new trajectories are launched (orange).

edge the first binder design framework that unifies a strong flow-based base generative model with flexible inference-time optimization utilizing the generative prior, combining the strengths of both.

Training an expressive generator for atomistic binder design requires large binder-target datasets, yet experimentally resolved multimers in the Protein Data Bank (PDB) are limited. To overcome this, we exploit domain-domain interactions from predicted monomer structures in the AlphaFold Database (AFDB). Using structural domain annotations from The Encyclopedia of Domains (TED) (Lau et al., 2024b), we partition proteins into domains and assemble artificial protein dimers. After clustering and filtering, we obtain *Teddymer*, a new large-scale dataset of synthetic binder-target complexes.

To train on this data, Complexa builds on La-Proteína (Geffner et al., 2026), which combines a scalable partially latent protein representation with flow matching (Lipman et al., 2023) and efficient transformer neural networks for accurate fully atomistic protein generation. We extend La-Proteína's architecture to binder design through a novel latent target conditioning mechanism. Inspired by pretraining and post-training alignment strategies from language and image generation, we adopt a staged training scheme on diverse monomers, Teddymer, and experimental multimer structures.

Next, we enhance performance by scaling inference-time compute. We adapt diffusion- and flow-based test-time scaling algorithms to binder design, including best-of-N sampling, beam search, Feynman–Kac steering, and Monte Carlo Tree Search (Fernandes et al., 2025; Ramesh & Mardani, 2025; Yoon et al., 2025; Singhal et al., 2025). Using interface confidence scores from structure predictors as rewards, we steer the base model to high-quality in-silico binders (Fig. 1). This unifies hallucination and generative modeling by efficiently searching within the generative prior. We also show that BindCraft-style hallucination can be accelerated by initializing from a generative model sample. In contrast to prior methods, Complexa does not require sequence re-design from backbone structures.

We comprehensively evaluate Complexa on binder design tasks for protein as well as small molecule targets. Our base model outperforms all prior generative models on established in-silico binding success metrics, and when compared to hallucination methods we achieve higher success rates under normalized compute budgets, confirming Complexa's state-of-the-art performance. Since strong binding between proteins and targets is often facilitated through hydrogen bonds, we also analyze and show how we can explicitly optimize interface hydrogen bonding, proving the flexibility of our framework. Moreover, we qualitatively demonstrate how fold class conditioning allows us to enhance binder diversity in a controllable manner—previous methods often produce primarily alpha helical outputs. A key task in computational biology is enzyme design, and we also test Complexa on a recent enzyme design benchmark (Ahern et al., 2025), where we again outperform prior work by a large margin. Ablation studies on modeling decisions and the Teddymer data provide further insights.

Modern generative AI systems scale both data and compute; the former during training, the latter during inference. To our knowledge, Complexa is the first structure-based protein design method to follow this paradigm, bridging the false divide between generative vs. hallucination approaches. We hope to enable efficient in-silico protein generation and design of binders for previously inaccessible targets.

**Key Contributions:** *(a)* We propose to combine previously distinct generative and hallucination methods—a novelty in the field of protein design. *(b)* We introduce Teddymer, a new large-scale synthetic dataset of protein dimers derived from domain-domain interactions. *(c)* We present Complexa, which extends La-Proteína to binder design, utilizes Teddymer, and implements efficient inference-

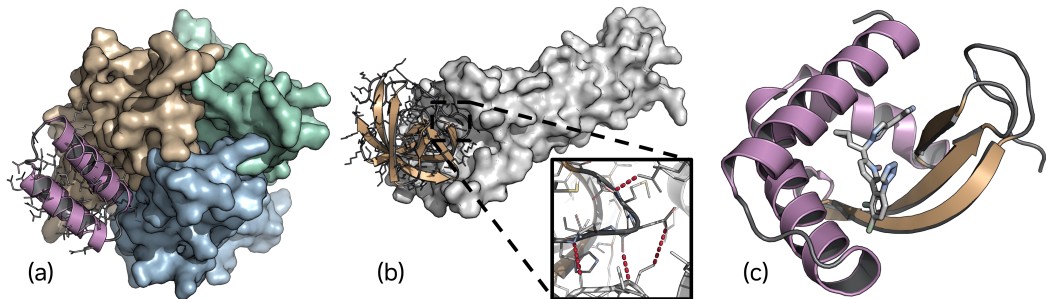

Figure 2: **Binders** generated by Complexa, passing in-silico success criteria (more visualizations in Sec. K). *(a)* TNF-$\alpha$ three-chain target. *(b)* Claudin-1 target, red interface hydrogen bonds. *(c)* OQO small molecule target.

time optimization accelerated by the generative prior. *(d)* We achieve state-of-the-art in-silico success rates for both protein and small molecule targets as well as in an enzyme design benchmark without the necessity for sequence re-design. *(e)* We further optimize and analyze interface hydrogen bonding, carry out ablation studies, and showcase fold class guidance for binder diversity. *(f)* We will publicly release source code, model weights and the Teddymer dataset to benefit the community.

## 2  BACKGROUND AND RELATED WORK

**Flow Matching and La-Proteína.** Complexa builds on top of La-Proteína (Geffner et al., 2026), whose core generation framework is flow-matching (Lipman et al., 2023; Albergo & Vanden-Eijnden, 2023), which models a probability path $p_t(\mathbf{x}_t)$ transforming tractable noise $p_{t=0}$ into data $p_{t=1}$ via an ordinary differential equation (ODE) $d\mathbf{x}_t = \mathbf{v}^\theta(\mathbf{x}_t, t)dt$ defined by a learnable vector field $\mathbf{v}^\theta$. Training uses conditional flow matching (CFM), where conditional paths $p_t(\mathbf{x}_t|\mathbf{x}_1)$ make the target vector field $\mathbf{u}_t(\mathbf{x}_t|\mathbf{x}_1)$ tractable for simple $p_0$. The network is trained by regressing $\mathbf{v}^\theta$ against $\mathbf{u}_t(\mathbf{x}_t|\mathbf{x}_1)$, yielding in expectation the same gradients as regression on the intractable marginal field $\mathbf{u}_t(\mathbf{x}_t)$. La-Proteína adopts the rectified flow formulation (Liu et al., 2023; Lipman et al., 2023; Geffner et al., 2025) with a linear interpolant $\mathbf{x}_t = t\mathbf{x}_1 + (1-t)\mathbf{x}_0$ and target $\mathbf{x}_1 - \mathbf{x}_0$ to model fully atomistic proteins using a *partially latent flow matching* framework. Specifically, it performs joint flow matching over residues' alpha carbon coordinates $\mathbf{x}^{C_\alpha}$ and per-residue continuous latent variables $\mathbf{z}$ that encode amino acid identities $\mathbf{s}$ and the residues' remaining atom coordinates $\mathbf{x}^{\neg C_\alpha}$. This partially latent representation emerges from La-Proteína's variational autoencoder (VAE) framework with encoder $\mathcal{E}(\mathbf{x}^{C_\alpha}, \mathbf{x}^{\neg C_\alpha}, \mathbf{s})$ and decoder $\mathcal{D}(\mathbf{x}^{C_\alpha}, \mathbf{z})$, corresponding to approximate posterior and conditional likelihood in the VAE formalism, respectively. La-Proteína's decoder outputs atom coordinates in an Atom37 protein representation. See La-Proteína details in Sec. B.

Geffner et al. (2026) introduces partially latent flow matching, shows high-quality fully atomistic monomer generation, analyzes biophysical validity and performs motif scaffolding. In contrast, our work is entirely focused on protein binder design and is therefore fully orthogonal and complementary.

**Generation vs. Hallucination.** Structure-based protein binder design with deep learning has traditionally followed two distinct routes: *Generative methods* train generative models, often flow or diffusion models (Ho et al., 2020; Song et al., 2021), on binder-target complexes and produce new binders conditioning on unseen targets. This was first shown for protein targets by the seminal RFDiffusion (Watson et al., 2023). RFDiffusion-AllAtom (Krishna et al., 2024) extended this to diverse target modalities. While these models generate backbone structures only, Protpardelle (Chu et al., 2024; Lu et al., 2025) and APM (Chen et al., 2025) enable fully atomistic binder generation. The *hallucination* approach to binder design corresponds to directly optimizing binder amino acid sequences towards high confidence and alignment scores under structure prediction models without training any generators. The term was coined by Anishchenko et al. (2021), and recently BindCraft (Pacesa et al., 2025), building on Goverde et al. (2023), scaled the approach via gradient-based optimization, using AlphaFold2 (Jumper et al., 2021). BoltzDesign (Cho et al., 2025) uses Boltz-1 (Wohlwend et al., 2025) and extends to small molecules, DNA and other targets. These methods rely on complex and ad-hoc modifications and relaxations of the sequence representation to obtain gradients. AlphaDesign (Jendrusch et al., 2025) does not use gradients, leveraging genetic algorithms instead. BAGEL (Lála et al., 2025) focuses on applications such as peptide design and intrinsically disordered targets. Many approaches, both generative and hallucination-based, apply inverse-folding methods ProteinMPNN (Dauparas et al., 2022) or LigandMPNN (Dauparas et al., 2025) to generated binder backbones for sequence re-design. Other binder design methods include PXDesign (Team et al., 2025c), LatentX (Team et al., 2025b), Chai-2 (Team et al., 2025a) and AlphaProteo (Zambaldi et al., 2024); these are proprietary models without available source code or model weights and in the last three cases lack methodological details.

## 3 PROTEÍNA-COMPLEXA

**Overview.** Our Complexa binder design framework consists of several key components: *(a)* To train Complexa's generative model to high performance, we first derive a new large-scale dataset of synthetic protein dimers, Teddymer. This is described in Sec. 3.1. *(b)* Complexa's base generative model builds on top of La-Proteína and conditions La-Proteína's partially latent flow matching component on the binder target without the need for adapting La-Proteína's autoencoder. This novel latent target conditioning mechanism as well as adjusted training objectives and strategies are presented in Sec. 3.2. *(c)* We discuss in-silico success metrics, including interface hydrogen bonds, in Sec. 3.3. *(d)* To enhance performance during inference and unify generation-based and optimization-based modeling, we adapt test-time compute scaling methods from the diffusion literature to Complexa—a first in the area of structure-based binder design to the best of our knowledge (see Sec. 3.4). To this end, rewards to guide the generation are derived from the previously discussed success criteria.

### 3.1 TEDDYMER: BINDER-TARGET DATA FROM INTERACTING PROTEIN DOMAINS

Training an expressive base generative model for binder design requires paired binder-target multimer data; however, such data is limited primarily to experimental structures in the protein data bank (PDB) (Berman et al., 2000). Meanwhile, the AlphaFold database (AFDB) (Varadi et al., 2021) provides a large set of computationally predicted monomers, but no similar repository of synthetic multimers exists. However, most AFDB monomers are multi-domain proteins, and recently Lau et al. (2024b) released The Encyclopedia of Domains (TED) of structural domain assignments for the AFDB (Lau et al., 2024a). Inspired by prior work (Sen & Madhusudhan, 2022), we argue that the biophysical interactions between structural domains of AFDB monomers are qualitatively similar to the interactions between chains in multimeric structures (see Fig. 3). Hence, we propose to split the AFDB multi-domain monomers into their individual domains and treat the resulting multimer structures as binder-target training data for Complexa. We start from AFDB50, a smaller, clustered version of the AFDB, and select the subset of structures with TED domain annotations, corresponding to 47M samples. We then split these structures into multimers, treating each domain as a separate chain. Next, we extract dimers from those multimers, filtering for spatial proximity of the dimer chains, and we use only structures with complete CAT annotations (Dawson

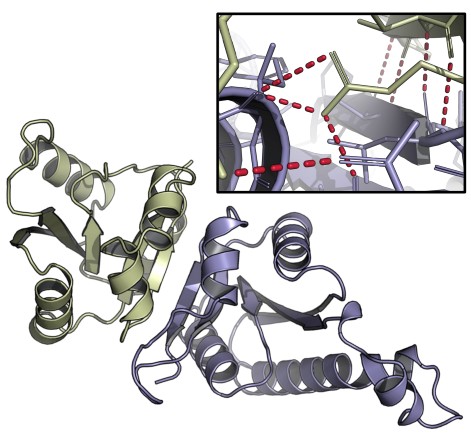

Figure 3: **Teddymer dimers** resemble realistic binder-target structures, including interface hydrogen bonding (zoom-in). Also see Sec. C.

et al., 2016). This results in 10M dimers. Finally, we cluster the data to reduce redundancy, resulting in 3.5M clusters. We name this database of TED-based dimers *Teddymer*. Please see Sec. C for data processing details and extended analyses comparing Teddymer interfaces with PDB multimers.

Teddymer is substantially larger than the PDB, which consists of ≈225k entries and requires further filtering to extract high-quality dimers suitable for training. As we show, our large-scale Teddymer provides a valuable additional resource for training at scale. In practice, we use four datasets in our experiments: *(a)* Foldseek AFDB monomer cluster representatives (van Kempen et al., 2024; Barrio-Hernandez et al., 2023), also used in previous work on protein generation (Geffner et al., 2025; Lin et al.,

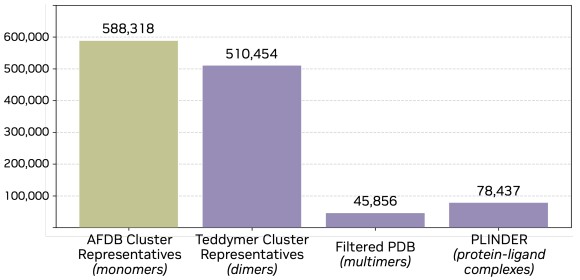

Figure 4: Filtered **training datasets** used by Complexa.

2024). *(b)* Teddymer dimer cluster representatives, filtered with interface pLDDT>70, ipAE<10, interface length >10. *(c)* Protein multimers filtered from PDB. *(d)* Filtered PLINDER protein-ligand dataset (Durairaj et al., 2024). See Sec. D for data processing details and Fig. 4 for dataset sizes.

## 3.2 COMPLEXA'S BASE GENERATIVE MODEL

We build on top of La-Proteína (Geffner et al., 2026) for two reasons: On the one hand, La-Proteína offers state-of-the-art accurate fully atomistic protein generation capabilities, necessary to produce precise atomistic binder-target interfaces. On the other hand, the framework is scalable and efficient, relying on fast transformer networks (Geffner et al., 2025) without slow triangular multiplicative or attention layers (Jumper et al., 2021). This is critical as binder design often involves in-silico generation of many candidates, especially when scaling compute during inference. Complexa introduces a series of adaptations to extend the La-Proteína framework to binder design, described below.

**Target- and Hotspot-Conditioning.** We modify La-Proteína to generate partially latent representations of binder proteins only. Thus, only the partially latent flow matching model conditions on the target, while the autoencoder simply encodes and decodes monomeric binders. To represent the target, we use the Atom37 scheme: each residue is assigned up to 37 three-dimensional atomic coordinates, determined by its amino acid type. These residue-wise Atom37 features are combined with amino acid identity features and binary hotspot tokens that mark interface residues near which the binder should be generated. During training, we extract hotspots from interface residues of binder-target training pairs; during inference, hotspots are typically known (for all benchmarks and in most applications; otherwise, preprocessing could include hotspot identification).

The resulting target-conditioning features, denoted as $\mathbf{c}^{\text{target}}$, are linearly embedded and concatenated in the token dimension to La-Proteína's sequence of alpha carbon coordinates $\mathbf{x}^{C_\alpha}$ and latent variables $\mathbf{z}$ that encode the binder. This extended sequence of noisy binder embeddings and clean target embeddings is then jointly processed by the transformer denoiser network of the partially latent flow model, which applies pair-biased attention (Geffner et al., 2025; Jumper et al., 2021). Pair representations are formed jointly over the extended sequence representation of binder and target. Our novel latent target conditioning mechanism is illustrated in Fig. 5 (note that a related architecture, but without the pair representation, was used by Geffner et al. (2026) for motif scaffolding tasks).

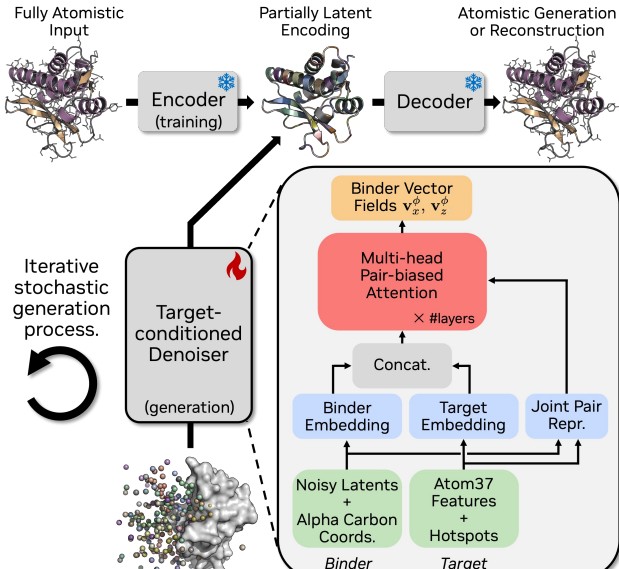

Figure 5: Complexa's **latent target conditioning**. When training the conditional denoiser, the encoder and decoder are frozen.

For small molecule targets, we featurize at the atomic level, using atom type, atom name, 3D atom coordinates, charge, and graph Laplacian positional encodings as sequence features. These are embedded and concatenated to the binder embeddings as before. Pair features are derived both from bond order and bond masks within the small molecule, and from distances between target atoms and binder residues' backbone atoms. Further architectural details in Sec. G.

**Objective with Translation Noise.** Using our latent target conditioning architecture, the partially latent flow model's training objective of Complexa's extended La-Proteína component is

$$
\min_\phi \mathbb{E}_{t_x,t_z,(\mathbf{x},\mathbf{c}^{\text{target}})\sim p^{\text{data}},\mathbf{x}_0^{C_\alpha}\sim p_0^{C_\alpha},\mathbf{z}_0\sim p_0^{\mathbf{z}},d\sim p_0^d}\left[\left\|\mathbf{v}_z^\phi\left(\mathbf{x}_{t_x}^{C_\alpha},\mathbf{z}_{t_z},\mathbf{c}^{\text{target}},t_x,t_z\right)-(\mathcal{E}(\mathbf{x})-\mathbf{z}_0)\right\|^2\right.
$$
$$
\left.+\left\|\mathbf{v}_x^\phi\left(\mathbf{x}_{t_x}^{C_\alpha},\mathbf{z}_{t_z},\mathbf{c}^{\text{target}},t_x,t_z\right)-\left(\mathbf{x}^{C_\alpha}-\left[\mathbf{x}_0^{C_\alpha}+d\,\mathbf{1}\right]\right)\right\|^2\right],
\tag{1}
$$

where $\mathbf{x}$ and $\mathbf{c}^{\text{target}}$ denote binder and target drawn from the training data distribution $p^{\text{data}}$, $\mathcal{E}(\mathbf{x})$ is the monomer encoder applied to the binder $\mathbf{x}$, and $t_x$ and $t_z$ are the interpolation times of alpha carbon coordinates and latents, respectively, drawn following Geffner et al. (2026) (the model uses separate schedules for $t_x$ and $t_z$ both during training and inference). Moreover, we have $\mathbf{x}_{t_x}^{C_\alpha}=t_x\mathbf{x}^{C_\alpha}+(1-t_x)(\mathbf{x}_0^{C_\alpha}+d\,\mathbf{1})$, $\mathbf{z}_{t_z}=t_z\mathcal{E}(\mathbf{x})+(1-t_z)\mathbf{z}_0$, $p_0^{C_\alpha}=\mathcal{N}(\mathbf{x}_0^{C_\alpha}|\mathbf{0},\boldsymbol{I})$, and $p_0^{\mathbf{z}}=\mathcal{N}(\mathbf{z}_0|\mathbf{0},\boldsymbol{I})$. In contrast

to Geffner et al. (2026), we additionally perturb the binder's alpha carbon coordinates with a scalar random global translation $d \sim p_0^d = \mathcal{N}(d|0, c_d^2)$ when applying the interpolant (we choose $c_d = 0.2$; all units are in $nm$). This explicit *translation noise* forces the model to reason over global positioning of the protein. This is irrelevant in monomer generation, but critical in binder design, when the binder needs to be accurately positioned at the interface. We can also understand this from a Fourier analysis perspective: Regular flow and diffusion models generate the lowest data frequencies at the beginning of the generative process without further refinement (Falck et al., 2025). Global translation corresponds to the lowest frequency mode, and our additional translation noise forces the model to refine positioning throughout generation (a related technique was used by Ahern et al. (2025)).

**Stagewise Training.** Inspired by the training strategies of large-scale generative AI systems, we adopt a multi-stage training pipeline. First, the autoencoder of Complexa's La-Proteína component is trained on AFDB monomers and then fine-tuned on PDB structures, as synthetic AFDB structures alone are overly idealized, being generated by a folding model. Next, we also pretrain the partially latent flow-matching model on encoded AFDB Foldseek cluster representative monomers, enabling it to acquire general protein structure generation capabilities. Only afterwards we train on binder target pairs: for protein binders, using Teddymer and PDB multimers; for small molecule binders, using PLINDER and AFDB monomers via LoRA (Hu et al., 2022). In the latter case, we use LoRA to avoid overfitting considering the small size of PLINDER. Full architecture, training, and sampling details in Sec. G.

**Complexa Design Considerations.** A key advantage of our model design is that the same autoencoder can be used regardless of the target type, as the autoencoder only needs to model monomeric chains—the de novo binders in our case. In fact, we employ the same autoencoder in all models, simplifying the framework. Our design parallels modern generation frameworks in vision, which are typically latent diffusion or flow models where only the latent generator, not the autoencoder, is conditioned on text or other signals (Rombach et al., 2022; Blattmann et al., 2023; Esser et al., 2024; Brooks et al., 2024), and where inference-time optimization of latent generation can likewise be applied (Fernandes et al., 2025; Singhal et al., 2025). Combined with our streamlined, fully transformer-based networks, this makes Complexa not only a modern, but also a fast and highly efficient binder design framework.

### 3.3 In-silico Success Metrics and Interface Hydrogen Bonds

**Protein Structure Prediction Scores.** Interface confidence and alignment scores from structure prediction models correlate with wet-lab success when evaluated on designed binder–target pairs (Overath et al., 2025). Consequently, such scores have become standard in-silico metrics for assessing binder quality. *Protein targets:* Following Zambaldi et al. (2024), we use AlphaFold2-Multimer (Evans et al., 2022), implemented through ColabDesign (Ovchinnikov et al., 2025). A generated binder sequence $\mathbf{s}$ is considered successful if, together with the target sequence and structure (omitted), it satisfies $f_{\text{pLDDT}}(\mathbf{s}){>}90$, $f_{\text{ipAE}}(\mathbf{s}){<}7.0$, and $f_{\text{Binder-RMSD}}(\mathbf{s}){<}1.5\text{Å}$. *Small molecule targets:* Following Cho et al. (2025), we use an AlphaFold3-like model, specifically RosettaFold-3 (RF3) (Corley et al., 2025), and define success by $f_{\text{min-ipAE}}(\mathbf{s}){<}2$, $f_{\text{Binder-RMSD}}(\mathbf{s}){<}2\text{Å}$ and $f_{\text{Ligand-RMSD}}(\mathbf{s}){<}5\text{Å}$. We generally directly evaluate Complexa's generated sequences (co-generated with atomistic binder structures), in contrast to prior works which usually re-design the sequence with ProteinMPNN or LigandMPNN from backbones—this is not necessary in Complexa.

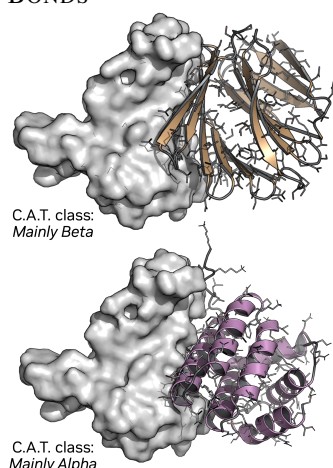

C.A.T. class: *Mainly Beta*

C.A.T. class: *Mainly Alpha*

Figure 6: **Fold class-conditional binder generation.** Samples pass success criteria. IFNAR2 target.

Importantly, structure prediction model confidence and alignment scores can also be used as rewards when searching for strong binders during inference. In particular, we use $f_{\text{ipAE}}$ as reward (see Sec. 3.4).

**Interface Hydrogen Bonding** is central in mediating strong protein-target interactions and its modeling can play a key role in protein structure prediction and design (O'Meara et al., 2015; Herschlag & Pinney, 2018). Therefore, we also explore optimizing interface hydrogen bond energies $f_{\text{H-Bond}}(\mathbf{s})$ of re-folded structures of our generated binder sequences $\mathbf{s}$ (again, we also use the target to fold and evaluate energies, but omit this for brevity). To calculate hydrogen bond energies, we follow Rosetta (Alford et al., 2017), implemented via Tmol (Leaver-Fay et al., 2025), and we use HBPlus (McDonald & Thornton, 1994) to detect hydrogen bonds in generated structures. See Sec. F for details on metrics.

## 3.4 INFERENCE-TIME OPTIMIZATION

Prior hallucination methods can be seen as pure inference-time optimization *without* use of a generative model. To *combine* Complexa's flow-based generative component with inference-time optimization, we adapt test-time scaling methods from the diffusion literature. We generally sample our partially latent flow model stochastically with reduced noise injection (see Geffner et al. (2026)).

**Best-of-N Sampling** is the simplest version of test-time compute scaling: Given an increasing compute budget, we grow the number of generated samples $N$ and select all binders with $f_{\text{ipAE}}(\mathbf{s}) < 7.0$.

**Beam Search** (visualization in Fig. 1). We maintain a set of $N$ (beam width) denoising trajectories $\mathcal{B}_{t_x,t_z} = \{(\mathbf{x}^{C_\alpha}_{t_x}, \mathbf{z}_{t_z})\}^{i=N}_{i=1}$, the "beam". From each beam element $i$, we initiate $L$ (branching factor) new stochastic denoising trajectories that we run for $K$ denoising steps to obtain a total of $N \times L$ new candidate states $\mathcal{C}_{t_x+\Delta t^K_x, t_z+\Delta t^K_z} = \{(\mathbf{x}^{C_\alpha}_{t_x+\Delta t^K_x}, \mathbf{z}_{t_z+\Delta t^K_z})_i\}^{i=NL}_{i=1}$. We now stochastically simulate all candidates $i$ towards clean partially latent states, decode, fold the resulting sequences, and calculate the candidates' rewards $R((\mathbf{x}^{C_\alpha}_{t_x+\Delta t^K_x}, \mathbf{z}_{t_z+\Delta t^K_z})_i)$, for instance using $f_{\text{ipAE}}$ or $f_{\text{H-Bond}}$. We can now select the top-$N$ candidates to form the updated beam after $K$ steps, which can be formalized as

$$\mathcal{B}_{t_x+\Delta t^K_x, t_z+\Delta t^K_z} = \arg\max_{\mathcal{T} \subseteq \mathcal{C}, |\mathcal{T}|=N} \sum_{i \in \mathcal{T}} R((\mathbf{x}^{C_\alpha}_{t_x+\Delta t^K_x}, \mathbf{z}_{t_z+\Delta t^K_z})_i). \tag{2}$$

This procedure is continued until we obtain fully denoised samples. In contrast to prior work (Fernandes et al., 2025; Ramesh & Mardani, 2025), we do not use Tweedie's formula to estimate the reward of average one-shot denoising predictions and instead iteratively roll out all candidate states. While this leads to stochastic rewards, the resulting samples on which decoder and structure predictor operate are clean—this is necessary, because our structure prediction-based rewards are only reliable on realistic sequences. Due to the efficiency of Complexa's generator, rolling out full generation trajectories is computationally inexpensive, and we perform this search only every $K$ steps.

**Feynman–Kac Steering** (FKS) (Singhal et al., 2025) is related to beam search, but instead of top-$N$ uses importance sampling to sample the tilted distribution $p^\phi(\mathbf{x}^{C_\alpha}, \mathbf{z})\exp\{\beta R(\mathbf{x}^{C_\alpha}, \mathbf{z})\}$, where $p^\phi$ denotes the model distribution and $\beta$ is an inverse temperature scaling (we omit indicating $\mathbf{c}^{\text{target}}$). Specifically, we sub-sample $N$ new states from the $N \times L$ candidates $\mathcal{C}_{t_x+\Delta t^K_x, t_z+\Delta t^K_z}$ with probability

$$p((\mathbf{x}^{C_\alpha}_{t_x+\Delta t^K_x}, \mathbf{z}_{t_z+\Delta t^K_z})_i) \propto \exp\{\beta R((\mathbf{x}^{C_\alpha}_{t_x+\Delta t^K_x}, \mathbf{z}_{t_z+\Delta t^K_z})_i)\}. \tag{3}$$

**Monte Carlo Tree Search** (MCTS) (Yoon et al., 2025; Ramesh & Mardani, 2025) treats the iterative generation process of flow and diffusion models as a tree, where different paths in the tree correspond to different stochastic denoising trajectories, and MCTS then searches within that tree. The method explores many possible denoising paths and then chooses the best next state along the tree. It is critical to balance exploration and exploitation when traversing the denoising tree (via parameter $C$ in Eq. (4)) and child states $(\mathbf{x}^{C_\alpha}_{t_x+\Delta t^K_x}, \mathbf{z}_{t_z+\Delta t^K_z})_i$ are chosen according to the index selection criterion

$$i = \arg\max_i \underbrace{\frac{R((\mathbf{x}^{C_\alpha}_{t_x+\Delta t^K_x}, \mathbf{z}_{t_z+\Delta t^K_z})_i)}{V((\mathbf{x}^{C_\alpha}_{t_x+\Delta t^K_x}, \mathbf{z}_{t_z+\Delta t^K_z})_i)}}_{\text{exploitation}} + C \underbrace{\sqrt{\frac{\ln(V((\mathbf{x}^{C_\alpha}_{t_x}, \mathbf{z}_{t_z})_i))}{V((\mathbf{x}^{C_\alpha}_{t_x+\Delta t^K_x}, \mathbf{z}_{t_z+\Delta t^K_z})_i)}}}_{\text{exploration}}, \tag{4}$$

where $(\mathbf{x}^{C_\alpha}_{t_x}, \mathbf{z}_{t_z})_i$ denotes the parent of $(\mathbf{x}^{C_\alpha}_{t_x+\Delta t^K_x}, \mathbf{z}_{t_z+\Delta t^K_z})_i$ and $V(\cdot)$ counts how often a (noisy) partially latent state has already been visited during previous searches. Further technical innovations are necessary to meaningfully apply MCTS in our flow's continuous state/action space; see Sec. H.

**Generate and Hallucinate** (G&H). The above approaches integrate search directly into the model's generative denoising process. We can also take a simpler approach to combine generative with hallucination methods: We propose to initialize a binder candidate with our generative model, and then refine its sequence through an established hallucination method, for which we choose the BindCraft framework (Pacesa et al., 2025). BindCraft uses several stages of optimization that partially rely on ad-hoc sequence relaxations to enable backpropagation through discrete sequences (in contrast to our Best-of-N, Beam Search, FKS, and MCTS variants, which are all principled and stable search algorithms). We study different optimization stages when combining BindCraft with generative model initialization.

Table 2: **Complexa's generative performance for protein targets** without optimization vs. baselines. *Self* denotes model sequence evaluation, *MPNN* full backbone-based re-design, and *MPNN-FI* the same with fixed interface amino acids. Note that RFDiffusion and Protpardelle only generate backbones and not their own sequences. Complete results in Sec. I. If methods tie on unique and absolute successes, we do not count this, see Sec. F.

| Model | # Unique Successes ↑ | | | # Times Best Method ↑ | | | Time [s] ↓ | Novelty ↓ |
|---|---|---|---|---|---|---|---|---|
| | Self | MPNN-FI | MPNN | Self | MPNN-FI | MPNN | | |
| RFDiffusion (Watson et al., 2023) | – | – | 4.68 | – | – | 3 | 70.8 | 0.87 |
| Protpardelle-1c (Lu et al., 2025) | – | – | 0.73 | – | – | 0 | **8.13** | **0.77** |
| APM (Chen et al., 2025) | 0.31 | 1.52 | 3.15 | 1 | 0 | 1 | 73.1 | 0.86 |
| Complexa *(ours)* | **9.10** | **13.6** | **14.4** | 14 | 14 | 14 | 15.6 | 0.80 |

To our knowledge, we are the first to systematically explore test-time scaling for protein design with generative models and search and optimization algorithms. Details & algorithms in Sec. H.

## 4 EXPERIMENTS

We train two Complexa generative base models, one for protein targets, one for small molecule targets, using the stagewise training protocol and latent target conditioning mechanism (Sec. 3.2). Training and model details provided in Sec. G, selected test targets in Sec. E. Generated binders shown in Fig. 2 and Sec. K.

Table 1: **Complexa's generative performance for small molecule targets** without optimization. RFDiffusion-AllAtom uses LigandMPNN, we evaluate sequences produced by Complexa.

| Model | # Unique Successes ↑ | | | | Times [s] ↓ | Novelty ↓ |
|---|---|---|---|---|---|---|
| | SAM | OQO | FAD | IAI | | |
| RFDiffusion-AllAtom | 2 | 3 | 5 | 8 | 87.4 | 0.72 |
| Complexa *(ours)* | **10** | **6** | **17** | **19** | **13.5** | **0.71** |

### 4.1 GENERATIVE BASE MODEL BENCHMARKING

**Protein Targets.** We first evaluate Complexa's generative model without test-time optimization and compare to publicly available generative methods that similarly do not rely on hallucination: RFDiffusion, Protpardelle-1c, and APM. For each method and target, we generate 200 binders from 40 to 250 residues. As prior approaches often collapse into producing the same binder repeatedly, we report the average number of *unique* successes, that is, we calculate success following Sec. 3.3, cluster successful samples, and count the clusters. We also report novelty against PDB, per-sample generation time, and the frequency with which each method achieves the best score across targets. In addition, we evaluate both self-generated model sequences and backbone-based re-design with ProteinMPNN, with and without preserving interface residues. Results in Tab. 2. Complexa significantly outperforms the baselines, producing more unique successful binders across all settings and winning on most targets. Its sampling time is substantially faster than RFDiffusion and APM, while also yielding more novel binders. Protpardelle is somewhat faster and favors novelty, but its success rates are poor. Importantly, while MPNN-based re-design can improve outcomes, even Complexa's self-generated sequences outperform all baselines, including those relying on re-design, making such additional steps unnecessary.

**Small Molecule Targets.** The only publicly available purely generative method for small molecule binder design is RFDiffusion-AllAtom (Krishna et al., 2024). We evaluate four molecules (SAM, OQO, FAD, IAI) following Cho et al. (2025), with results in Tab. 1, based on Complexa's self-generated sequences (Sec. I for extended results). We again significantly surpass the baseline, matching novelty and achieving much faster sampling speed. Combined with the protein target benchmarks, these results underscore Complexa's versatility and establish the state-of-the-art performance of its generative base model—without sequence re-design. Generated binders shown in Figs. 2 and 31.

**Diverse Binders via Fold Class Guidance.** Previous protein generators often produce primarily alpha helical outputs. Recently, Geffner et al. (2025) used fold class guidance to enhance secondary structure control in monomer modeling. Here, we extend this conditioning to binder design and train a Complexa model conditioned on CAT labels (Dawson et al., 2016). Qualitative results in Figs. 6 and 32 show how we can explicitly control the model to output beta sheet or alpha helix binders.

### 4.2 INFERENCE-TIME COMPUTE SCALING AND OPTIMIZATION

**Protein Targets.** Next, we evaluate Complexa's inference-time compute scaling techniques (we use only $f_{\text{ipAE}}$ as reward, details in Sec. H) against hallucination baselines, which can be viewed as brute-force optimization. We compare to the publicly available BindCraft (Pacesa et al., 2025), BoltzDesign (Cho et al., 2025) and AlphaDesign (Jendrusch et al., 2025). As optimization can be run with varying compute and different targets imply different computational demands, we

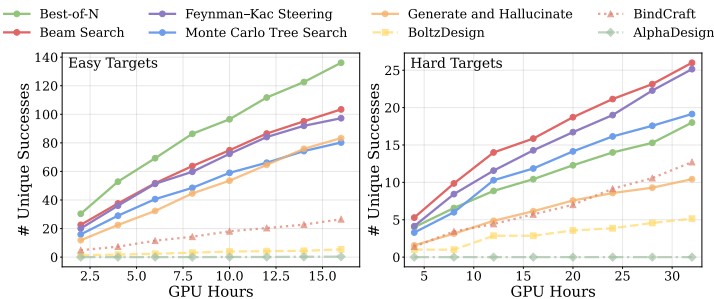

Figure 7: **Scaling analysis of inference-time optimization across different target difficulties.** Average unique success rate on 12 easy and 7 hard targets against test-time GPU hours. Five inference-time optimization algorithms from Sec. 3.4 are applied to Complexa and compared with hallucination baselines BoltzDesign and BindCraft.

plot success as a function of runtime and group targets into easy and hard classes (see Sec. E). As shown in Fig. 7, for easy targets simple methods like *Best-of-N* already outperform baselines, while on hard targets advanced searches such as *Beam Search*, *FKS*, and *MCTS* are required. This aligns with intuition: brute-force sampling suffices for easy cases, but structured search is essential when sampling becomes inefficient. Across both target sets, BindCraft, BoltzDesign and AlphaDesign perform poorly under matched compute, while our approaches consistently lead (on some targets, BindCraft is ahead, but overall Complexa wins by a large margin, see Sec. I.6).

Initializing BindCraft from our samples (*G&H*) accelerates search on easy but not hard targets, which is expected as the initial sample will often be poor for hard targets. Fig. 8 presents a case study on the hard two-chain target VEGFA, highlighting the superior performance of our inference-time optimization methods. Note that our methods here generally use Complexa's self-generated sequences without requiring re-design, in contrast to BindCraft and BoltzDesign, which rely on ProteinMPNN (note that BindCraft and BoltzDesign both achieve 0.80 novelty score, on-par with Complexa, Tab. 2). We attribute Complexa's superior performance to our direct integration of a strong generative prior with search, in contrast to the hallucination baselines, which do naive optimization.

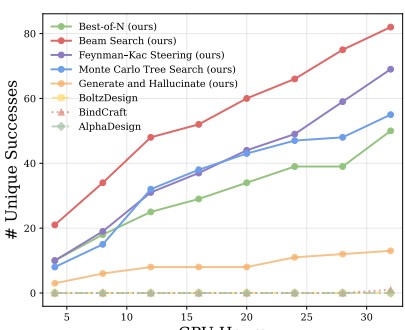

Figure 8: **Inference-time scaling for VEGFA** multi-chain target (hard).

**Small Molecule Targets.** We compare Complexa to BoltzDesign, the only available hallucination baseline for small molecules; see averaged results over the four molecule targets in Fig. 9. Again, Complexa's scaling methods are far superior. Results for all individual targets can be found in Sec. I.6.

**Interface Hydrogen Bond Optimization.** In Tab. 3, we show results of optimizing not only with $f_{ipAE}$, but including an interface hydrogen bond energy reward $f_{H-Bond}$ (cf. Sec. 3.3). Optimizing both $f_{ipAE}$ and $f_{H-Bond}$ can boost the average unique success rate. Importantly, including $f_{H-Bond}$ significantly enhances the number of interface hydrogen bonds (also Fig. 11). These results highlight the generality of Complexa's inference-time optimization framework and show that structure-based and physical energy-based rewards can be optimized, too—prior hallucination methods only considered folding model scores. This has promising future applications, for instance, when designing proteins with complex structural motif constraints.

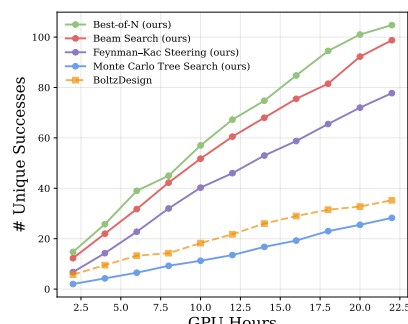

Figure 9: **Inference-time optimization for small molecule targets.**

**TNF-$\alpha$, H1, IL17A** are challenging multi-chain targets (Team et al., 2025c), not part of the benchmark above. None of the publicly available baselines were able to achieve any successes on them within <32 GPU hours of optimization. To showcase Complexa's scalability, we extended our search horizon for these targets beyond >100 GPU hours, allowing us to find 15 unique successes for TNF-$\alpha$, 7 for H1, and 1 for IL17A. This highlights Complexa's ability to find in-silico binder candidates even for very difficult targets as well as the flexibility of our test-time optimization framework. Sec. I.4 for experiment details. Successful binders shown in Figs. 2 and 30.

Table 3: Inference-time optimization using beam search with different **combinations of folding and hydrogen bond rewards.** Details in Sec. I.6.

| Model | # Unique Successes ↑ | # H-Bonds (avg.) ↑ |
|---|---|---|
| Complexa (no reward) | 77.00 | 5.271 |
| Complexa w/ $f_{ipAE}$ | 83.36 | 5.524 |
| Complexa w/ $f_{H-Bond}$ | 82.36 | **7.154** |
| Complexa w/ $f_{ipAE} + f_{H-Bond}$ | **86.26** | 6.518 |

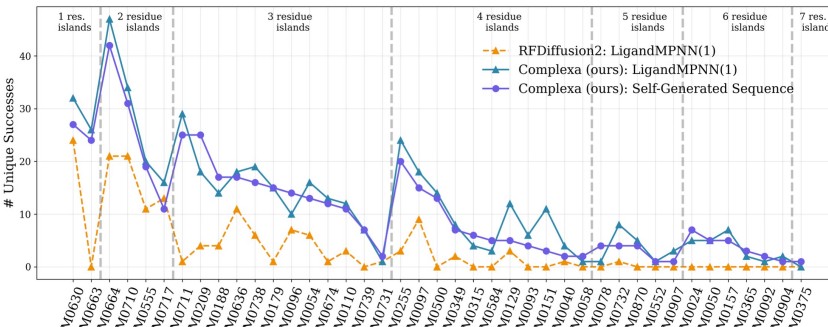

Figure 10: **Enzyme Design Benchmark.** Complexa significantly outperforms RFDiffusion2 in 38/41 AME benchmark tasks, both with re-designed and self-generated sequences. Extended results in Sec. I.9.

## 4.3 ENZYME DESIGN

A critical task in computational biology is enzyme design, where a protein is designed to catalyze a chemical reaction of a substrate target molecule. Recently, RFDiffusion2 (Ahern et al., 2025) introduced the Atomic Motif Enzyme (AME) benchmark, where an atomistic motif of the enzyme active site together with the substrate molecule are given, and a protein needs to be designed that reconstructs the motif and binds the substrate molecule. To tackle this task, we extended our Complexa model for small molecule targets with atomistic motif reconstruction capabilities. The AME benchmark consists of 41 tasks with varying numbers of catalytic residues that need to be reconstructed, organized in separated residue islands. A designed protein is considered successful, if the catalytic residues are reconstructed and there are no clashes with the ligand (Ahern et al., 2025). As previously, we only consider unique success. We find that Complexa significantly outperforms RFDiffusion2 on almost all tasks, both with self-generated sequences and re-designed sequences using LigandMPNN (Fig. 10). Sec. I.9 for implementation, evaluation details and extended results.

## 4.4 ABLATION STUDIES

In Sections I.1 and I.2, we perform ablation studies over our novel Teddymer data used for training Complexa and the translation noise (cf. Sec. 3.2). Please see Tab. 7 in the Appendix. We find that both are critical and hypothesize that without translation noise, the model cannot reason well over the binder's positioning. In early experiments we observed poor binder placement without translation noise. Without Teddymer-based training data, performance plummets. The data is critical to learn diverse protein-protein interactions, and filtered PDB data alone is too small (cf. Fig. 4). Therefore, we attribute our strong generative performance reported in Tab. 2 in part to the Teddymer data.

Please see our Appendix for extended results (Sec. I), additional visualization of generated binders (Sec. K), and complete training, model, architecture, algorithm, sampling, evaluation and data details.

## 5 CONCLUSIONS

We have introduced Proteína-Complexa, a fully atomistic framework for protein binder generation that bridges large-scale generative modeling with test-time compute scaling. Pretrained on Teddymer, a new dataset of synthetic dimers from AFDB domain–domain interactions, Complexa achieves state-of-the-art de novo binder design without the need for sequence re-design. Adapting test-time scaling techniques from the diffusion literature, we outperform prior hallucination methods by directly unifying generation and optimization. We also optimize interface hydrogen bonding, underscoring both the flexibility of our framework and opportunities for integrating physics- and learning-based

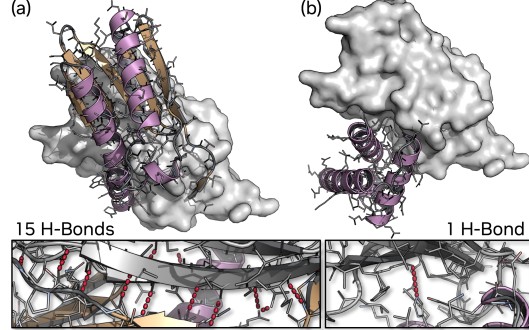

Figure 11: *(a)* **Interface hydrogen bond optimization** can create binders with extended interacting regions, forming strong hydrogen bonding (red, zoom-in). *(b)* Regular binders, even though passing success criteria, can be smaller with less interactions. Visualization shows successful binders for TrkA target.

modeling. Future work could train a single unified model capable of targeting and generating different molecular modalities, similar to recent work on peptide, small molecule and antibody design (Kong et al., 2025). Complexa paves the way toward efficient and scalable binder design, unlocking new and challenging targets, and motivates systematic exploration of test-time scaling in AI for protein design.

## REPRODUCIBILITY STATEMENT

To guarantee the reproducibility of our work, we provide complete details with respect to our novel methodology as well as training, evaluation and data processing. Moreover, we will release source code, models and the new dataset. In the following we describe each aspect in more detail.

**Methodological Details.** We ensure that all methodological innovations are explained appropriately to enable their reimplementation. All methods are explained on a high level in the main text (Sec. 3), and details as well as complete algorithms (in particular for all inference-time optimization methods) are provided in the Appendix in Sections G and H.

Moreover, background on La-Proteína is provided in Sec. B.

**Training Details.** Model training is discussed in Sec. 3.2 and complete training details and hyperparameters are provided in Sec. G.

**Evaluation Details.** Evaluation is discussed throughout the main paper on a high level, while evaluation metric details are provided in Sec. F. Furthermore, Sec. G covers how we sample from our base generative model and Sec. J explains how we sampled all baselines. Algorithms for generation with our inference-time optimization framework are provided in Sec. H. The selected protein and small molecule targets are described in Sec. E.

**Data Processing Details.** Training data and data processing details are explained in the main paper in Sec. 3.1 and in detail in Sections C and D. In that context, please note that our novel Teddymer data is based on the existing and publicly available datasets AFDB (Varadi et al., 2021) and TED (Lau et al., 2024a), and its careful processing is described in detail in Sec. C, making the process fully reproducible.

**Code, Model, and Data Release.** Upon acceptance of the work we will publicly release source code, model weights, and the novel Teddymer dataset, under permissive licensing.

## ETHICS STATEMENT

Generative approaches to de novo protein binder design promise to unlock broad advances across science, technology, and society. In medicine, they could speed the discovery of vaccines, antibodies, and targeted therapeutics that address urgent health challenges such as infectious diseases and cancer. In biotechnology, they may enable more efficient and flexible enzyme engineering. Beyond applications, these models have the potential to transform basic science. For instance, by generating and testing large libraries of protein-target interactions, they offer powerful tools to probe the fundamental principles of biophysical interactions, molecular recognition and protein folding. While these tools hold enormous promise, it is equally critical to acknowledge that generative models for binder design could be misapplied in ways that pose risks. For this reason, their use demands prudent oversight and a strong emphasis on responsible deployment.

## FUNDING ACKNOWLEDGMENTS

M.B. is partially supported by the EPSRC Turing AI World-Leading Research Fellowship No. EP/X040062/1 and EPSRC AI Hub No. EP/Y028872/1. M.S is supported by the National Research Foundation of Korea grants (NRF) (2020M3-A9G7-103933, RS-2021- NR061659, RS-2021- NR056571, RS-2024-00396026 and RS-2020-NR049543), the Novo Nordisk Foundation NNF24SA0092560 and the Creative-Pioneering Researchers Program. J.T. acknowledges funding from the Canada CIFAR AI Chair Program

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

# Appendix

# A   LIMITATIONS AND FUTURE WORK

The focus of this work is to introduce and validate Complexa's novel atomistic binder design framework, which unifies state-of-the-art generative modeling with modern inference-time optimization, combining the strengths of previously separate generative and hallucination approaches. To this end, we restrict our applications to designing binders and enzymes for protein and small molecule targets. A natural extension would be to apply Complexa to other molecular modalities, such as DNA and RNA. To this end, it would be interesting to also train a single, unified model that can both condition on and flexibly generate different molecular modalities, including proteins, peptides, small molecules, nucleic acids, antibodies and more. Joint training on diverse molecules may also boost overall performance due to transfer learning across the modalities. This idea was recently explored by UniMoMo in the context of small molecule, peptide and antibody design (Kong et al., 2025).

While our evaluations are limited to in-silico success metrics, an important next step will be experimental validation of generated binders in the wet lab. Future work could also target additional molecular properties—such as specificity or thermostability—by integrating suitable computational predictors into Complexa's inference-time search framework. We leave these directions to future investigation.

# B   LA-PROTEÍNA BACKGROUND

Below, we describe La-Proteína (Geffner et al., 2026) on a high-level. This section is purely for reference purposes. We adopt their notation, and emphasize that all content here is explained in more detail in Geffner et al. (2026).

## B.1   OVERVIEW

La-Proteína is a generative model of atomistic proteins, producing both the amino acid sequence and fully atomistic three dimensional structures. It was developed for unconditional monomer generation and atomistic motif scaffolding tasks. The model achieves state-of-the-art performance, generates structures with higher biophysical validity than baseline methods, and is designed to be efficient and scalable (by avoiding the use of computationally expensive triangular update layers).

The core modeling choice in La-Proteína is its partially latent representation of proteins. In this scheme, the coordinates of the $\alpha$-carbon atoms ($C_\alpha$) are modeled explicitly, while the sequence and all other atomistic details (non-$C_\alpha$ atoms and side chains) are encoded into continuous, fixed-size, per-residue latent variables using an autoencoder. La-Proteína then trains a flow matching model in this continuous, fixed-size space to serve as its main generative component. This partially latent approach offers several key advantages: (1) It transforms the complex generative problem from a mixed discrete/continuous space (sequence and coordinates) with variable dimensionality (different number of side chains atoms for different residue types) into a more manageable, fixed-size (per residue), and fully continuous space. La-Proteína then uses flow matching in this partially latent space as its generative model. (2) Separately modeling the $\alpha$-carbon coordinates allows for the use of distinct generation schedules during inference. This enables a faster generation scheme for $\alpha$-carbon backbone than the latent variables, which has been observed to be critical to achieve high performance in the La-Proteína empirical evaluation. (3) This design enhances scalability. The per-residue latent variables function as additional channels on top of the $\alpha$-carbon coordinates, which enables the use of powerful transformer architectures without increasing the model's sequence length representations.

La-Proteína is composed of three neural networks: an *encoder* that maps fully atomistic proteins to their partially latent representation; a *decoder* that reconstructs fully atomistic proteins from these la-

tent variables and the corresponding $\alpha$-carbon coordinates; and a *denoiser* network that predicts clean samples from noisy latent variables and $\alpha$-carbon coordinates as part of the flow matching process.

## B.2 AUTOENCODER

The autoencoder in La-Proteína is a Variational Autoencoder (VAE), responsible for mapping between the full protein structure and its partially latent representation. Following the notation used in La-Proteína, we use $L$ to denote number of residues in a protein, $\mathbf{x}^{C_\alpha} \in \mathbb{R}^{L \times 3}$ for $\alpha$-carbon coordinates, $\mathbf{x}^{\neg C_\alpha} \in \mathbb{R}^{L \times 36 \times 3}$ for non-$\alpha$-carbon atom coordinates (using the Atom37 representation without $\alpha$-carbon atoms), $\mathbf{s} \in \{0, ..., 19\}^L$ for the protein sequence, and $\mathbf{z} \in \mathbb{R}^{L \times 8}$ for the latent variables.

**Decoder.** The VAE decoder takes latent variables $\mathbf{z}$ and $\alpha$-carbon coordinates $\mathbf{x}^{C_\alpha}$ as input and outputs a distribution over sequences and coordinates of non-$C_\alpha$ atoms. Formally, the decoder is defined by the conditionally independent distribution

$$p_\phi(\mathbf{x}^{\neg C_\alpha}, \mathbf{s}|\mathbf{x}^{C_\alpha}, \mathbf{z}) = p_\phi(\mathbf{s}|\mathbf{x}^{C_\alpha}, \mathbf{z})\, p_\phi(\mathbf{x}^{\neg C_\alpha}|\mathbf{x}^{C_\alpha}, \mathbf{z}), \tag{5}$$

where $p_\phi(\mathbf{s}|\mathbf{x}^{C_\alpha}, \mathbf{z})$ is modeled as a factorized categorical distribution for the sequence (the decoder network outputs the logits) and $p_\phi(\mathbf{x}^{\neg C_\alpha}|\mathbf{x}^{C_\alpha}, \mathbf{z})$ is a factorized Gaussian with unit variance for the atomic coordinates (the decoder network outputs the mean).

**Encoder.** The encoder maps a fully atomistic protein to its corresponding latent representation. Formally, the encoder parameterizes $q_\psi(\mathbf{z}|\mathbf{x}^{C_\alpha}, \mathbf{x}^{\neg C_\alpha}, \mathbf{s})$, a factorized Gaussian. The network inputs the complete protein and outputs the mean and log-scale parameters for this distribution.

**VAE training.** The encoder and decoder are trained jointly by maximizing the $\beta$-ELBO

$$\max_{\phi, \psi} \mathbb{E}_{p_{\text{data}}(\mathbf{x}^{C_\alpha}, \mathbf{x}^{\neg C_\alpha}, \mathbf{s}), q_\psi(\mathbf{z}|...)} [\log p_\phi(\mathbf{x}^{\neg C_\alpha}, \mathbf{s}|\mathbf{x}^{C_\alpha}, \mathbf{z})] - \beta KL(q_\psi(\mathbf{z}|\mathbf{x}^{C_\alpha}, \mathbf{x}^{\neg C_\alpha}, \mathbf{s})||p(\mathbf{z})), \tag{6}$$

where the prior over latent variables $p(\mathbf{z})$ is a standard Gaussian distribution.

## B.3 PARTIALLY LATENT FLOW MATCHING

With the trained autoencoder, the primary generative task is simplified to learning the joint distribution over $\alpha$-carbon coordinates and latent variables, $p(x_{C_\alpha}, z)$, defined in a continuous, per-residue, fixed-size space. La-Proteína employs a flow matching model for this purpose, which learns to transport samples from a standard Gaussian distribution to the target data distribution. This is achieved by training a denoiser network $v_\theta$ minimizing the conditional flow matching objective

$$\min_\theta \mathbb{E}[||\mathbf{v}_\theta^x(\mathbf{x}_{t_x}^{C_\alpha}, \mathbf{z}_{t_z}, t_x, t_z) - (\mathbf{x}^{C_\alpha} - \mathbf{x}_0^{C_\alpha})||^2 + ||\mathbf{v}_\theta^z(\mathbf{x}_{t_x}^{C_\alpha}, \mathbf{z}_{t_z}, t_x, t_z) - (\mathbf{z} - \mathbf{z}_0)||^2], \tag{7}$$

where the expectation is taken over $(\mathbf{x}^{C_\alpha}, \mathbf{z}) \sim p_{\text{data}}$, noise variables $\mathbf{x}_0^{C_\alpha} \sim \mathcal{N}(0, I)$ and $\mathbf{z}_0 \sim \mathcal{N}(0, I)$, and interpolation time distributions $p_{t_x}(t_x)$ and $p_{t_z}(t_z)$. For the latter La-Proteína uses

$$p_{t_x} = 0.02\,\text{Unif}(0, 1) + 0.98\,\text{Beta}(1.9, 1) \quad \text{and} \quad p_{t_z} = 0.02\,\text{Unif}(0, 1) + 0.98\,\text{Beta}(1, 1.5). \tag{8}$$

## B.4 SAMPLING

La-Proteína generates new protein samples by numerically simulating a system of stochastic differential equations (SDEs) from $(t_x, t_z) = (0, 0)$ to $(t_x, t_z) = (1, 1)$. These equations use the score, denoted by $\zeta$, which represents the gradient of the log-probability of the intermediate densities (this score can be computed directly from the learned velocity field $v_\theta$ (Ma et al., 2024)). The SDEs are given by:

$$d\mathbf{x}_{t_x}^{C_\alpha} = \mathbf{v}_\theta^x(\mathbf{x}_{t_x}^{C_\alpha}, \mathbf{z}_{t_z}, t_x, t_z)dt_x + \beta_x(t_x)\zeta^x(\mathbf{x}_{t_x}^{C_\alpha}, \mathbf{z}_{t_z}, t_x, t_z)dt_x + \sqrt{2\beta_x(t_x)\eta_x}d\mathcal{W}_{t_x} \tag{9}$$

$$d\mathbf{z}_{t_z} = \mathbf{v}_\theta^z(\mathbf{x}_{t_x}^{C_\alpha}, \mathbf{z}_{t_z}, t_x, t_z)dt_z + \beta_z(t_z)\zeta^z(\mathbf{x}_{t_x}^{C_\alpha}, \mathbf{z}_{t_z}, t_x, t_z)dt_z + \sqrt{2\beta_z(t_z)\eta_z}d\mathcal{W}_{t_z}.$$

La-Proteína uses scaling functions $\beta_x(t_x) = 1/t_x$ and $\beta_z(t_z) = \frac{\pi}{2}\tan(\frac{\pi}{2}(1 - t_z))$ to modulate the Langevin-like term in the SDEs. It also employs noise scaling parameters $\eta_x, \eta_z < 1$ (typically set to 0.1), which functions as a form of "low temperature" sampling, which improves (co-)designability.

La-Proteína discretizes the SDEs using the Euler-Maruyama scheme. A critical aspect of the sampling process is the use of different schedules for the $\alpha$-carbons and latent variables $\mathbf{z}$; specifically, an *exponential* schedule is used for $\mathbf{x}^{C\alpha}$ and a *quadratic* schedule for $\mathbf{z}$, which effectively means that the alpha carbon coordinates are denoised at a faster rate than the the latent variables.

After the flow matching model generates a pair of $(\mathbf{x}^{C_\alpha}, \mathbf{z})$, these are passed through the VAE decoder to produce the final, fully atomistic protein structure and sequence. While the decoder defines a distribution over sequence and coordinates, La-Proteína deterministically selects the final output by taking the mean of the Gaussian distribution for the non-$C_\alpha$-carbon coordinates and the argmax of the categorical distribution's logits for the amino acid sequence $\mathbf{s}$.

## B.5 ARCHITECTURES

All three neural networks in La-Proteína (encoder, decoder, denoiser) are built upon the architecture of Proteina, which relies on transformers with pair-biased attention mechanisms. The denoiser network additionally conditions on the interpolation times $(t_x, t_z)$ by integrating adaptive layer normalization and output scaling techniques directly into its transformer blocks (Peebles & Xie, 2023).

## C  TEDDYMER DATA

For Teddymer, we first processed the entire AlphaFold database (AFDB) with TED annotations (Lau et al., 2024a), treating each domain as a chain. Since the database contains about 203M structures, which is extremely large, we filtered the database to retain only the structures belonging to AFDB50, a clustered version of AFDB generated by MMseqs2 at 50% sequence identity (Barrio-Hernandez et al., 2023). We then extracted dimers from the database and filtered those where both chains were annotated by CATH at least up to the C.A.T. level. Finally, we clustered the database to reduce redundancy, resulting in 3,556,223 clusters. The detailed procedure is described as below.

1. Teddb can be downloaded from foldseek. This database contains 203,057,497 structures and a total of 364,806,077 chains. To provide a smaller version, we generated teddb_afdb50. This database contains 47,180,623 structures and a total of 77,743,546 chains.

2. Dimers in teddb_afdb50 (123M entries): We extracted 123,606,001 dimers from multimers in teddb_afdb50. Here, a dimer is defined as a pair of chains in which each chain has at least 4 residues within 10 Å of the other in terms of CA-CA distance. To ensure reliable and precise structures, we used CATH annotations. We generated a subset of this dimer database in which both chains (originally domains) are annotated by CATH at least up to the C.A.T. level. The final database contains 10,089,503 dimers.

3. Clustered dimerdb (3.5M clusters): We clustered the dimer database with GPU-accelerated Foldseek-Multimer (Kallenborn et al., 2025; Kim et al., 2025) using both chain-level structural similarity and interface similarity. This resulted in 3,556,223 clusters: 587,687 nonsingletons and 2,968,536 singletons. The clustering was performed with the following command: foldseek easy-multimercluster teddb_afdb50_dimerdb_cath teddb_afdb50_dimerdb_cath_clustered tmp -c 0.6 –chain-tm-threshold 0.7 –interface-lddt-threshold 0.3 –cluster-mode 0

From this database, we generate the final *Teddymer*-based training dataset by filtering for interface length $> 10$, interface-pAE $< 10$ and interface-pLDDT $> 70$, leading to 510.454 cluster reps and 7.112.609 overall datapoints. We only train on the cluster representatives.

### C.1 COMPARISON TO PROTEIN-PROTEIN INTERFACES FROM PROTEIN DATABANK

To quantitatively compare the synthetic protein-protein interfaces of Complexa with experimental PDB multimers, we compute 6 metrics on the complex structures. Those metrics include: number of hydrogen bonds across the interface, hydrophobicity of the binder interface, hydrophobicity of the binder surface, shape complementarity of the binder interface, delta Solvent Accessible Surface Area (dSASA) of the binder interface, and the number of binder residues on the interface. We randomly selected 2,000 complexes from both PDB multimers and Teddymer and plot the distributions of the metrics in Fig. 12. On some metrics, we notice slightly more diversity of the PDB data, which is expected as Teddymer was created from the domain-domain interactions of synthetic AFDB

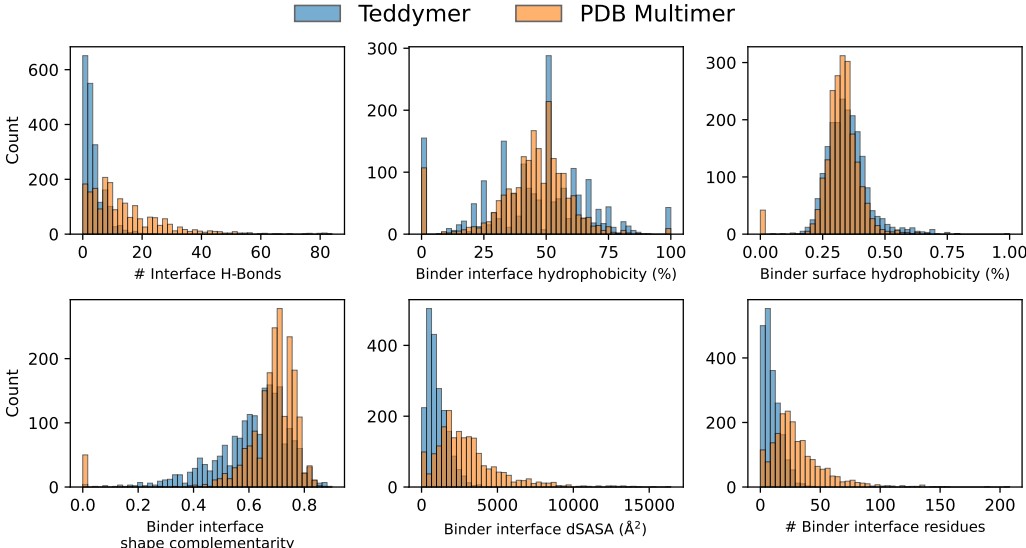

Figure 12: **Experimental vs. synthetic interfaces.** Comparison between Teddymer and PDB multimer on various protein-protein interface metrics.

monomers. Importantly, we find that there is nonetheless significant overlap in the distributions, indicating that Teddymer can serve as a source of data augmentation when training large-scale protein binder generative models like Complexa. This is aligned with our numerical results and ablations studies (Sec. I.1), which demonstrate that including Teddymer during training dramatically boosts performance.

## D  OTHER DATASETS

We also create multiple other datasets for training as outlined below; for PDB multimer and PLINDER data, we use cropping to crop and extract binder-target complexes as detailed in Sec. G.5.

### D.1  PDB DATA

We utilized the PDB multimer dataset for training protein binder models, with particular emphasis on the multimer aspect to handle not just dimers but complex multi-chain targets. We applied a rigorous three-stage filtering strategy to ensure the quality of PDB data. Firstly, the dataset was processed to only include binders with lengths between 50 and 250 residues and target chains with minimum length of 50 residues. Next, we filtered structures based on crystallographic resolution, retaining only entries with resolution better than 5 Å to ensure high-quality structural data. Thirdly, we leveraged the power of folding models to filter the PDB multimer dataset based on predicted metrics. Specifically, we folded the remaining samples after the second stage of filtering using Boltz2 (Passaro et al., 2025) and computed the ipAE (inter-chain predicted Aligned Error), complex_ipLDDT (complex predicted Local Distance Difference Test with interface residues being upweighted), and ipTM (inter-chain predicted Template Modeling score) of the folded complex structures. We then keep only the samples that satisfy $\{\text{ipAE} \leq 10\text{Å}, \text{complex\_ipLDDT} \geq 0.8, \text{ipTM} \geq 0.5\}$ and have $\geq 1$ residue at the predicted interface given 8 Å cutoff. This stage served as an alignment of the training data to the predictive power of the state-of-the-art folding models, because these metrics provide confidence estimates for the accuracy of predicted protein-protein interfaces. We want to emphasize that we included the original PDB structures in the training data and only used the folding model as a filtering tool. The final size of PDB multimer dataset is 45,856 after all filtering stages.

Table 4: Selected protein targets with their structural information, binding specifications, and data source.

| Target Name | PDB ID | Target Chain & Residues | Hotspot Residues | Binder Length | Source |
|---|---|---|---|---|---|
| BBF-14 | 9HAG | A1-112 | – | 70-250 | BindCraft |
| Betv1 | AF2 prediction | A1-159 | A24 | 70-185 | BindCraft |
| CbAgo (PAZ) | AF2 prediction | A180-276 | A268 | 70-160 | BindCraft |
| CD45 | AF2 prediction | d1 domain | A1 | 80-200 | BindCraft |
| Claudin1 | AF2 prediction | A1-188 | A31, A46, A55, A152 | 80-175 | BindCraft |
| CrSAS6 | AF2 prediction | A1-145 | – | 90-200 | BindCraft |
| Derf21 | AF2 prediction | A1-118 | A10 | 70-185 | BindCraft |
| Derf7 | AF2 prediction | A1-196 | A115 | 70-185 | BindCraft |
| HER2-AAV | 1N8Z | C23-268 | C165, C170 | 60-100 | BindCraft |
| IFNAR2 | 2LAG | A1-103 | A45, A73, A75, A77, A89, A91 | 60-175 | BindCraft |
| PD-1 | AF2 prediction | A1-115 | A33, A95, A97, A102 | 80-150 | BindCraft |
| SpCas9 | AF2 prediction | A96-174, A306-446 | A360 | 70-150 | BindCraft |
| BHRF1 | 2WH6 | A2-158 | A65, A74, A77, A82, A85, A93 | 80-120 | AlphaProteo |
| H1 | 5VLI | A1-50, A76-80, A107-111, A258-322, B1-68, B80-170 | B21, B45, B52 | 40-120 | AlphaProteo |
| IL7RA | 3DI3 | B17-209 | B58, B80, B139 | 50-120 | AlphaProteo |
| IL17A | 4HSA | A17-29, A41-131, B19-127 | A94, A116, B67 | 50-140 | AlphaProteo |
| Insulin | 4ZXB | E6-155 | E64, E88, E96 | 40-120 | AlphaProteo |
| PD-L1 | 5O45 | A17-132 | A56, A115, A123 | 50-120 | AlphaProteo |
| SARS-CoV-2 RBD | 6M0J | E333-526 | E485, E489, E494, E500, E505 | 80-120 | AlphaProteo |
| TNF-$\alpha$ | 1TNF | A12-157, B12-157, C12-157 | A113, C73 | 50-120 | AlphaProteo |
| TrkA | 1WWW | X282-382 | X294, X296, X333 | 50-120 | AlphaProteo |
| VEGFA | 1BJ1 | V14-107, W14-107 | W81, W83, W91 | 50-140 | AlphaProteo |

## D.2 PLINDER

For training small molecule binder models, we employed the PLINDER (Protein-Ligand INteraction Dataset and Evaluation Resource) dataset (Durairaj et al., 2024). Starting with the full PLINDER dataset, we applied similar filtering procedures used by the PLINDER authors for DiffDock training.

Following the established PLINDER curation protocol, we used their pre-filtered data and applied additional filtering for sanitized SMILES strings and ligands processable by RDKit. The curation process involved identifying small molecules that are not considered PLINDER ligands, including artifacts and single non-hydrogen atom or ion entities. Entities containing single atoms (excluding basic organic elements C, H, N, O, P, S) were retained as auxiliary entities referred to as "ions," regardless of oxidation state (e.g., $Fe^{2+}$, Xe, $Cl^-$).

To enrich for biologically and therapeutically relevant ligands, we excluded molecules with fewer than 2 carbons or fewer than 5 non-hydrogen atoms, those that are highly charged (absolute charge $> 2$), contain long unbranched hydrocarbon linkers ($> 12$ atoms), contain unspecified atoms, or are common experimental buffer artifacts. Artifact identification was based on common definitions and included additional buffer reagents while excluding biologically relevant cofactors and sugars.

Each detected ligand was assigned a unique canonical SMILES representation. For single residue ligands, we used the Chemical Component Dictionary (CCD) for SMILES identification.

## D.3 AFDB MONOMER PRETRAINING DATA

We utilized cluster representatives from the AlphaFold Database (AFDB) (Barrio-Hernandez et al., 2023) for both autoencoder training and latent model pretraining. The AFDB monomer data was also incorporated in a mixed manner when training small molecule binders. This dataset was filtered for a minimum average pLDDT of 80 and lengths between 32 and 256.

## E PROTEIN AND SMALL MOLECULE TARGETS

For protein targets, we aggregated the 10 protein targets from the main results of AlphaProteo (Zambaldi et al., 2024), including BHRF1, SC2RBD, IL-7RA, PD-L1, TrkA, IL-17A, VEGF-A, insulin, H1 and TNF-$\alpha$. We then added 12 of the 14 protein targets from BindCraft (Pacesa et al., 2025), including BBF-14, Betv1, CbAgo(PAZ), Claudin1, CrSAS6, Derf21, Derf7, IFNAR2, PD-1 and SpCas9) to create a total set of 22 targets (we excluded the two PD-L1 targets since PD-L1 was already represented in the AlphaProteo targets). Details below in Table 4, for each target we chose the corresponding hotspots and croppings from the relevant papers; if there were multiple hotspot specifications available we chose one of them as listed below in the table.

Table 5: Selected small molecule targets with their structural information, binding specifications, and data source.

| Target Name | PDB ID | Target Chain | Binder Length |
|---|---|---|---|
| SAM | A | A | 100 |
| FAD | B | A | 100 |
| IAI | C | A | 100 |
| OQO | D | A | 100 |

From the original publications we exclude H1, IL17A and TNF-$\alpha$ from AlphaProteo from our main benchmarks since none of the methods achieve success in 20h runtime as indicated in the target table.

This way, we end up with 19 targets in our final benchmarks. For CD45 from BindCraft, the intended hotspots are different domains instead of specific residues, which does not align with the rest of our benchmark; we therefore choose the d1 domain as the specified target and residue A1 as hotspot as indicated in the table.

For inference time scaling experiments, we consider targets either as easy or hard targets depending on the performance of all generative methods in the fixed sample budget benchmark. Based on the results of the sample budget benchmark (Fig. 20), we choose the following categories:

- Easy: IFNAR2, BHRF1, BBF14, DerF21, TrkA, PD1, Insulin, DerF7, PDL1, IL7RA, CrSAS6, Claudin1
- Hard: VEGFA, SpCas9, SC2RBD, CbAgo, CD45, BetV1, HER2-AAV

For small molecule targets, we adapt the same targets as chosen in RFDiffusion-AllAtom (Krishna et al., 2024) and BoltzDesign1 (Cho et al., 2025) as described in the table below and adopt the benchmark setup of generating 200 structures of length 100 per target.

# F    EVALUATION METRICS

**Protein target evaluation pipeline:** We generate 200 samples for every target. For each sample, we generate 8 re-designed binder sequences using ProteinMPNN (Dauparas et al., 2022), and call those them *MPNN* sequences. For models that generate all-atom binders, we additionally generate another 8 re-designed binder sequences with the interface residue identity fixed (*MPNN-FI* sequences). The re-designed sequences, plus the self-generated sequence by model (*Self* sequences, both *MPNN-FI* and *Self* sequences are exclusively to all-atoms models) are then folded by structure prediction model. We used ColabDesign implementation of AF2 for protein-protein complex folding. For this we enable target templating as well as the initial guess option in line with previous work (Bennett et al., 2023) that initialises the AF2 pair representation with the pairwise distances of the designed binder. Inter-chain predicted alignment error (ipAE, $\downarrow$), complex predicted local distance difference test (complex_pLDDT, $\uparrow$), and self-consistent RMSD of the binder (binder_scRMSD, $\downarrow$) metrics are then calculated for each folded structure. Next, we apply AlphaProteo (Zambaldi et al., 2024) filters on the best metric values of all sequences in each one of the MPNN/MPNN-FI/Self categories. A sample is considered to be *successful* in a category if it satisfies all three requirements:

1. ipAE $< 7$Å
2. complex_pLDDT $> 0.9$
3. binder_scRMSD $< 1.5$Å

**Ligand target evaluation pipeline:** We generate 200 samples for every target. For each sample we generate a single re-designed binder sequence with LigandMPNN (Dauparas et al., 2025) (*MPNN* sequence) as well as a re-designed binder sequences with the interface (within 6Å) residue identity fixed (*MPNN-FI* sequence). The *MPNN*, the *MPNN-FI*, and self-generated sequences by model (*MPNN-FI* and *Self* sequences are exclusively to all-atoms models), are then folded by RF3 (Corley et al., 2025) for ligand-protein complex co-folding. A protein-ligand sample is considered to be *successful* in a category if it satisfies all three requirements:

1. min_ipAE $< 2$
2. binder_C$\alpha$_scRMSD $< 2$Å

3. binder_aligned_ligand_scRMSD < 5Å

where min ipAE calculates the minimum entry of the cross-chain elements in the pAE matrix predicted by RF3. The ligand scRMSD is calculated after the complex has been aligned to the binder structure to make sure that the ligand pose/pocket is not altered significantly.

The successes are clustered by Foldseek (van Kempen et al., 2024) based on the binder structures, and the number of clusters are reported as the unique successes. The novelty metrics is based on the TM-Score (Zhang & Skolnick, 2004), which measures the structural similarity between generated successes and a reference set. TM-Score ranges from 0 to 1 and lower score indicates lower structural similarity hence higher novelty. In this work, we use the PDB database from Foldseek as the reference set. The novelty values reported in Tab. 2 are the averaged TM-Score of all successes for each target.

**Sampling time:** For each diffusion-based models, we set the sampling batch size as 1 and recorded the sampling time of each sample. For hallucination-based model such as BindCraft and BoltzDesign1, we recorded the sampling time as the time the algorithm took to generate 1 final optimized design. The sampling time of each target was the averaged sampling time of all 200 samples of various length. We set a time limit of 4 hours on 1 NVIDIA A100 GPU for each sample. The sampling time of AlphaDesign sometimes exceeded the time limit for large target complexes. Therefore, the averaged sampling time of AlphaDesign is the averaged sampling time of all finished samples for each target.

For evaluation of hydrogen bonds, we utilise HBPlus (McDonald & Thornton, 1994) during evaluation and search. For search we also tested tmol (Leaver-Fay et al., 2025), an open-source GPU implementation of the Rosetta energy function (Alford et al., 2017) and calculate the hydrogen bond energy at the interface of binder and target chains via this.

## G   ARCHITECTURE, MODEL, TRAINING AND SAMPLING DETAILS

In this work we introduce Complexa, an extension of the La-Proteína framework to the task of de novo binder design. Complexa operates in a conditional setting: it receives a target protein structure as input and generates a novel, fully atomistic protein binder predicted to interact with that target. Our methodology closely follows La-Proteína. We leverage exactly the same VAE architecture, trained on monomeric proteins, to learn a partially latent representation of proteins. The core modification lies in the partially latent flow matching model, which is now trained to generate a binder protein conditioned on a given target. We emphasize that our approach does not generate the full target-binder complex; the target protein serves exclusively as conditioning information. While our neural network architecture processes the binder and target jointly to learn their structural relationship, the generative output of our flow matching model is solely the binder.

### G.1   DENOISER ARCHITECTURE

Following La-Proteína, our model's denoiser is built upon a transformer architecture that uses pair-biased attention mechanisms. The denoiser network is designed to process a target protein (coordinates and sequence) alongside a potentially noisy, partially latent representation of a binder ($\alpha$-carbon coordinates and latent variables). These inputs are jointly featurized into two tensors: a sequence representation with shape $[L_{\text{binder}} + L_{\text{target}}, c_{\text{seq}}]$ and a pair representation with shape $[L_{\text{binder}} + L_{\text{target}}, L_{\text{binder}} + L_{\text{target}}, c_{\text{pair}}]$. Intuitively, the sequence representation encodes residue-level features, while the pair representation captures features between pairs of residues (e.g., inter-atomic distances). The transformer blocks iteratively update the sequence representation, with the pair representation used to bias the attention logits through linear layers (Jumper et al., 2021). Following La-Proteína, the pair representation remains static throughout the network. While prior work in structural biology has used triangular update layers to refine the pair representation (Lin et al., 2024; Abramson et al., 2024), these layers are computationally and memory-expensive, which severely limits architectural scalability. By omitting them, our model is highly efficient, a property that is particularly advantageous for applying inference-time scaling techniques. We use $c_{\text{seq}} = 768$, $c_{\text{pair}} = 256$, and 14 transformer layers. This leads to a denoiser architecture of $\sim 160M$ parameters.

Although the denoiser network processes a joint representation of the target and binder, its generative output is the velocity field for the binder. This architectural choice mirrors the approach used for unindexed motif scaffolding in La-Proteína. In that context, features corresponding to the structural

motif were concatenated to the input representation of the protein being generated, yet the model's output was similarly restricted to the generated scaffold, not the motif itself.

**Conditioning on diffusion times and CAT embeddings.** The denoiser is conditioned on several inputs beyond the protein structures themselves. First, it receives the diffusion times, $t_x$ and $t_z$, which correspond to the noise levels of the binder's $\alpha$-carbon coordinates and latent variables, respectively. Following La-Proteína, these time values are integrated into the network's transformer blocks using adaptive layer normalization and output scaling techniques (Peebles & Xie, 2023). Additionally, we condition the model on the structural class of the target and binder proteins using their CAT labels. Each CAT label is mapped to a learnable embedding, which is then incorporated into the network using the same adaptive normalization and output scaling layers, as done by Geffner et al. (2025).

### G.2 Features used as input to the Denoiser

#### G.2.1 Initial Sequence Representation.

The initial sequence representation is a concatenation of two independently constructed tensors: one for the binder, with shape $[L_{\text{binder}}, c_{\text{seq}}]$, and one for the target, with shape $[L_{\text{target}}, c_{\text{seq}}]$. For the binder, the per-residue feature vector is derived from three sources: (i) the raw coordinates of the noisy $\alpha$-carbons, (ii) the raw values of the noisy latent variables, and (iii) a Fourier embedding of the residue index. These features are concatenated and projected through a linear layer to produce the initial binder feature tensor. For the target, the feature vector is constructed from five sources: (i) a one-hot representation of the amino acid sequence, (ii) the full-atom structure (Atom37 representation), (iii) a binary indicator for hotspot residues, (iv) backbone torsion angles, binned into 20 bins between $-\pi$ and $\pi$, and (v) side chain angles, also binned into 20 bins between $-\pi$ and $\pi$. These are similarly concatenated and passed through a linear layer to yield the target's feature tensor. Finally, the binder and target sequence representations are concatenated along the length dimension to form a single input tensor of shape $[L_{\text{binder}} + L_{\text{target}}, c_{\text{seq}}]$.

#### G.2.2 Initial Pair Representation.

A similar modular construction is used for the initial pair representation, which has a final shape of $[L_{\text{binder}} + L_{\text{target}}, L_{\text{binder}} + L_{\text{target}}, c_{\text{pair}}]$. This tensor is assembled from three distinct sub-tensors: one describing interactions within the binder, one for interactions within the target, and one for the cross-interactions between the binder and target.

The **binder-binder pair representation**, with shape $[L_{\text{binder}}, L_{\text{binder}}, c_{\text{pair}}]$, is constructed following La-Proteina. Its features include (i) the relative sequence separation, one-hot encoded and limited to $\pm 64$ residues, and (ii) the pairwise distances between the noisy $\alpha$-carbon coordinates, bucketized into 1Å bins between 1Å and 30Å. These features are concatenated and projected through a linear layer.

The **target-target pair representation**, with shape $[L_{\text{target}}, L_{\text{target}}, c_{\text{pair}}]$, is derived from features that describe intra-target residue pairs. These include: (i) the relative sequence separation, one-hot encoded and limited to $\pm 64$ residues, (ii) the pairwise distances between backbone atoms, bucketized into 1Å bins between 1Å and 30Å, (iii) a binary chain index (0 if residues are from the same chain, 1 if different), and (iv) a hotspot pair variable (0 if neither residue is a hotspot, 1 if one is, and 2 if both are). These features are concatenated and fed into a linear layer for the target pair representation.

The **cross-pair representation**, describing binder-target interactions with shape $[L_{\text{binder}}, L_{\text{target}}, c_{\text{pair}}]$, is built from features capturing inter-protein residue relationships. (Note: these features are analogous to the target-target features but are computed between binder-target residue pairs). They include: (i) the pairwise distances between the binder's noisy $\alpha$-carbon coordinates and the target's backbone atoms, bucketized into 1Å bins between 1Å and 30Å, (ii) a binary chain index, and (iii) a hotspot pair variable (note that there are no hotspots in the binder, therefore in this case this variable can only be 0 or 1). These features are concatenated and fed into a linear layer for the cross-pair representation.

Finally, these three tensors are assembled into the full initial pair representation that is passed to the network. We note that this feature construction scheme is inherently compatible with multi-chain targets; such targets are simply treated as a single entity, where the chain index feature naturally distinguishes between residues belonging to different chains.

### G.2.3 SMALL MOLECULES AS CONDITIONING INFORMATION

Small molecule conditioning largely follows that of protein targets, with the key difference being that we directly featurize the fully atomistic structure, as small molecules do not have a natural sequence representation (Abramson et al., 2024).

The sequence length of our protein-ligand complexes (binder-ligand pairs) is $L_{\text{binder}} + N_{\text{target}}$, where $N_{\text{target}}$ is the number of ligand heavy atoms. The protein, or binder, components of both the sequence and pair representations are largely similar to those described in the prior section, with a minor modification: we replace residue indices with binary chain break features.

The key differences from our protein-target approach lies in how we enable fully atomistic target sequence conditioning and binder-target pair conditioning. Specifically, we focus on featurizing the target and the interactions between the binder and target.

The target feature vector is constructed from five sources: (i) a one-hot representation of the heavy atom element type, (ii) the full-atom structure ($N_{\text{target}} \times 3$), (iii) heavy atom charges, (iv) graph Laplacian positional encoding (Cao et al., 2025; Wang et al., 2023), and (v) one-hot representation of the atom name (Abramson et al., 2024). These features are concatenated and passed through a linear layer to yield the target's feature tensor of shape $[N_{\text{target}}, c_{\text{seq}}]$, which is then concatenated with that of the binder along the sequence dimension.

Similar to protein targets, the pair tensor is assembled from interactions within the binder, interactions within the target, and cross-interactions between the binder and target.

The **target-target pair representation**, with shape $[N_{\text{target}}, N_{\text{target}}, c_{\text{pair}}]$, is derived from features that describe intra-target atom pairs, including (i) pairwise distances between all atoms bucketized into 1 Å bins between 1 Å and 30 Å, (ii) the bond adjacency matrix, and (iii) the bond order matrix to indicate how atoms are bonded, if at all. These features are concatenated and fed through a linear layer to obtain the target pair representation.

The **cross-pair representation**, describing binder-target interactions with shape $[L_{\text{binder}}, N_{\text{target}}, c_{\text{pair}}]$, is built from features capturing inter-chain residue relationships, specifically the protein binder backbone (N-CA-C-CB)-ligand target pairwise distances. These features are concatenated and fed into a linear layer for the cross-pair representation. The binder-binder, binder-target, and target-target pair representations are then combined to obtain a final shape of $[L_{\text{binder}} + N_{\text{target}}, L_{\text{binder}} + N_{\text{target}}, c_{\text{pair}}]$.

### G.3 TRAINING

**Time sampling distributions.** We follow the time sampling distribution from Eq. (8) in La-Proteína.

**Data centering / translation noise.** During training, we center the complex so that the target lies at the origin. However, if we directly add noise to the binder using linear interpolation, $\mathbf{x}_{t_x}^{C_\alpha} = t_x \mathbf{x}^{C_\alpha} + (1 - t_x)\mathbf{x}_0^{C_\alpha}$ with random Gaussian noise $\mathbf{x}_0^{C_\alpha}$, then the center of mass becomes $C(\mathbf{x}_{t_x}^{C_\alpha}) = t_x C(\mathbf{x}^{C_\alpha}) + (1 - t_x)C(\mathbf{x}_0^{C_\alpha}) = t_x C(\mathbf{x}^{C_\alpha})$, where $C(\cdot)$ denotes the center of mass. In this setting, the center of the clean binder can be trivially recovered as $C(\mathbf{x}^{C_\alpha}) = C(\mathbf{x}_{t_x}^{C_\alpha})/t_x$. This shortcut allows the model to bypass learning how to position the binder relative to the target in a generalizable way. Such an issue does not arise in monomer generation, where the model only needs to learn protein structure without reasoning about global placement (Geffner et al., 2025).

To mitigate this problem, we perturb the binder's $C_\alpha$ coordinates with a random global translation $d \sim p_0^d = \mathcal{N}(d|0, c_d^2)$ during interpolation (we set $c_d = 0.2$ nm). The interpolated states then become $\mathbf{x}_{t_x}^{C_\alpha} = t_x \mathbf{x}^{C_\alpha} + (1 - t_x)(\mathbf{x}_0^{C_\alpha} + d\,\mathbf{1})$, $\quad \mathbf{z}_{t_z} = t_z \mathcal{E}(\mathbf{x}) + (1 - t_z)\mathbf{z}_0$, with priors $p_0^{C_\alpha} = \mathcal{N}(\mathbf{x}_0^{C_\alpha}|\mathbf{0}, \mathbf{I})$ and $p_0^{\mathbf{z}} = \mathcal{N}(\mathbf{z}_0|\mathbf{0}, \mathbf{I})$. During training, the model predicts velocities relative to this globally perturbed noise. This explicit *translation noise* breaks the aforementioned shortcut and forces the model to reason about the binder's global positioning. Conceptually, this can be viewed as applying an additional diffusion or flow-matching process over the binder's center of mass, gradually moving it toward the correct global position. From a Fourier perspective, standard flow and diffusion models generate low-frequency components first (Falck et al., 2025), and global translation corresponds to the lowest-frequency mode. By injecting translation noise, we force the model to refine the global position throughout generation, a technique related to prior work (Ahern et al., 2025).

### G.3.1 STAGEWISE TRAINING

This section describes the main choices regarding model training. Full training hyperparameters details can be found in Tab. 14. All models are trained using Adam (Kingma, 2014).

**Training VAE.** We train the VAE in two stages. First, on the AFDB monomer pretraining dataset (Sec. D.3) for 500k steps on 16 NVIDIA A100-80GB GPUs. Besides pretraining on synthetic monomer data, in a second stage we also finetune it on a realistic PDB dataset. The dataset is constructed by extracting single chains from monomers and multimers with max. 10 oligomeric states from PDB. We filter out chains shorter than 50, longer than 256, or resolution worse than 5.0Å, with over 50% loops or with radius of gyration over 3.0Å. This results in a dataset with 110,976 chains. We fine-tune the VAE on this dataset for 40k steps on 16 GPUs with batch size 16 per GPU.

**Training Latent Diffusion for Protein Targets.** We also use a two-stage approach to train the partially latent flow matching model. In the first stage we train on the AFDB monomer pretraining dataset for 540K steps on 32 NVIDIA A100-80GB GPUs (in this stage there is no target). On the second stage we further fine-tune the model checkpoint for 290K steps using 96 NVIDIA A100-80GB GPUs on the mixture of Teddymer and PDB dataset, filtered as detailed in App. C and App. D.1. The mixture coefficient of Teddymer and PDB is 8:2. We also fine-tune the same model checkpoint from the first stage on Teddymer-only data with additional CAT label conditioning. The CAT-conditioning model is fine-tuned on Teddymer dataset for 200K steps with 96 NVIDIA A100-80GB GPUs.

**Stagewise Training for Small Molecule Targets.** In contrast to protein targets, we employ a 3-stage training protocol for ligand conditioning. In the first stage, we pretrain our VAE on AFDB monomers with lengths in the range $[32, 512]$ for 140K steps on 32 NVIDIA A100-80GB GPUs. We increase the maximum length from the protein target variant due to our filtered PLINDER subset having an average system size of 400. In the second stage, we train the partially latent flow matching model on the same AFDB subset for 270K steps on 48 NVIDIA A100-80GB GPUs. In the third and final stage, we leverage LoRA (Hu et al., 2022) fine-tuning on a mixture of AFDB monomers and PLINDER protein-ligand complexes cropped to a length of 512 for 60K steps on 96 NVIDIA A100-80GB GPUs. Additionally, we mask out all ligand target features 50% of the time to prevent overfitting to the ligand due to the low ligand diversity of the training data.

### G.4 SAMPLING

For all protein and ligand targets we follow the same sampling algorithm used by La-Proteína (see Sec. B.4), using 400 steps to simulate the corresponding SDEs, the same Langevin scaling terms $\beta_x(t_x)$ and $\beta_z(t_z)$, noise scaling factors $\eta_x = 0.1$ and $\eta_z = 0.1$, and the exponential schedule for the binder's alpha carbon coordinates and the quadratic schedule for its latent variables. To align with the translation noise perturbing scheme in training, during inference we also sample a global translation noise $d \sim p_0^d = \mathcal{N}(d|0, c_d^2)$ with $c_d = 0.2$ nm, and add it to the initial noisy structure.

**CAT-Conditioned Sampling.** We follow Proteína to provide CAT labels during inference to support CAT conditioning. In this work, we only consider C-level conditioning for generation, *i.e.*, Mainly Alpha, Mainly Beta and Mixed Alpha Beta. Visualizations are shown in Fig. 6 and Fig. 32.

### G.5 CROPPING

Protein samples were cropped using a hierarchical procedure designed to preserve binding interfaces while limiting the number of residues for model input. The process operated as follows:

For binder chain selection, residues at intermolecular interfaces were identified based on backbone C$\alpha$ distances, using an 8.0 Å cutoff to define interface contacts. One interface residue was then selected at random as a *binder seed*, and the chain containing this residue was designated the binder chain. All other chains were considered target chains.

After that, a contiguous subsequence of the binder chain was extracted. The subsequence length was randomly chosen between 1 and 250 residues, and the binder seed residue was required to lie within this subsequence. No additional flanking residues were enforced on either side of the seed residue.

---

**Algorithm 1** Protein Binder Best-of-N Sampling

---

1: **Input:** Model $p^\phi$, success criterion function $S(\cdot)$, number of samples $N$
2: **Output:** Set of successful designs $\mathcal{S}$
3: Initialize success set $\mathcal{S} \leftarrow \emptyset$
4: **for** $i = 1$ **to** $N$ **do**
5:      Generate candidate $(\mathbf{x}^{C_\alpha}, \mathbf{z})_i \sim p^\phi(\mathbf{x}, \mathbf{z})$
6:      Decode $(\mathbf{x}^{C_\alpha}, \mathbf{z})_i$ into structure-sequence tuple $(\mathbf{x}_i, \mathbf{s}_i)$
7:      Fold and score to compute metrics for $(\mathbf{x}_i, \mathbf{s}_i)$
8:      **if** $S(\mathbf{x}_i, \mathbf{s}_i) = \text{True}$ **then**
9:         Add $(\mathbf{x}_i, \mathbf{s}_i)$ to $\mathcal{S}$
10:     **end if**
11: **end for**
12: **Return:** $\mathcal{S}$

---

Then, cropping of the target chains was performed in two sequential stages:

- **Spatial cropping:** Candidate target residues were identified as those lying within 15 Å of any residue in the cropped binder subsequence.

- **Contiguous cropping:** From these candidates, contiguous stretches of residues were selected to fit into the remaining budget. At least 50 target residues were retained, and when more residues were available than required, those closest to the binder subsequence were prioritized.

Globally, the combined binder and target were restricted to a maximum of 500 residues in total. The number of target chains was not limited, and the cropped structure retained atomic coordinates.

This procedure ensured that cropped protein samples emphasized interfacial regions while maintaining a controlled and computationally tractable input size for model training.

In the case of small molecules, we perform a spatial cropping similar to AF3 (Abramson et al., 2024) and use a modified implementation of the atomworks transform (Corley et al., 2025). First, we perform the standard spatial crop until a system size of 512 (binder residues + ligand heavy atoms). We then identify all connected components in the binder and discard any fragment with length $<20$ that has no atom in that fragment within 15Å of the ligand.

## H    INFERENCE-TIME OPTIMIZATION METHOD DETAILS

In the following, we explain our inference-time optimization approaches in detail, provide algorithms, and describe how the methods are adapted for protein binder generation in Complexa and differ from their typical implementations in the literature.

### H.1    BEST-OF-N SAMPLING

Best-of-N sampling is the simplest form of inference-time scaling. We generate $N$ independent samples from the generative model, evaluate each using a folding model, and return the subset of designs that meet pre-defined success criteria. In practice, we gradually increase the total number of generated samples $N$ up to a maximum of 51,200. Generation is performed in batch mode, while evaluation (folding and scoring) is run in single-sample inference mode following the implementation of ColabDesign and RF3. Sampling is stopped early if the wall-clock time limit is reached. Pseudo-code of the algorithm is shown in Algorithm 1.

### H.2    BEAM SEARCH

Beam search is a structured inference-time search strategy that balances exploration and exploitation by maintaining a fixed-size set of the top $N$ partial trajectories, referred to as the *beam*. Starting from $N$ independent noise samples at $t_x = 0, t_z = 0$, the algorithm proceeds iteratively by advancing each trajectory by $K$ denoising steps. At each search step, every trajectory in the beam is branched $L$ times, yielding $N \times L$ intermediate candidates. These candidates form the set $\mathcal{C}_{t_x + \Delta t_x^K, t_z + \Delta t_z^K} =$

---

**Algorithm 2** Protein Binder Beam Search with Reward Approximation

---

1: **Input:** Generative model $p^\phi$, success criterion function $S(\cdot)$, reward function $R(\cdot)$, beam width $N$, branch factor $L$, denoising step length $K$
2: **Output:** Set of successful designs $\mathcal{S}$
3: Initialize success set $\mathcal{S} \leftarrow \emptyset$
4: Initialize beam $\mathcal{B} \leftarrow \{(\mathbf{x}_{t_x=0}^{C_\alpha}, \mathbf{z}_{t_z=0})_i\}_{i=1}^N$          // $N$ initial noise samples
5: **while** $t_x < 1$ or $t_z < 1$ **do**
6:      Initialize candidate set $\mathcal{C} \leftarrow \emptyset$
7:      **for** each state $(\mathbf{x}_{t_x}^{C_\alpha}, \mathbf{z}_{t_z})$ in beam $\mathcal{B}$ **do**
8:          **for** $j = 1$ **to** $L$ **do**
9:              Run denoising for $K$ steps with $p^\phi$ to obtain intermediate state $(\mathbf{x}_{t_x+\Delta t_x^K}^{C_\alpha}, \mathbf{z}_{t_z+\Delta t_z^K})$
10:              Roll out from $(\mathbf{x}_{t_x+\Delta t_x^K}^{C_\alpha}, \mathbf{z}_{t_z+\Delta t_z^K})$ to $(t_x = 1, t_z = 1)$ to get clean sample $(\mathbf{x}, \mathbf{s})$
11:              Compute reward $r = R(\mathbf{x}, \mathbf{s})$
12:              Add $(\mathbf{x}_{t_x+\Delta t_x^K}^{C_\alpha}, \mathbf{z}_{t_z+\Delta t_z^K}, r)$ to candidate set $\mathcal{C}$
13:              **if** $S(\mathbf{x}, \mathbf{s}) = \text{True}$ **then**
14:                  Add $(\mathbf{x}, \mathbf{s})$ to success set $\mathcal{S}$
15:              **end if**
16:          **end for**
17:      **end for**
18:      Update beam: $\mathcal{B} \leftarrow \text{Top}_N(\mathcal{C})$ by reward
19:      Update time index $t_x \leftarrow t_x + \Delta t_x^K$, $t_z \leftarrow t_z + \Delta t_z^K$
20: **end while**
21: // Evaluate final beam
22: **for** each state $(\mathbf{x}^{C_\alpha}, \mathbf{z})$ in beam $\mathcal{B}$ **do**
23:      Decode $(\mathbf{x}^{C_\alpha}, \mathbf{z})$ into structure–sequence pair $(\mathbf{x}, \mathbf{s})$
24:      Compute reward $r = R(\mathbf{x}, \mathbf{s})$
25:      **if** $S(\mathbf{x}, \mathbf{s}) = \text{True}$ **then**
26:          Add $(\mathbf{x}, \mathbf{s})$ to success set $\mathcal{S}$
27:      **end if**
28: **end for**
29: **Return:** $\mathcal{S}$

---

$\left\{(\mathbf{x}_{t_x+\Delta t_x^K}^{C_\alpha}, \mathbf{z}_{t_z+\Delta t_z^K})_i\right\}_{i=1}^{NL}$, where $\Delta t_x^K$ and $\Delta t_z^K$ correspond to a block of $K$ discretized denoising steps for both backbones and latent states. We then approximate the rewards of these candidates, for example using $f_{\text{ipAE}}$ or $f_{\text{H-Bond}}$, and select the top $N$ candidates to form the updated beam. This procedure is repeated until fully denoised samples are obtained.

In contrast to prior work (Fernandes et al., 2025; Ramesh & Mardani, 2025), we do not use Tweedie's formula to estimate rewards from one-shot denoising predictions, as intermediate states are too noisy for direct evaluation. Instead, we roll out all candidates to fully denoised samples using low-temperature sampling (Geffner et al., 2025; 2026). Different from standard beam search, instead of discarding these clean samples simulated during reward estimation, we add them into the success set if they meet the criteria. This significantly improves the total number of successes.

In practice, we set $N = 4$, $L = 4$, and use 400 discretized denoising steps following the LaProteina schedule, with $K = 100$ steps per beam search update. To further scale inference-time compute, the entire beam search procedure is repeated multiple times until the wall-clock time limit is reached. Pseudo-code of the algorithm is shown in Algorithm 2.

## H.3 FEYNMAN–KAC STEERING (FK STEERING)

Feynman–Kac Steering (FK Steering) (Singhal et al., 2025) is a flexible framework for guiding diffusion-based generative models with arbitrary reward functions. The key idea is to sample from a *tilted distribution* that up-weights high-reward samples and down-weights low-reward samples: $\tilde{p}(\mathbf{x}, \mathbf{s}) \propto p^\phi(\mathbf{x}, \mathbf{s}) \exp\{\beta R(\mathbf{x}, \mathbf{s})\}$, where $R(\mathbf{x}, \mathbf{s})$ is the reward and $\beta$ is an inverse temperature controlling the bias toward high-reward trajectories. Instead of relying on fine-tuning or gradient-

---

**Algorithm 3** Protein Binder Feynman–Kac Steering with Reward Approximation

---

1: **Input:** Generative model $p^\phi$, success criterion $S(\cdot)$, reward function $R(\cdot)$
2: **Input:** Beam width $N$, branch factor $L$, denoising step length $K$, inverse temperature $\beta$
3: **Output:** Set of successful designs $\mathcal{S}$
4: Initialize success set $\mathcal{S} \leftarrow \emptyset$
5: Initialize beam $\mathcal{B} \leftarrow \{(\mathbf{x}_{t_x=0}^{C_\alpha}, \mathbf{z}_{t_z=0})_i\}_{i=1}^N$           // $N$ initial noise samples
6: **while** $t_x < 1$ or $t_z < 1$ **do**
7:      Initialize candidate set $\mathcal{C} \leftarrow \emptyset$
8:      **for** each state $(\mathbf{x}_{t_x}^{C_\alpha}, \mathbf{z}_{t_z})$ in beam $\mathcal{B}$ **do**
9:          **for** $j = 1$ **to** $L$ **do**
10:              Run denoising for $K$ steps with $p^\phi$ to obtain intermediate state $(\mathbf{x}_{t_x+\Delta t_x^K}^{C_\alpha}, \mathbf{z}_{t_z+\Delta t_z^K})$
11:              Roll out from $(\mathbf{x}_{t_x+\Delta t_x^K}^{C_\alpha}, \mathbf{z}_{t_z+\Delta t_z^K})$ to $(t_x = 1, t_z = 1)$ to get clean sample $(\mathbf{x}, \mathbf{s})$
12:              Compute reward $r = R(\mathbf{x}, \mathbf{s})$
13:              Add $(\mathbf{x}_{t_x+\Delta t_x^K}^{C_\alpha}, \mathbf{z}_{t_z+\Delta t_z^K}, r)$ to candidate set $\mathcal{C}$
14:              **if** $S(\mathbf{x}, \mathbf{s}) = $ True **then**
15:                  Add $(\mathbf{x}, \mathbf{s})$ to success set $\mathcal{S}$
16:              **end if**
17:          **end for**
18:      **end for**
19:      Compute sampling probabilities for candidates $c \in \mathcal{C}$:

$$p(c) = \frac{\exp\{\beta\, R(c)\}}{\sum_{c' \in \mathcal{C}} \exp\{\beta\, R(c')\}}$$

20:      Sample $N$ candidates from $\mathcal{C}$ according to $p(c)$; update beam $\mathcal{B} \leftarrow$ sampled candidates
21:      Update time index $t_x \leftarrow t_x + \Delta t_x^K$, $t_z \leftarrow t_z + \Delta t_z^K$
22: **end while**
23: // Evaluate final beam
24: **for** each state $(\mathbf{x}^{C_\alpha}, \mathbf{z})$ in beam $\mathcal{B}$ **do**
25:      Decode $(\mathbf{x}^{C_\alpha}, \mathbf{z})$ into structure–sequence pair $(\mathbf{x}, \mathbf{s})$
26:      Compute reward $r = R(\mathbf{x}, \mathbf{s})$
27:      **if** $S(\mathbf{x}, \mathbf{s}) = $ True **then**
28:          Add $(\mathbf{x}, \mathbf{s})$ to success set $\mathcal{S}$
29:      **end if**
30: **end for**
31: **Return:** $\mathcal{S}$

---

based guidance, FK Steering uses a particle-based inference-time procedure inspired by Feynman–Kac interacting particle systems (FK-IPS) (Moral, 2004; Vestal et al., 2008). Multiple stochastic diffusion trajectories (*particles*) are simulated in parallel, scored using intermediate reward potentials, and resampled according to their potentials at intermediate steps. This resampling eliminates low-reward particles and preferentially propagates high-reward trajectories, making it effective even when high-reward samples are rare under the original generative model.

In practice, FK Steering can be implemented similarly to beam search, but instead of selecting the top-$N$ candidates deterministically, $N$ particles are resampled according to probabilities computed from their rewards. Here we use the same hyperparameters as our beam search ($N = 4$, $L = 4$, $K = 100$), and set the inverse temperature to $\beta = 10$ to strongly favor high-reward trajectories. Pseudo-code of the algorithm is shown in Algorithm 3.

### H.4 MONTE CARLO TREE SEARCH (MCTS)

Monte Carlo Tree Search (MCTS) (Coulom, 2006) is an algorithm that incrementally builds a search tree by simulating possible future outcomes. Each iteration consists of four steps: *selection*, where the tree is traversed with a strategy to balance exploration and exploitation (Kocsis & Szepesvári, 2006); *expansion*, where a new node for an unexplored action is added; *simulation*, where a rollout estimates the expected reward from that state; and *backpropagation*, where the results are propagated

---

**Algorithm 4** Protein Binder MCTS with Stochastic Expansion

---

1: **Input:** Generative model $p^\phi$, success criterion $S(\cdot)$, reward $R(\cdot)$, number of simulations $N_{\text{sim}}$, roll-out length $K$, exploration probability $\epsilon$
2: **Output:** Set of successful designs $\mathcal{S}$
3: Initialize success set $\mathcal{S} \leftarrow \emptyset$
4: Initialize root node $v_{\text{cur}} \leftarrow (\mathbf{x}_{t_x=0}^{C_\alpha}, \mathbf{z}_{t_z=0})$
5: Initialize global time indices $t_x, t_z \leftarrow 0$
6: **while** $t_x < 1$ or $t_z < 1$ **do**
7:     **for** simulation $s = 1$ **to** $N_{\text{sim}}$ **do**
8:         Set node $v \leftarrow v_{\text{cur}}$
9:         Initialize local simulation time $t_x^{\text{sim}}, t_z^{\text{sim}} \leftarrow t_x, t_z$
10:         **while** $t_x^{\text{sim}} < 1$ or $t_z^{\text{sim}} < 1$ **do**
11:           **if** With probability $\epsilon$ or if $v$ has no children / near $t = 1$ **then**
12:             Expand new child by running $K$ denoising steps as $v_{\text{new}}$ with our model $p^\phi$:
13:             $(\mathbf{x}_{t_x^{\text{sim}}}^{C_\alpha}, \mathbf{z}_{t_z^{\text{sim}}}) \rightarrow (\mathbf{x}_{t_x^{\text{sim}}+\Delta t_x^K}^{C_\alpha}, \mathbf{z}_{t_z^{\text{sim}}+\Delta t_z^K})$
14:             Add $v_{\text{new}}$ to the children set of node $v$.
15:           **else**
16:             Select child of $v$ with highest UCB score as $v_{\text{new}}$
17:           **end if**
18:           $v \leftarrow v_{\text{new}}$
19:           Update local simulation time $t_x^{\text{sim}} \leftarrow t_x^{\text{sim}} + \Delta t_x^K, t_z^{\text{sim}} \leftarrow t_z^{\text{sim}} + \Delta t_z^K$
20:         **end while**
21:         Decode fully denoised state $(\mathbf{x}_{t_x^{\text{sim}}}^{C_\alpha}, \mathbf{z}_{t_z^{\text{sim}}}) \rightarrow (\mathbf{x}, \mathbf{s})$
22:         Compute reward $r = R(\mathbf{x}, \mathbf{s})$
23:         **if** $S(\mathbf{x}, \mathbf{s}) = \text{True}$ **then**
24:           Add $(\mathbf{x}, \mathbf{s})$ to success set $\mathcal{S}$
25:         **end if**
26:         Backpropagate reward $r$ to update statistics of ancestor nodes
27:     **end for**
28:     Move current node $v_{\text{cur}}$ to child with highest expected reward
29:     Update global time indices $t_x \leftarrow t_x + \Delta t_x^K, t_z \leftarrow t_z + \Delta t_z^K$
30: **end while**
31: **Return:** $\mathcal{S}$

---

back up the tree to update visit counts and reward estimates. Repeating this process focuses the search on promising actions, making MCTS effective for complex tasks (Granter et al., 2017).

Monte Carlo Tree Diffusion (Yoon et al., 2025; Ramesh & Mardani, 2025) treats the denoising process as a tree, where each node corresponds to a partial latent state $(\mathbf{x}_{t_x}^{C_\alpha}, \mathbf{z}_{t_z})$. Starting from the root node at $t_x = t_z = 0$, the algorithm iteratively selects, expands, and evaluates trajectories while balancing exploration and exploitation (via a exploration constant $C$) with a modified Upper Confidence Bound (UCB) score:

$$\text{UCB score}((\mathbf{x}_{t_x+\Delta t_x^K}^{C_\alpha}, \mathbf{z}_{t_z+\Delta t_z^K})_i) = \underbrace{\frac{R((\mathbf{x}_{t_x+\Delta t_x^K}^{C_\alpha}, \mathbf{z}_{t_z+\Delta t_z^K})_i)}{V((\mathbf{x}_{t_x+\Delta t_x^K}^{C_\alpha}, \mathbf{z}_{t_z+\Delta t_z^K})_i)}}_{\text{exploitation}} + C \underbrace{\sqrt{\frac{\ln(V((\mathbf{x}_{t_x}^{C_\alpha}, \mathbf{z}_{t_z})_i))}{V((\mathbf{x}_{t_x+\Delta t_x^K}^{C_\alpha}, \mathbf{z}_{t_z+\Delta t_z^K})_i)}}}_{\text{exploration}},$$

where $(\mathbf{x}_{t_x}^{C_\alpha}, \mathbf{z}_{t_z})_i$ denotes the parent state of $(\mathbf{x}_{t_x+\Delta t_x^K}^{C_\alpha}, \mathbf{z}_{t_z+\Delta t_z^K})_i$ and $V(\cdot)$ counts how often a (noisy) partially latent state has already been visited during previous searches. Empirically, each node maintains a set of children representing possible next states. When expanding a leaf node, several denoising steps are performed to generate a new child, which is then added to the node's children set. For every node, we record both the visit count and the average reward obtained after visiting that node. Once a trajectory is fully simulated to completion, its final reward is backpropagated to all ancestor nodes, updating their visit counts and average rewards to guide future search decisions.

In classical MCTS with a finite action space, each node has a fixed number of children (branching factor $L$), and the search iteratively explores these discrete options. However, in flow matching, the

state space is effectively continuous and unbounded, making traditional MCTS with a fixed branching factor inefficient. To address this, our implementation merges selection and expansion into a single stochastic decision at each node. At each step, with probability $\epsilon$, a new child is expanded by running $K$ denoising steps from the current node; with probability $1 - \epsilon$, the child with the highest expected reward is selected. If the current node has no children or the next step reaches $t_x = t_z = 1$, a new child is always expanded. Once the current node reaches $(t_x, t_z) = 1$, it will be decoded into a structure–sequence pair $(\mathbf{x}, \mathbf{s})$, and the reward is computed. This reward is then backpropagated to all ancestor nodes to update visit counts and expected rewards. Fully denoised states meeting the success criteria are added to a success set $\mathcal{S}$, similar to beam search.

Here, the probability $\epsilon$ plays a role analogous to $L$: instead of deterministically expanding every possible child, we stochastically control the growth of the tree. Specifically, if a node is visited $V$ times, it will have, in expectation, $V \cdot \epsilon$ children over the course of the search. Thus, $\epsilon$ acts as a soft, probabilistic branching factor, balancing between aggressive exploration (large $\epsilon$) and focused refinement of promising states (small $\epsilon$). The constant $C$, by contrast, governs the exploration-exploitation trade-off among the children that have already been expanded. A larger $C$ encourages exploration of less-visited children, while a smaller $C$ prioritizes selecting the currently best-performing states based on their rewards. Together, $\epsilon$ and $C$ jointly determine the behavior of the search.

Another advantage of the modified stochastic MCTS is its *compatibility with batched inference*. In practice, our neural network implementation only supports denoising a batch of samples at the same time step. In standard MCTS, different trajectories in the search tree may be at different time steps, making it difficult to perform batched denoising efficiently. By using the same $\epsilon$ across all states in a batch, our algorithm ensures that the denoising steps for multiple nodes can be executed in parallel, even when expanding different children stochastically. This design not only preserves the exploration–exploitation behavior of MCTS but also maintains the computational efficiency required for diffusion models that rely on synchronized batched evaluation.

In practice, we set $K = 100$, $\epsilon = 0.5$, $C = 1.0$, and run $N_{\text{sim}} = 20$ simulations per decision step. After each round of simulations, the current node is moved to the child with the highest expected reward, and the global time indices $t_x, t_z$ are updated to match the selected child. This procedure repeats until the state reaches $(t_x, t_z) = 1$, producing a set of successful fully denoised states. Pseudo-code of the algorithm is presented in Algorithm 4.

## H.5 GENERATE & HALLUCINATE (G&H)

In addition to search-based inference methods, we explore a simpler approach that combines generative initialization with sequence refinement, which we call *Generate & Hallucinate (G&H)*. We first generate candidate binder sequences using our generative model, and then refine these sequences using the BindCraft hallucination framework (Pacesa et al., 2025).

BindCraft optimizes binder sequences using the ColabDesign implementation of AlphaFold2 (AF2), minimizing a multi-term loss that accounts for binder confidence, interface confidence, predicted alignment errors (pAE) within the binder and between binder and target, residue contact constraints, radius of gyration, backbone helicity, and optional termini constraints. Gradients are backpropagated through AF2 to iteratively update the sequence, with loss weights set according to the original BindCraft. The number of recycles in AF2 prediction is set to 3 to ensure accurate approximation.

After generating sequences with our model, BindCraft is initialized in "wild type" mode using the generated sequence and refines it through staged optimization. We focus on the last three stages, which are used in our implementation:

- **Stage 2 (Softmax normalization):** Sequence logits are converted to probabilities via softmax with temperature annealing, $\text{softmax}(\text{logits}/T)$, where the temperature $T$ decreases over iterations as $T = 1 \times 10^{-2} + (1 - 1 \times 10^{-2}) \cdot (1 - (\text{step} + 1)/\text{iterations})$. Learning rates are scaled according to this schedule. This stage runs for 45 iterations.
- **Stage 3 (Straight-through estimator):** One-hot discrete representations are used for forward passes while gradients are backpropagated through softmax. This allows refinement of discrete sequences while maintaining differentiability. This stage runs for 5 iterations.
- **Stage 4 (Discrete sequence optimization):** Sequences are fully discretized. Random mutations are sampled from the softmax distribution of the previous stage, and only mutations that

Table 6: **Translation noise and Teddymer ablation studies.** average unique successes and the number of times each method ranks best across 19 targets. Results are shown for Complexa, and variants without Teddymer data and without translation noise.

| Model | # Unique Successes ↑ | | | # Times Best Method ↑ | | |
|---|---|---|---|---|---|---|
| | Self | MPNN-FI | MPNN | Self | MPNN-FI | MPNN |
| Complexa | **9.10** | **13.5** | **14.4** | **19** | **13** | **14** |
| Complexa w/o Teddymer | 0.15 | 1.68 | 3.84 | 0 | 0 | 0 |
| Complexa w/o Translation Noise | 1.47 | 3.89 | 3.73 | 0 | 6 | 5 |

Table 7: **Translation noise and Teddymer ablation studies (detailed results).** Unique successes across 19 targets for Complexa and its variants without Teddymer data or translation noise, evaluated with different sequence redesign methods. Results are visualized in Fig. 13.

| Target | Complexa | | | Complexa w/o Teddymer | | | Complexa w/o Translation Noise | | |
|---|---|---|---|---|---|---|---|---|---|
| | MPNN | MPNN-FI | Self | MPNN | MPNN-FI | Self | MPNN | MPNN-FI | Self |
| PD-L1 | 16 | 14 | 9 | 8 | 4 | 0 | 1 | 4 | 1 |
| BHRF1 | 26 | 29 | 21 | 18 | 7 | 2 | 28 | 28 | 14 |
| IFNAR2 | 51 | 52 | 39 | 5 | 2 | 0 | 4 | 2 | 0 |
| BBF14 | 24 | 25 | 20 | 12 | 10 | 0 | 4 | 4 | 0 |
| DerF7 | 24 | 16 | 10 | 7 | 0 | 0 | 1 | 2 | 2 |
| DerF21 | 35 | 31 | 22 | 5 | 4 | 1 | 3 | 4 | 1 |
| PD1 | 27 | 21 | 15 | 2 | 0 | 0 | 6 | 4 | 4 |
| Insulin | 16 | 20 | 8 | 1 | 0 | 0 | 3 | 1 | 0 |
| IL7RA | 6 | 12 | 6 | 2 | 2 | 0 | 1 | 0 | 0 |
| Claudin1 | 4 | 4 | 2 | 0 | 0 | 0 | 2 | 0 | 0 |
| CrSAS6 | 12 | 8 | 3 | 4 | 1 | 0 | 0 | 1 | 0 |
| TrkA | 25 | 20 | 16 | 4 | 1 | 0 | 3 | 4 | 0 |
| SC2RBD | 0 | 0 | 0 | 0 | 0 | 0 | 1 | 1 | 0 |
| CbAgo | 0 | 0 | 0 | 0 | 0 | 0 | 1 | 2 | 0 |
| CD45 | 0 | 0 | 0 | 2 | 0 | 0 | 3 | 3 | 0 |
| BetV1 | 0 | 0 | 0 | 1 | 0 | 0 | 2 | 1 | 2 |
| HER2-AAV | 0 | 0 | 0 | 0 | 0 | 0 | 0 | 0 | 0 |
| SpCas9 | 3 | 4 | 1 | 0 | 0 | 0 | 2 | 7 | 1 |
| VEGFA | 5 | 2 | 1 | 0 | 1 | 0 | 2 | 3 | 1 |

improve the design loss are retained. The number of mutations per iteration is proportional to the binder length ($0.01\times$ binder length). This stage is performed for 15 iterations.

In our implementation, generative samples are refined using either (i) stages 2+3+4 or (ii) stage 4 alone. In the main paper, we use (ii) as our optimization method and we show a comparison between the two choices in Sec. I.3. This approach provides a simple and computationally efficient combination of generative initialization with principled hallucination-based sequence refinement, and can be flexibly integrated with any of the search algorithms described above.

# I  ABLATION STUDIES AND ADDITIONAL EXPERIMENTS

In this section, we present our ablation studies and additional experimental results. We perform two experiments to assess the importance of two core choices in our Complexa model training: the impact of the Teddymer training data in Sec. I.1, and translation noise in Sec. I.2. We also explore different choices of using BindCraft optimization stages in Generate and Hallucinate in Sec. I.3. Finally, we provide detailed results for designing binders for very challenging targets in Sec. I.4, generative model benchmark results in Sec. I.5, inference-time scaling results across all targets in Sec. I.6 and Sec. I.7, and interface hydrogen-bond optimization results for all targets in Sec. I.8.

## I.1  ABLATION EXPERIMENT: TRAINING WITHOUT TEDDYMER

To assess the impact of Teddymer on model performance, we conduct an ablation study in which the model is trained only on the PDB data described in Sec. D.1. All other model configurations and training hyperparameters are kept identical to the base Complexa; only the dataset is changed. The model is trained for 235k steps with 64 GPUs and batch size 6 per GPU and evaluated using the same protocol as the base model. Overall results are reported in Tab. 6, per-target results are shown in

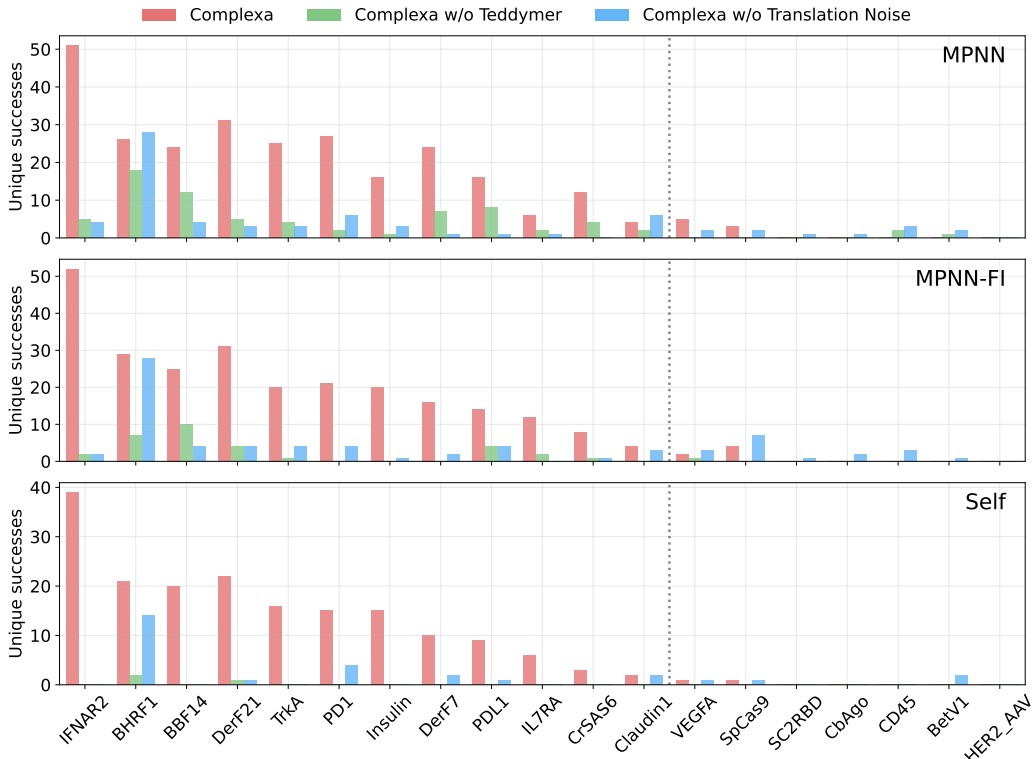

Figure 13: **Translation noise and Teddymer ablation studies.** Unique successes of different Complexa variants (base, without Teddymer data, and without translation noise) on easy targets (left of the dashed line) and hard targets (right of the dashed line).

Tab. 7 and visualized in Fig. 13. As clearly shown, the number of unique successes drops significantly across most targets, demonstrating the critical importance of including Teddymer in training.

### I.1.1 TEDDYMER ABLATION WITH SEPARATE FOLDING MODELS FOR SCORING

Teddymer is derived from structures from the AFDB (Varadi et al., 2021), which were predicted using AlphaFold2 (Jumper et al., 2021), and our main in-silico success criteria use ipAE scores calculated with the ColabDesign implementation (Ovchinnikov et al., 2025) of AlphaFold2-Multimer (Evans et al., 2022) (following Zambaldi et al. (2024)). To alleviate concerns that we are overly relying on AlphaFold2-type models for both data creation (Teddymer) and model evaluation, we additionally assess the benefit of including Teddymer in the training data by evaluating two variants of our model (training with and without Teddymer) with distinct third-generation structure prediction models RosettaFold-3 (Corley et al., 2025) (RF3) and Boltz-2 (Passaro et al., 2025), instead of AlphaFold2-Multimer. In other words, we are repeating the Teddymer ablation study using ipAE scores from RF3 and Boltz-2 for evaluation, applied to the same generated binders as before. See detailed per target unique success results in Fig. 14 for Boltz-2 and Fig. 15 for RF3. The aggregated results across all targets are in Tab. 8 for Boltz-2 and Tab. 9 for RF3. Multiple sequence alignments (MSAs) (Steinegger & Söding, 2017) are used for the targets when folding with Boltz-2. Template conditioning is used for targets when folding with RF3. We use the same threshold values for all metrics described in Sec. F to filter for success. We find through this additional ablation study that training on Teddymer shows clear benefits, even when evaluating with either RF3 or Boltz-2, thereby alleviating concerns about relying too much on AlphaFold2-type models specifically.

Note that overall absolute success rates vary between the evaluations with the different folding models. This is in part because we used the same ipAE success thresholds as before, without recalibration specifically for RF3 and Boltz-2. Also the use of MSAs and target structure templates affects ipAE scores. We did not bother to adjust and re-calibrate ipAE thresholds here, because we are only

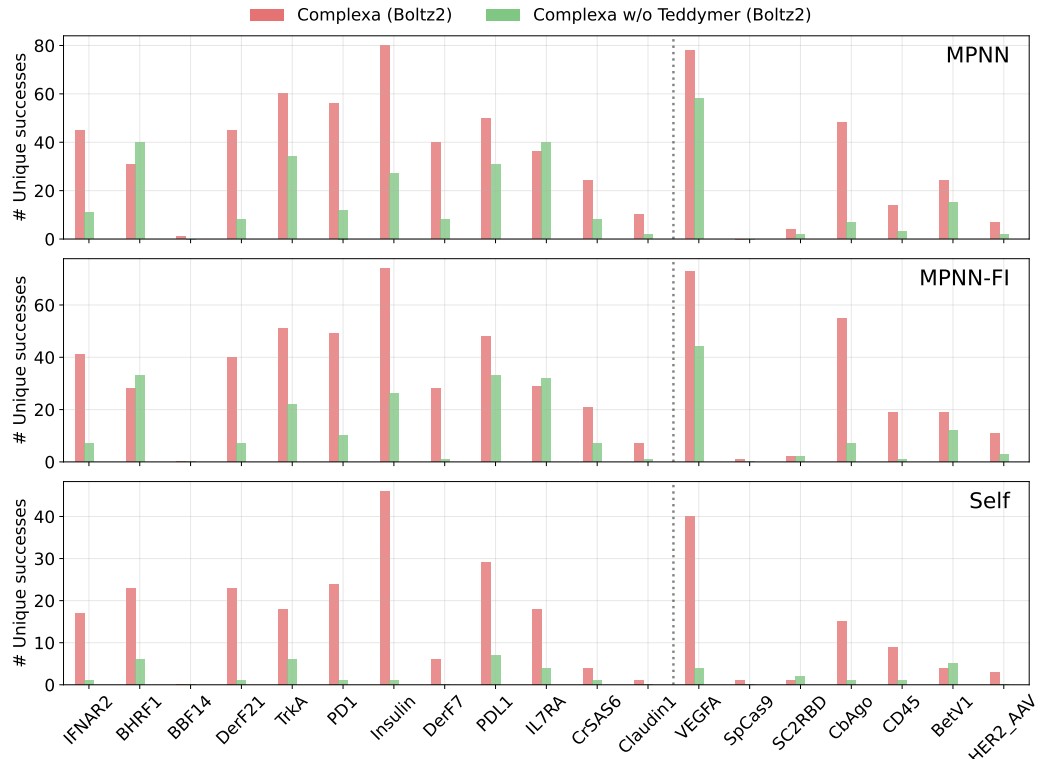

Figure 14: **Teddymer ablation study with Boltz-2.** Boltz-2-based evaluation of the unique successes of different Complexa variants (trained with vs. without Teddymer data) on easy targets (left of the dashed line) and hard targets (right of the dashed line).

Table 8: **Teddymer ablation study with Boltz-2.** Boltz-2-based evaluation of the average unique successes across 19 targets. Results are shown for Complexa and the variant trained without Teddymer data.

| Model | # Unique Successes ↑ | | |
|---|---|---|---|
| | MPNN | MPNN-FI | Self |
| Complexa | **34.4** | **31.4** | **14.8** |
| Complexa w/o Teddymer | 16.2 | 13.1 | 2.2 |

interested in the relative performance of models trained with and without Teddymer for the same RF3-based or Boltz-2-based evaluations.

## I.2 ABLATION EXPERIMENT: TRAINING WITHOUT TRANSLATION NOISE

As introduced in Sec. 3.2, the base Complexa objective incorporates translation noise to encourage reasoning over the global positioning of the protein. To evaluate its effect, we train the model using the same dataset and hyperparameters as the base model, but without the translation noise. Evaluation results are reported in Tab. 6, with per-target results in Tab. 7 and visualizations in Fig. 13. The results show that, in general, removing translation noise substantially reduces the number of unique successes across most targets. Some hard targets show minor improvements, likely because the additional noise in the objective can make learning already difficult structures even harder. Overall, these findings confirm that the translation noise component is beneficial for model generalization.

## I.3 ABLATION EXPERIMENT: BINDCRAFT STAGES IN GENERATE & HALLUCINATE

For our Generate & Hallucinate algorithm proposed in Sec. 3.2 (detailed in Sec. H.5), we choose to use our generated sequence as a prior instead of from some random initialization and run discrete sequence optimization through mutation (stage 4 only) with BindCraft. Here we explore the options

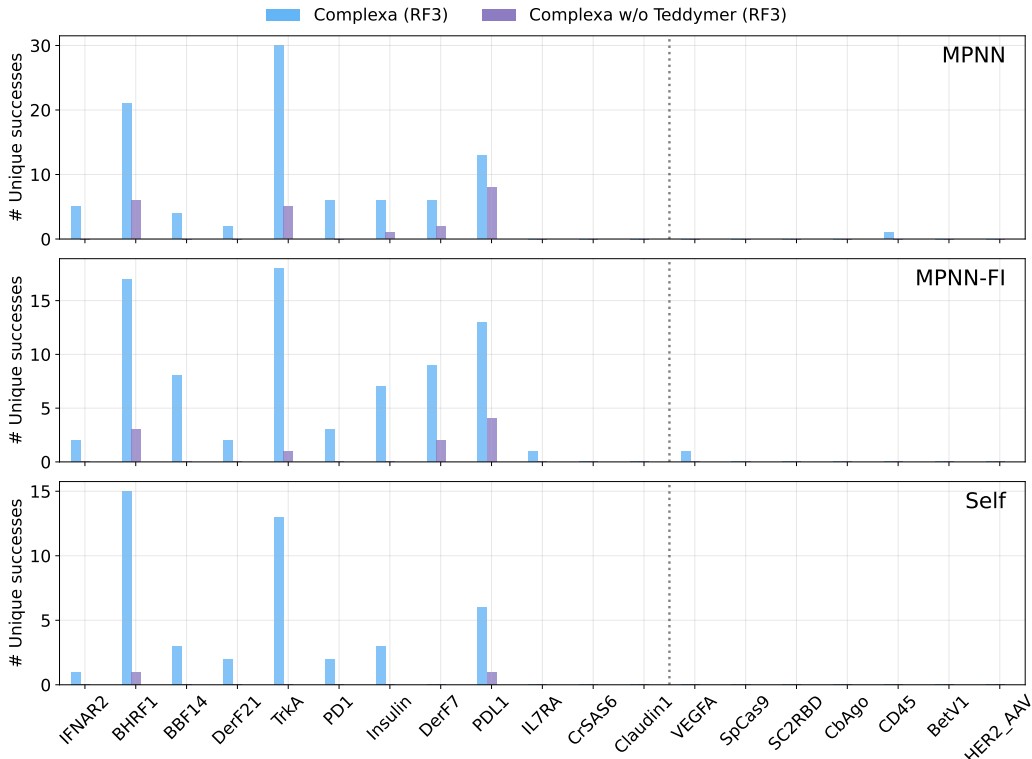

Figure 15: **Teddymer ablation study with RF3.** RF3-based evaluation of the unique successes of different Complexa variants (trained with vs. without Teddymer data) on easy targets (left of the dashed line) and hard targets (right of the dashed line).

Table 9: **Teddymer ablation study with RF3.** RF3-based evaluation of the average unique successes across 19 targets. Results are shown for Complexa and the variant trained without Teddymer data.

| Model | # Unique Successes ↑ | | |
|---|---|---|---|
| | MPNN | MPNN-FI | Self |
| Complexa | **4.9** | **4.3** | **2.4** |
| Complexa w/o Teddymer | 1.2 | 0.5 | 0.1 |

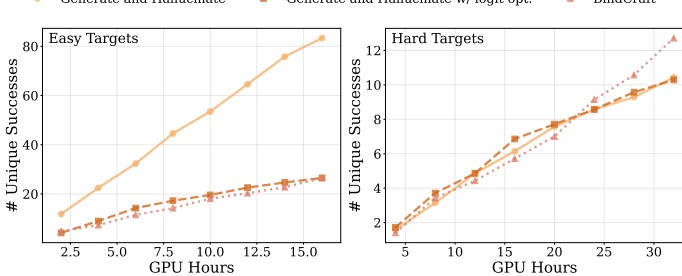

Figure 16: **Generate & Hallucinate ablation study.** Numbers of unique successes against inference-time GPU hours, for Generate & Hallucinate, its variant with logit optimization, and BindCraft, averaged over 12 easy and 7 hard targets.

whether to do the sequence logits optimization (stages 2+3+4). We run the experiments with the same setting as Generate & Hallucinate, but also add the logit optimization stages. We compare the two methods under the same inference-time compute budget following our experiments in Sec. 4.2. The averaged results are shown in Fig. 16 and per-target results are shown in Fig. 17 and Fig. 18. On easy targets, we see that our Generate & Hallucinate outperforms its variant with logit optimization

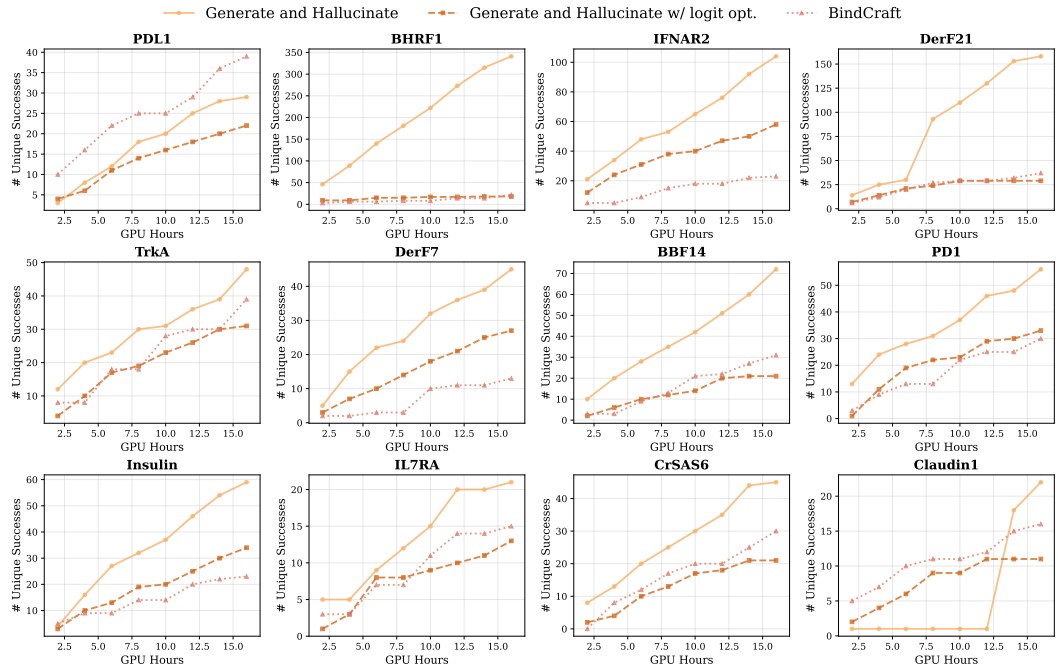

Figure 17: **Generate & Hallucinate ablation study (easy targets).** Number of unique successes over 12 easy targets against inference-time GPU hours, for Generate & Hallucinate, its variant with logit optimization, and BindCraft. Successes are measured with self-generated sequences and inference-time compute includes both generation and evaluation time.

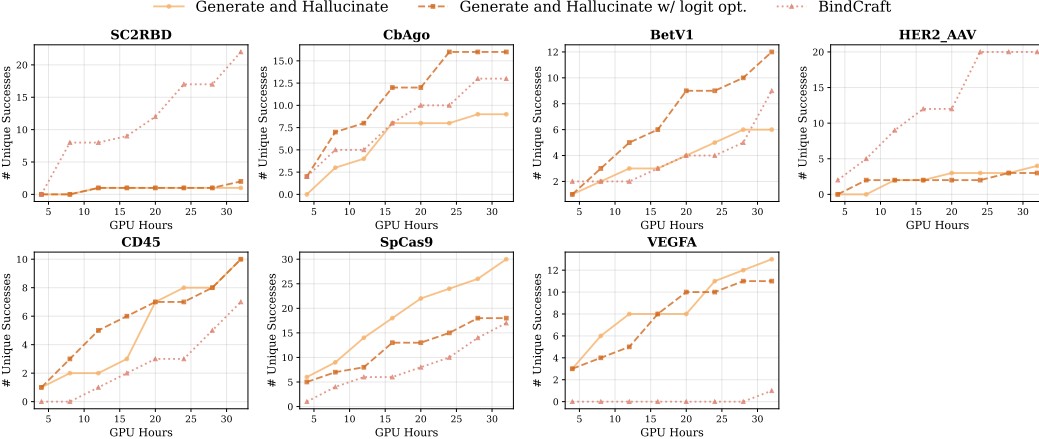

Figure 18: **Generate & Hallucinate ablation study (hard targets).** Number of unique successes over 7 hard targets against inference-time GPU hours, for Generate & Hallucinate, its variant with logit optimization, and BindCraft. Successes are measured with self-generated sequences and inference-time compute includes both generation and evaluation time.

and BindCraft on most targets and by a large margin on the average performance, which proves that our generative model provides a strong prior for sequence optimization without the need to run logit optimization. On hard targets, these three methods compare on par in terms of average performance and shine on different targets. We argue that this is highly affected by whether our generative model can provide a good prior on these challenging targets. Overall, we find adding logit optimization stages does not show clear advantages, so we decide not to include them.

Having presented our numerical ablation results on the hallucination stages of Generate & Hallucinate, we would now like to discuss them in more detail. Why is logit optimization not universally helpful, even on easy targets when starting from a good initial binder candidate? The logit optimization stages of hallucination are intrinsically slow as they rely on gradients that need to be backpropagated through the structure prediction model that provides the reward. Moreover, the logit optimization corresponds to an approximate discrete sequence relaxation, which is necessary for the gradient-based updates. Therefore, the logit optimization is an approximate and coarse process, but it can help steering a random input towards reasonable sequence logits relatively quickly, and these logits can then be further refined with the later stages. But in our case, especially for easy targets, we already start the optimization from a good sequence generated by the base model. The approximate early logit optimization stages of hallucination therefore do not offer additional benefits, and the corresponding compute is better spent on carefully refining the sequence via stage 4 mutations instead. For hard targets, the situation can change: The initial binder candidate by Complexa's generative model is often not strong and significant refinement is necessary, for which the early logit optimization stages that more radically update the sequence can be helpful as well, before switching to stage 4 mutation-based refinement. Note, however, that these mutations are random and unguided. This stands in contrast to the efficient inference-time search algorithms developed for Complexa. The optimization in algorithms like Beam Search is effectively guided by Complexa's pre-trained generative prior, and the generation trajectories are steered early on. Hallucination can only ever modify a potentially flawed initial sample. This is exactly a key point of Complexa's framework: Directly integrating generative pre-training with inference-time optimization for binder design enables enhanced and more scalable binder design compared to previous hallucination methods.

### I.4 INFERENCE-TIME SCALING FOR VERY HARD TARGETS

In Sec. 4.2, we explore inference-time search for three highly challenging multi-chain targets: TNF-$\alpha$, H1, and IL17A. For these targets, we observe that AlphaFold2-Multimer often produces low-confidence predictions, resulting in many binders with $\text{ipAE} < 7$ but $\text{plddt} < 90$. To address this issue, we incorporate $\text{plddt}$ into our reward function. Specifically, we normalize the $\text{ipAE}$ score to the range $[0, 1]$ by dividing it by 31, then add it to $\text{plddt}$ to form the final reward. For TNF-$\alpha$ and H1, we run MCTS with parameters $K = 100$, $\epsilon = 0.5$, $C = 1$, and $N_{\text{sim}} = 20$. After the MCTS search, we apply Generate and Hallucinate for local sequence optimization, increasing the number of mutations per iteration to $0.05\times$ the binder length to fully explore the local sequence neighborhood. For IL17A, we use $K = 50$, $\epsilon = 0.5$, $C = 1$, and $N_{\text{sim}} = 20$, allowing for a deeper search tree that more effectively leverages the available inference-time compute. The subsequent Generate and Hallucinate steps use the same $0.05\times$ binder length mutation rate.

Using this approach, we identify 15 unique successes for TNF-$\alpha$ within 475 GPU hours, 7 unique successes for H1 within 604 GPU hours, and 1 unique success for IL17A within 387 GPU hours. The inference-time search scaling curves for the three very hard targets are shown in Fig. 19. Note that for IL17A, we found one in-silico success early on during the process, but not another one. We show visualizations of these binders in Fig. 30. Note that as all samples go through BindCraft refinement for sequence optimization, which changes the underlying sequences, we visualize the refolded structures predicted by AlphaFold2-Multimer instead of our generated structure. These results highlight the strength and flexibility of our framework for tackling extremely challenging binder design tasks through inference-time search and optimization.

### I.5 EXTENDED RESULTS: GENERATIVE MODEL BENCHMARK WITH SEQUENCE RE-DESIGN

In the generative model benchmark we generate 200 samples per model for each of the 19 targets and evaluate them in three different ways: for backbone design models RFDiffusion and Protapardelle-1c we just run ProteinMPNN eight times for each backbone and choose the best refolded sequence; for APM and Complexa we also test MPNN-redesign but keeping the interface fixed or just evaluation of the full generated sequence.

For MPNN-redesign (Fig. 20 top plot), one can see that Complexa clearly outperforms the baselines on most tasks. For MPNN-redesign with fixed interface, performance for Complexa is similar, while APM performs significantly worse, indicating that our generated interfaces are of higher quality. Finally, for model generated sequences the relevant APM baseline collapses for most targets to zero

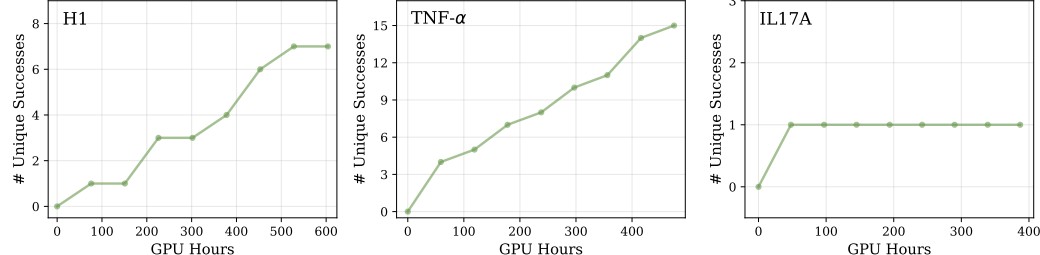

Figure 19: **H1, TNF-α, and IL17A targets.** Inference-time optimization compute scaling plots for three very hard targets. See Sec. I.4 for details.

Table 10: **De novo binder design performance for all protein targets**. Unique successes for different models and three sequence redesign methods per target.

| Target | RFDiffusion | Protpardelle-1c | APM | | | Complexa (Ours) | | |
|---|---|---|---|---|---|---|---|---|
| | MPNN | MPNN | MPNN | MPNN-FI | Self | MPNN | MPNN-FI | Self |
| PD-L1 | 8 | 2 | 1 | 0 | 0 | 16 | 14 | 9 |
| BHRF1 | 5 | 4 | 20 | 10 | 5 | 26 | 29 | 21 |
| IFNAR2 | 11 | 2 | 3 | 3 | 1 | 51 | 52 | 39 |
| BBF14 | 2 | 0 | 0 | 0 | 0 | 24 | 25 | 20 |
| DerF7 | 5 | 0 | 0 | 0 | 0 | 24 | 16 | 10 |
| DerF21 | 4 | 4 | 11 | 8 | 0 | 35 | 31 | 22 |
| PD1 | 7 | 0 | 1 | 0 | 0 | 27 | 21 | 15 |
| Insulin | 8 | 1 | 5 | 3 | 0 | 16 | 20 | 15 |
| IL7RA | 4 | 0 | 1 | 1 | 0 | 6 | 12 | 6 |
| Claudin1 | 2 | 0 | 11 | 1 | 0 | 4 | 4 | 2 |
| CrSAS6 | 8 | 0 | 1 | 1 | 0 | 12 | 8 | 3 |
| TrkA | 7 | 0 | 0 | 0 | 0 | 25 | 20 | 16 |
| SC2RBD | 2 | 0 | 0 | 0 | 0 | 0 | 0 | 0 |
| CbAgo | 0 | 0 | 3 | 1 | 0 | 0 | 0 | 0 |
| CD45 | 1 | 0 | 0 | 1 | 0 | 0 | 0 | 0 |
| BetV1 | 3 | 0 | 3 | 0 | 0 | 0 | 0 | 0 |
| HER2-AAV | 1 | 0 | 0 | 0 | 1 | 0 | 0 | 0 |
| SpCas9 | 3 | 1 | 0 | 0 | 0 | 3 | 4 | 1 |
| VEGFA | 0 | 0 | 0 | 0 | 0 | 5 | 2 | 1 |

Table 11: **De novo binder design performance of models for all small molecule targets**. RFdiffusionAA only produces backbones and as a result can only be evaluated under full LigandMPNN re-design. We report the number of FoldSeek clusters out of the successful subset of 200 samples of length 100.

| Target | Model | # Unique Successes ↑ | | |
|---|---|---|---|---|
| | | Self | MPNN-FI | MPNN |
| SAM | RFdiffusionAA | - | - | 2 |
| | Complexa | **10** | **8** | **15** |
| OQO | RFdiffusionAA | - | - | 3 |
| | Complexa | **6** | **12** | **9** |
| FAD | RFdiffusionAA | - | - | 5 |
| | Complexa | **17** | **14** | **17** |
| IAI | RFdiffusionAA | - | - | 8 |
| | Complexa | **19** | **14** | **22** |

successes, while Complexa still produces many diverse successes for a variety of targets, highlighting that it is actually a practically useful co-design model.

Tab. 11 provides the per-target unique successes for Complexa compared to RFDiffusion-AllAtom. Here we report the unique successes for a single LigandMPNN re-design (both full and fixed interface). For some tasks, using LigandMPNN to re-design the interface or the entire sequence is beneficial, however the sequences generated by Complexa always perform well and outperform the baseline. RFDiffusion-AllAtom can only be evaluated with full sequence re-design since the model only generates backbones. Overall we find that the sequences generated from Complexa score well, with performance scaling as a function of inference-time compute (see Fig. 23).

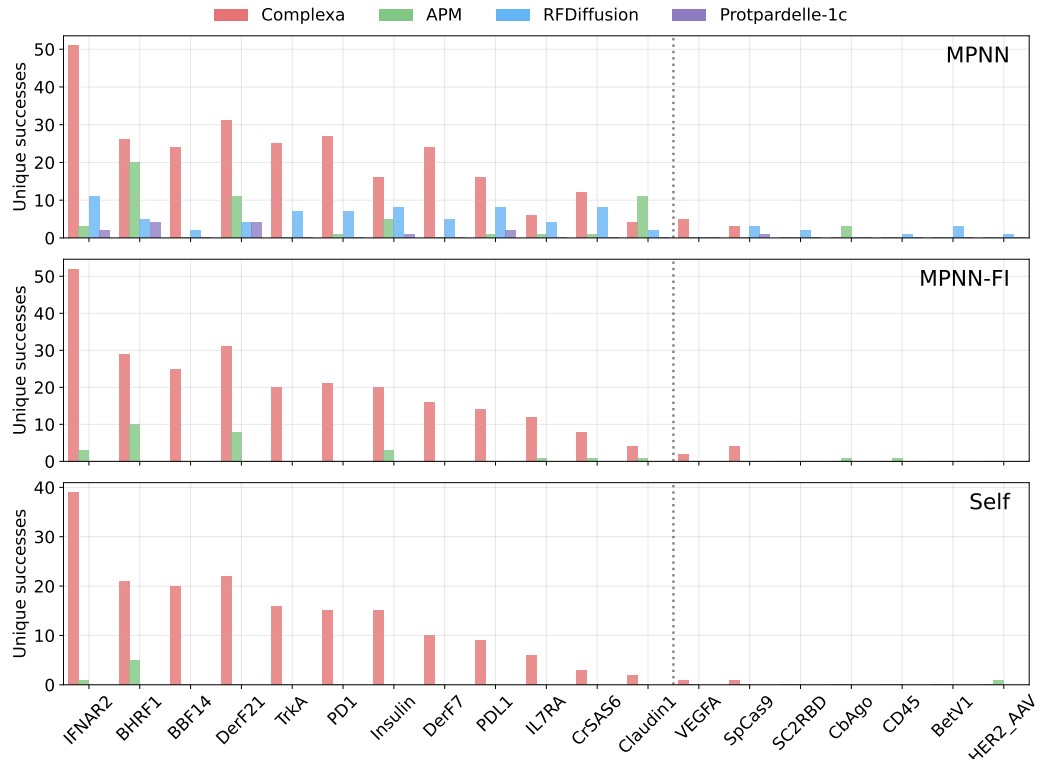

Figure 20: **Unique successes for different models with three sequence redesign methods** on easy targets (left of the dashed line) and hard targets (right of the dashed line). Detailed results are in Tab. 10

### I.6 EXTENDED RESULTS: INFERENCE-TIME SEARCH FOR ALL PROTEIN TARGETS

Accompanied by the averaged inference-scaling results shown in Sec. 4.2, we present per-target results for 12 easy targets (Fig. 21) and 7 hard targets (Fig. 22). We use the ipAE value from AlphaFold2-Multimer as the reward and run different algorithms to a fixed inference-time compute budget. We use 16 GPU hours for easy targets and 32 GPU hours for hard targets at maximum.

For the easy targets, our inference-time search algorithms consistently outperform the three hallucination baselines. This is expected: while hallucination-based methods such as BindCraft often achieve high success rates, they require significantly more compute due to the large number of backpropagations through AlphaFold2. Among the 12 easy targets, the Best-of-N strategy achieves the highest number of unique successes on 8 targets. This suggests that for easy cases where our base model already performs well, advanced algorithms like beam search may be unnecessary, as they risk wasting compute on generating redundant successes. In these scenarios, simply sampling repeatedly with Best-of-N is both efficient and effective.

For moderately challenging targets, such as BHRF1, IL7RA, and Claudin1, where the base model produces fewer successes under a fixed budget, more sophisticated algorithms become beneficial. By fully exploiting promising regions of the search space, methods like Beam Search or Monte Carlo Tree Search can discover additional high-quality solutions. This effect is even more pronounced on the 7 hard targets (Fig. 22), where the base model alone struggles to generate successes. Here, advanced search algorithms, including Beam Search, Feynman-Kac Steering, and MCTS, demonstrate clear advantages by systematically exploring and refining candidate solutions.

However, on two extremely difficult targets, SARS-CoV-2 RBD and HER2-AAV, our approach yields fewer unique successes than BindCraft, and on BetV1, BoltzDesign is more efficient. These results highlight the complementary strengths of different methods. While our inference-time search framework is the most effective across most targets, hallucination-based approaches remain valuable. Because they do not require training and directly optimize for the final success criteria, they avoid potential overfitting and can occasionally outperform trained models on highly challenging targets.

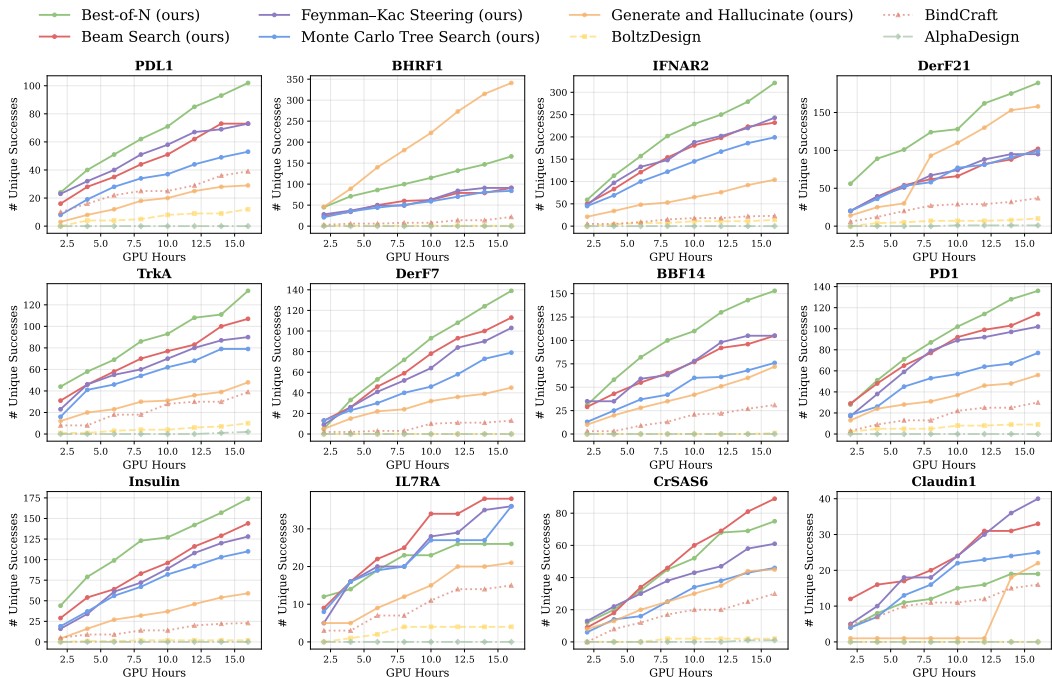

Figure 21: **Unique successes against inference-time GPU hours on 12 easy targets.** Successes are measured with self-generated sequences and inference-time compute includes both generation and evaluation time.

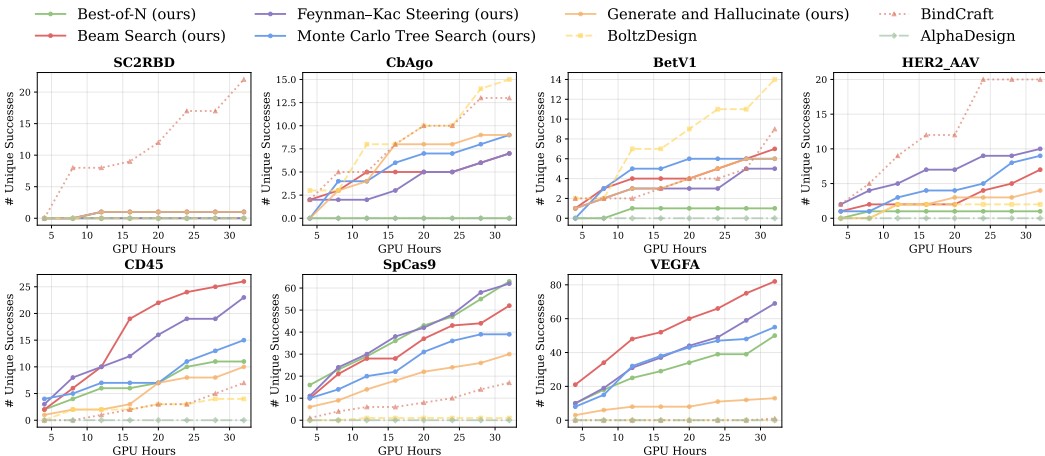

Figure 22: **Unique successes against inference-time GPU hours on 7 hard targets.** Successes are measured with self-generated sequences and inference-time compute includes both generation and evaluation time.

## I.7 EXTENDED RESULTS: INFERENCE-TIME SEARCH FOR ALL LIGAND TARGETS

In addition to the target-averaged inference-scaling results shown in Fig. 9, we plot the per-target results for four ligand targets in Fig. 23. While the SAM target proves challenging in Table 11, we observe significant success for inference-scaling methods best-of-N, beam search, and FK-steering. For the other three targets, best-of-N yields the highest unique success rate, with beam search as a close second. Notably, Monte Carlo tree search performs the worst across all targets. Overall across all targets both Complexa with best-of-N and beam-search significantly outperform BoltzDesign.

We hypothesize that best-of-N performs the strongest due to the reward function used in all Complexa ligand-conditional inference-scaling experiments, which is the 0-1 normalized RF3 min ipAE, and the

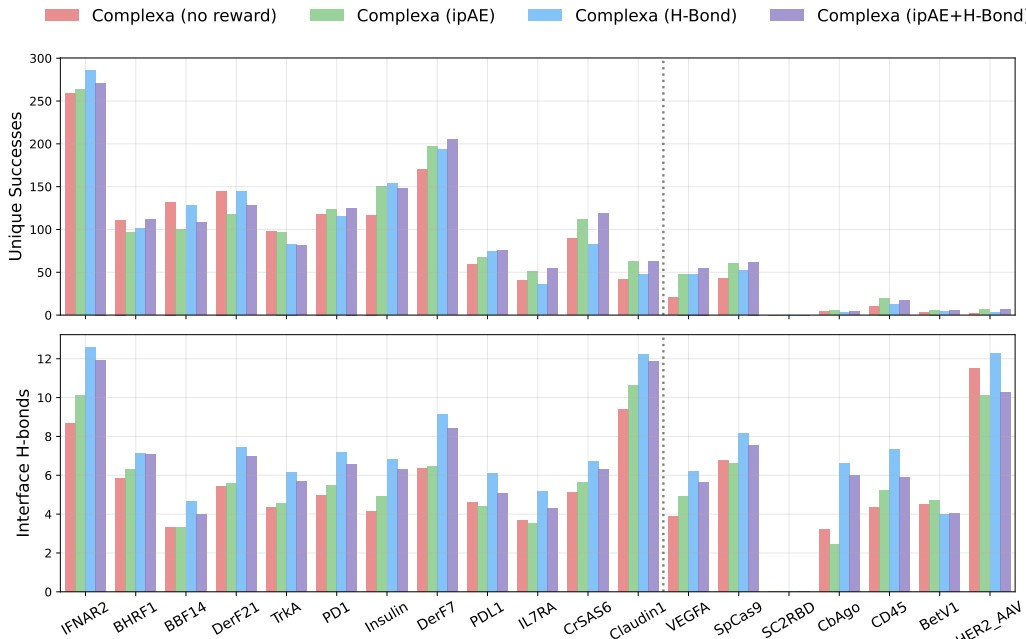

Figure 23: **Unique successes against inference-time GPU hours on 4 small molecule targets.** Successes are measured with self-generated sequences and inference-time compute includes both generation and evaluation time.

Figure 24: **Unique successes and number of interface hydrogen bonds on 19 targets using Beam Search with different combinations of folding and hydrogen bond rewards.** Successes are measured with self-generated sequences and hydrogen bond counts are computed on refolded structures over successful designs.

fact that, despite passing min ipAE and binder backbone scRMSD, the majority of samples fail due to having a ligand scRMSD $\geq 5$. As a result, our reward function, while a component of the success criteria, does not fully capture the primary challenge of protein-ligand co-design: maintaining binding site specificity. Generating protein binders with good pLDDT and PAE is relatively straightforward, as seen in Cho et al. (2025). However, imposing a stronger interface criterion via min ipAE and enforcing ligand binding site conservation during refolding significantly filters out samples. We leave extending rewards to incorporate both confidence and structural metrics as a direction for future work.

## I.8 EXTENDED RESULTS: INTERFACE HYDROGEN BOND OPTIMIZATION FOR ALL TARGETS

In addition to the standard folding reward-guided inference-time optimization, we further investigate interface hydrogen bond (H-Bond) optimization, as introduced in Sec. 4.2. Alongside the averaged results reported in Tab. 3, we provide per-target visualizations of both the number of unique successes and the number of interface hydrogen bonds in Fig. 24.

For this experiment, we run beam search with a fixed initial budget of 200 samples per target, using the standard setup of $N = 4$, $L = 4$, and $K = 100$ as described in Sec. H.2. This ensures that each target produces the same total number of generated samples, preventing any single target from dominating the hydrogen bond metric. However, because of differences in sequence lengths, this setup naturally results in varying inference-time compute across targets.

To establish a fair comparison, we also include a zero-reward baseline, which is conceptually equivalent to Best-of-N but follows the same sampling protocol to maintain consistency across methods. We then evaluate beam search with four reward configurations: (1) ipAE-only, (2) H-Bond-only, (3) a combined reward of ipAE and H-Bond with equal weights, and (4) the zero-reward baseline. Here, the H-Bond reward is defined as the number of interface hydrogen bonds.

As shown in Fig. 24, optimizing solely for the H-Bond reward yields the highest number of interface hydrogen bonds. Moreover, combining the H-Bond reward with the ipAE reward consistently increases the number of hydrogen bonds compared to ipAE-only optimization. Interestingly, this combined objective sometimes also improves the number of unique successes, suggesting that optimizing for hydrogen bonds can indirectly encourage successful binder generation and increase diversity. Across most targets, all three reward-guided strategies outperform the zero-reward baseline, showing the effectiveness of incorporating hydrogen bond information into inference-time optimization.

Overall, these results highlight the potential of interface hydrogen bond optimization as a complementary objective, improving both interface quality and design success rates.

## I.9  EXTENDED RESULTS: ENZYME DESIGN

**Atomic Motif Enzyme Design.** Building on the strong performance of Complexa in small molecule-conditioned tasks, we extend its capabilities to the more challenging domain of atomic motif enzyme (AME) design. Following Ahern et al. (2025), we assess the efficacy of our model on a diverse set of 41 tasks, which target the complexity and variability of the theozyme (active site) scaffolding problem. Specifically, we generate a protein binder conditioned on the ligand target(s), protein functional groups, and relevant reaction cofactors. The goal is to generate unique successful binders that preserve the enzymatic geometry of the theozyme.

**Modeling Details.** Following Sec. G.2.3, we extend the conditioning in the sequence dimension to accommodate the catalytic residue fragments, similar to our approach for the target. This results in a total sequence length of $L_{binder} + L_{target} + L_{catalytic}$. In contrast to the target featurization, we only require the Atom37 coordinates of the catalytic residue fragments, along with their corresponding amino acid types. The pair representation remains unchanged. By extending the sequence conditioning, Complexa is able to jointly infer the optimal positions in both sequence and 3D space to place and complete these fragments, while preserving co-designability and overall binder success.

**Training Details.** Similar to the base Complexa, we follow a 3-stage training protocol. In the first stage, we reuse the original atomistic autoencoder used to train the base protein-conditioned model, which has a maximum protein length of 256. We then pretrain the flow model on the AFDB cluster representatives, randomly selecting 1-8 pseudo-catalytic residues to condition on for 400k steps on 96 NVIDIA A100-80GB GPUs. Next, we fine-tune on an even ratio of AFDB and PLINDER with 50% target dropout for 100k steps on 48 NVIDIA A100-80GB GPUs. During this final fine-tuning stage, when a ligand is present, we restrict the choice of catalytic residues to those with $C_\alpha$ atoms within 10Å of any ligand atom.

To accurately scaffold a diverse set of residue fragments, we randomly sample one of the possible functional groups for each residue in the set defined by the AME benchmark. While we condition on raw Atom37 representations, we found that restricting to a specific diverse subset of atom configurations works better in practice than randomly sampling a subset of the residue's atoms. The specific configurations used are provided below:

```
PHE: {C, O}
     {CB, CD1, CD2, CE1, CE2, CG, CZ}
ALA: {C, CA, CB, N}
     {CA, N}
     {CA, CB}
     {C, CA, O}
     {C, O}
HIS: {CD2, CE1, CG, ND1, NE2}
     {CB, CD2, CE1, CG, ND1, NE2}
     {CD2, CE1, NE2}
     {CE1, CG, ND1}
     {CE1, ND1, NE2}
GLY: {CA, N}
     {C, CA, N}
TRP: {CD1, CD2, CE2, CG, CZ2, NE1}
     {CB, CD1, CD2, CE2, CE3, CG, CH2, CZ2, CZ3, NE1}
ASP: {CB, CG, OD1, OD2}
     {CG, OD2}
     {CG, OD1}
     {C, CA, CB, N}
     {C, CA, CB, CG, N, O}
ARG: {CZ, NE, NH1, NH2}
     {C, CA, CB, N}
     {CZ, NH1}
     {CD, CZ, NE}
     {CD, CG, CZ, NE, NH1, NH2}
     {CB, CD, CG, CZ, NE}
     {CZ, NH2}
     {C, O}
     {CD, CZ, NE, NH1, NH2}
LYS: {CD, CE, NZ}
     {CE, NZ}
THR: {CA, CB, CG2, OG1}
     {C, CA, CB, N}
     {CB, OG1}
GLU: {C, CA, O}
     {CA, N}
     {CD, OE2}
     {CD, OE1}
     {CD, CG, OE1, OE2}
ILE: {C, CA, O}
     {C, CA, CB, N}
     {CB, CD1, CG1, CG2}
SER: {CB, OG}
     {C, CA, O}
     {CA, CB, OG}
     {C, O}
     {CA, N}
     {C, CA, CB, N}
ASN: {CB, CG, ND2, OD1}
     {CA, N}
     {CG, ND2}
CYS: {CA, CB, SG}
     {CB, SG}
GLN: {CA, N}
     {CD, OE1}
     {CD, CG, NE2, OE1}
TYR: {CE1, CE2, CZ, OH}
     {CZ, OH}
     {CB, CD1, CD2, CE1, CE2, CG, CZ, OH}
PRO: {CD, CG, N}
     {CA, CB, CD, CG, N}
LEU: {CB, CD1, CD2, CG}
MET: {CE, CG, SD}
VAL: {CB, CG1, CG2}
```

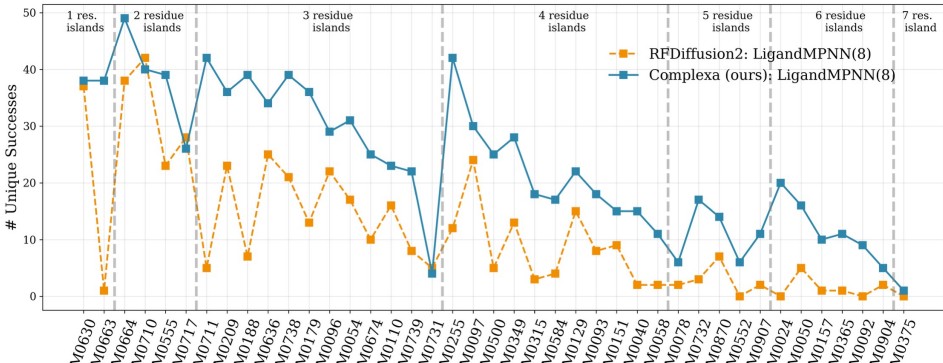

Figure 25: **Enzyme Design Benchmark—Best-of-8 Re-designs.** Complexa significantly outperforms RFDiffusion2 in 38/41 AME enzyme design benchmark tasks. Both methods use best-of-8 re-designed sequences.

**AME Benchmark Results.** Following the success criteria introduced in Ahern et al. (2025), we consider an enzymatic design a success if it meets the following criteria. Note that we use the recent RosettaFold-3 (Corley et al., 2025) (RF3) for co-folding due to its ability to template the input ligand conformation, which is critical for maintaining the geometry of the theozyme that the models are conditioned on. Criteria:

1. The catalytic residue types are recovered.

2. The generated binder backbone has a scRMSD of $\leq 2\text{Å}$ after refolding with RF3.

3. The all-atom catalytic functional groups are recovered with a scRMSD of $\leq 1.5\text{Å}$ after refolding with RF3.

4. There are no clashes with the ligand(s), defined as no binder atom within $1.5\text{Å}$.

In this unindexed task, we infer catalytic residue indices from the generated output using a greedy matching procedure (Chen et al., 2025). Specifically, for each residue in the ground-truth theozyme, we find its closest structural counterpart in the generated protein and compute the RMSD using this matched set of residues. This matching process is performed independently for each generated protein, accounting for potential shifts in motif placement across samples. As in our prior evaluations, we cluster the generated proteins with FoldSeek to report the number of unique successful designs.

We evaluate our model against RFDiffusion2, which generates protein backbones without sequences. To obtain full-atom complexes, RFDiffusion2 relies on LigandMPNN (Dauparas et al., 2025) for sequence generation, followed by refolding with RF3. We compare our model, using both self-generated sequences and single-shot LigandMPNN, to RFDiffusion2 with LigandMPNN. Additionally, we report the best-of-8 LigandMPNN re-designs, as per the original evaluation protocol (Ahern et al., 2025).

Following Ahern et al. (2025), we separate the design tasks by the number of residue islands, i.e. the number of contiguous segments of catalytic residues in the original PDB structure. As shown in Table 12, our model achieves a success rate of 41/41 with self-generated sequences and 40/41 with a single LigandMPNN re-design, outperforming RFDiffusion2's 30/41. Moreover, when considering the best-of-8 LigandMPNN sequence re-designs, our model surpasses RFDiffusion2 on 38/41 tasks, including all tasks with $\geq 4$ residue islands. Notably, in the three tasks where we underperform, our model achieves higher raw success rates, with the difference attributed to only 1-2 unique sample-level losses. We visualize these results in Fig. 10 (single sequence), Fig. 25 (best-of-8), and Fig. 26 (average results by residue island count).

Note that we did not sample RFDiffusion2 ourselves, but were provided the generated RFDiffusion2 samples directly by the RFDiffusion2 authors (Ahern et al., 2025).

Table 12: **Enzyme Design Benchmark—Detailed Quantitative Results.** "All" indicates total number of successes produced by the model (we produce 100 samples per task), while "Unique" indicates number of unique successes, obtained by clustering all successes. The method with the most unique successes for 8 sequence re-designs (LigandMPNN(8)) is highlighted in blue and for single sequence methods is highlighted in red.

| AME Task | # residue islands | RFDiffusion2 LigandMPNN(8) | | RFDiffusion2 LigandMPNN(1) | | Complexa (ours) LigandMPNN(8) | | Complexa (ours) LigandMPNN(1) | | Complexa (ours) Self-Generated | |
|---|---|---|---|---|---|---|---|---|---|---|---|
| | | All | Unique | All | Unique | All | Unique | All | Unique | All | Unique |
| M0630 | 1 | 78 | 37 | 55 | 22 | 88 | **38** | 76 | **32** | 70 | 27 |
| M0663 | 1 | 1 | 1 | 0 | 0 | 62 | **38** | 36 | **26** | 31 | 24 |
| M0710 | 2 | 72 | **42** | 39 | 22 | 91 | 40 | 67 | **34** | 64 | 31 |
| M0717 | 2 | 55 | **28** | 27 | 12 | 65 | 26 | 39 | **16** | 28 | 11 |
| M0664 | 2 | 72 | 38 | 36 | 19 | 88 | **49** | 70 | **47** | 60 | 42 |
| M0555 | 2 | 59 | 23 | 11 | 6 | 80 | **39** | 38 | **20** | 42 | 19 |
| M0636 | 3 | 39 | 25 | 17 | 16 | 52 | **34** | 24 | **18** | 19 | 17 |
| M0674 | 3 | 11 | 10 | 2 | 2 | 42 | **25** | 20 | **13** | 19 | 12 |
| M0739 | 3 | 13 | 8 | 4 | 3 | 52 | **22** | 17 | **7** | 13 | **7** |
| M0096 | 3 | 39 | 22 | 11 | 6 | 49 | **29** | 15 | 10 | 20 | **14** |
| M0738 | 3 | 31 | 21 | 8 | 8 | 59 | **39** | 29 | **19** | 23 | 16 |
| M0209 | 3 | 40 | 23 | 9 | 9 | 92 | **36** | 37 | 18 | 43 | **25** |
| M0054 | 3 | 31 | 17 | 7 | 4 | 49 | **31** | 24 | **16** | 15 | 13 |
| M0188 | 3 | 15 | 7 | 2 | 2 | 49 | **39** | 15 | 14 | 24 | **17** |
| M0711 | 3 | 10 | 5 | 3 | 3 | 57 | **42** | 35 | **29** | 31 | 25 |
| M0179 | 3 | 15 | 13 | 6 | 4 | 45 | **36** | 18 | **15** | 17 | **15** |
| M0110 | 3 | 19 | 16 | 6 | 6 | 35 | **23** | 15 | **12** | 14 | 11 |
| M0731 | 3 | 5 | **5** | 1 | 1 | 6 | 4 | 1 | 1 | 2 | **2** |
| M0097 | 4 | 41 | 24 | 17 | 12 | 34 | **30** | 19 | **18** | 16 | 15 |
| M0349 | 4 | 17 | 13 | 5 | 4 | 30 | **28** | 8 | **8** | 7 | 7 |
| M0129 | 4 | 20 | 15 | 6 | 6 | 26 | **22** | 13 | **12** | 5 | 5 |
| M0255 | 4 | 17 | 12 | 5 | 4 | 58 | **42** | 32 | **24** | 26 | 20 |
| M0500 | 4 | 6 | 5 | 1 | 1 | 34 | **25** | 14 | **14** | 16 | 13 |
| M0151 | 4 | 9 | 9 | 1 | 1 | 25 | **15** | 16 | **11** | 3 | 3 |
| M0058 | 4 | 2 | 2 | 0 | 0 | 12 | **11** | 1 | 1 | 2 | **2** |
| M0584 | 4 | 6 | 4 | 1 | 1 | 32 | **17** | 7 | 3 | 9 | **5** |
| M0040 | 4 | 7 | 2 | 3 | 1 | 32 | **15** | 6 | **4** | 5 | 2 |
| M0093 | 4 | 8 | 8 | 2 | 2 | 20 | **18** | 6 | **6** | 5 | 4 |
| M0315 | 4 | 3 | 3 | 0 | 0 | 18 | **18** | 4 | 4 | 6 | **6** |
| M0907 | 5 | 2 | 2 | 0 | 0 | 11 | **11** | 3 | **3** | 1 | 1 |
| M0870 | 5 | 7 | 7 | 2 | 2 | 34 | **14** | 11 | **5** | 9 | 4 |
| M0732 | 5 | 4 | 3 | 1 | 1 | 32 | **17** | 11 | **8** | 4 | 4 |
| M0552 | 5 | 0 | 0 | 0 | 0 | 9 | **6** | 1 | **1** | 1 | **1** |
| M0078 | 5 | 2 | 2 | 1 | 1 | 6 | **6** | 1 | 1 | 5 | **4** |
| M0904 | 6 | 2 | 2 | 0 | 0 | 12 | **5** | 2 | **2** | 3 | 1 |
| M0157 | 6 | 1 | 1 | 0 | 0 | 16 | **10** | 9 | **7** | 5 | 5 |
| M0050 | 6 | 5 | 5 | 1 | 1 | 23 | **16** | 7 | **5** | 6 | **5** |
| M0365 | 6 | 1 | 1 | 0 | 0 | 21 | **11** | 6 | 2 | 7 | **3** |
| M0024 | 6 | 0 | 0 | 0 | 0 | 22 | **20** | 5 | 5 | 7 | **7** |
| M0092 | 6 | 0 | 0 | 0 | 0 | 10 | **9** | 1 | 1 | 2 | **2** |
| M0375 | 7 | 0 | 0 | 0 | 0 | 3 | **1** | 0 | 0 | 1 | **1** |

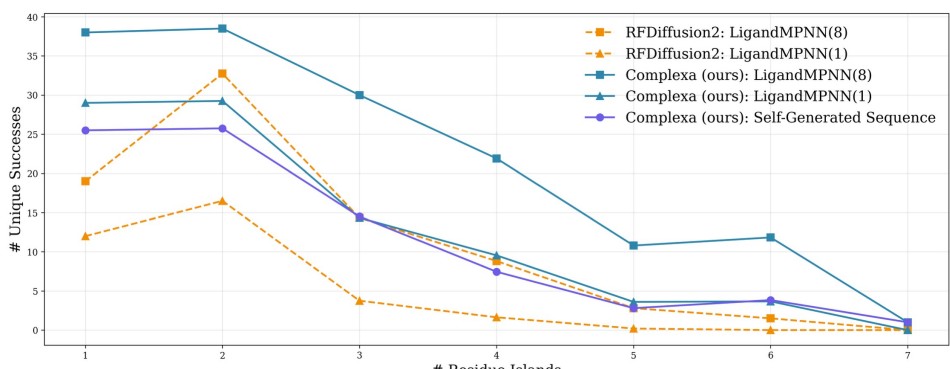

Figure 26: **Enzyme Design Benchmark—Aggregated Results.** Complexa significantly outperforms RFDiffusion2 across all sequence modes.

## I.10 CASE STUDY: LONG INFERENCE-TIME SEARCH AND PLATEAU BEHAVIOR

In addition to the relatively short-term inference-time optimization analysis on easy and hard targets shown in Fig. 7, we further investigate long-term behavior and explore whether it is possible to exhaustively generate and search for valid binders for a hard target. For this computationally expensive case study, we selected a single target, **SpCas9**. In Fig. 27, we analyze the solution space and long-term scaling by examining unique success counts using the standard template modeling score (TM-score) clustering threshold of 0.5, alongside coarser thresholds. We conducted an extensive beam search on the SpCas9 target to generate binders of varying lengths. We use the standard setup

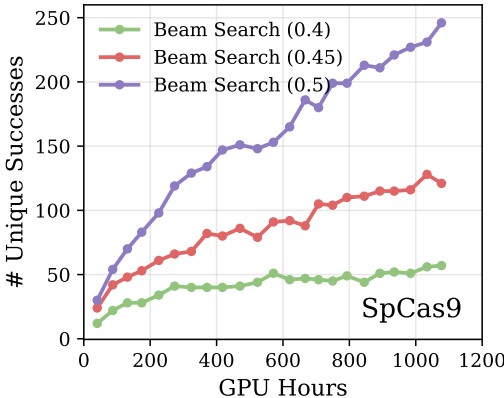

Figure 27: **Extensive Search Case Study.** Number of unique successes on the target **SpCas9** against inference-time GPU hours with different TM-score clustering thresholds used in the unique success calculation (green: 0.4, red: 0.45 and blue: 0.5). Successes are measured with self-generated sequences and inference-time compute includes both generation and evaluation time.

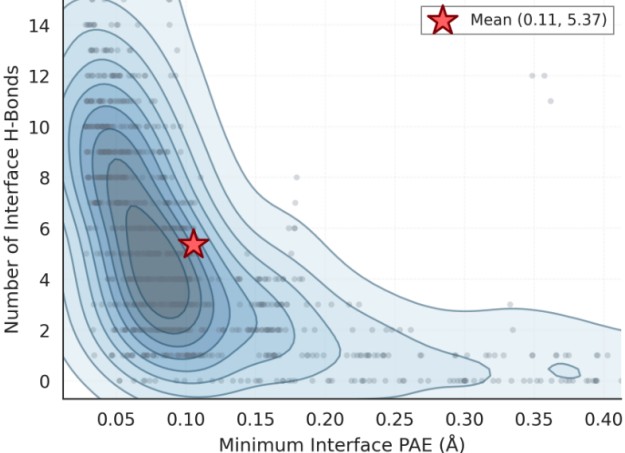

Figure 28: **Interface hydrogen bonding and interface predicted aligned error correlations.** The number of interface hydrogen bonds increases with decreasing ipAE values of ColabDesign during inference-time search with negative ipAE as a reward (Sec. I.11).

of $N = 4$, $L = 4$, and $K = 100$, as detailed in Sec. H.2. Within a computational budget of 1,100 GPU hours for optimization and evaluation, the model produced 13,596 successful binders, resulting in 246 unique successes under the above default unique success clustering criteria.

As illustrated in Fig. 27, the number of unique successes continues to increase up to 1,100 GPU hours when using tighter clustering thresholds, indicating a vast potential solution space. Interestingly, when the threshold is lowered to 0.4 for coarser clustering, a plateau slowly emerges after 600 GPU hours (at which point over 8,000 successful binders had been generated). While the total number of successful binders continues to rise, the discovery of structurally distinct clusters slows. Overall, we find that while Complexa can generate diverse successes, it eventually approaches saturation within the solution space.

## I.11 IPAE SCORES AND HYDROGEN BOND CORRELATIONS

To investigate whether inference-time optimization of one property like ipAE score can lead to adversarial effects such as degradation in biophysical metrics like hydrogen bonds, we investigated both ipAE scores and the number of hydrogen bonds in Complexa samples that were generated by optimisation for low ipAE only (Fig. 28). The results show that no adversarial optimization seems to

Table 13: **Binding affinity comparison of unique small molecule binders.** We report the average and best $pK_d$ in kcal/mol predicted by FLOWR-ROOT. ‡ denotes redesign of the binder sequence using LigandMPNN. † denotes co-folding with RosettaFold3, otherwise the model-generated structure is used. RFDiffusionAA requires both LigandMPNN redesign and co-folding as it generates a backbone without sequence.

| Target | Model | $pK_d$ ↑ | | # of Samples |
|---|---|---|---|---|
| | | Mean$_{\pm Std}$ | Best | |
| SAM | RFDiffusionAA ‡† | **5.65**$_{\pm 0.15}$ | **5.76** | 2 |
| | Complexa *(ours)* | 4.81$_{\pm 0.49}$ | 5.49 | 10 |
| | Complexa *(ours)* † | 4.94$_{\pm 0.54}$ | 5.74 | 10 |
| OQO | RFDiffusionAA ‡† | 7.01$_{\pm 0.31}$ | 7.35 | 3 |
| | Complexa *(ours)* | 7.19$_{\pm 0.14}$ | 7.37 | 6 |
| | Complexa *(ours)* † | **7.51**$_{\pm 0.28}$ | **8.05** | 6 |
| FAD | RFDiffusionAA ‡† | 6.49$_{\pm 0.76}$ | 7.39 | 5 |
| | Complexa *(ours)* | 6.52$_{\pm 0.59}$ | 7.39 | 17 |
| | Complexa *(ours)* † | **6.66**$_{\pm 0.59}$ | **7.65** | 17 |
| IAI | RFDiffusionAA ‡† | 5.67$_{\pm 0.40}$ | 6.47 | 8 |
| | Complexa *(ours)* | 5.58$_{\pm 0.49}$ | 6.71 | 19 |
| | Complexa *(ours)* † | **5.77**$_{\pm 1.05}$ | **8.25** | 19 |

take place, but rather the opposite: reducing the ipAE scores leads to an increase in hydrogen bond interactions between binder and target, with a Spearman correlation of -0.69.

## I.12 BINDING AFFINITIES FOR SMALL MOLECULE TARGETS

To further assess the physical validity of the ligand-conditioned protein binders generated by Complexa, we evaluate the predicted binding affinity of the unique protein-ligand complexes reported in Table 1. We utilize FLOWR.ROOT (Cremer et al., 2025), a recent machine learning-based affinity predictor trained on one of the largest combinations of datasets to date, offering state-of-the-art accuracy while being orders of magnitude faster than Boltz-2 (Passaro et al., 2025). Additionally, FLOWR.ROOT can predict affinity for given structures without requiring re-co-folding.

In Tab. 13, we compare our results with RFDiffusion-AllAtom, using the same generated samples that also underlie the results presented in the table in the main text. We outperform RFDiffusion-AllAtom on 3 out of 4 targets, generating the highest affinity binders on average. Notably, for SAM, RFDiffusion-AllAtom only generates 2 successful binders, and our best binder matches their best, demonstrating that we overall generate physically plausible binders. It worth noting that, as shown in Tab. 1, Complexa outperforms RFDiffusion-AllAtom by a large margin in terms of ipAE-based success metrics and does so while maintaining a much higher sampling efficiency.

## I.13 BIOPHYSICAL INTERFACE ANALYSES FOR PROTEIN TARGETS

To enrich the analysis of the generated Complexa binders beyond folding score metrics, we conducted an analysis of the biophysical properties of the generated binder-target interfaces and compared them to the interfaces in the PDB multimers we used for training, specifically the same subset as depicted in Fig. 12 for the Teddymer analysis (this is, we used the same 2,000 PDB multimer samples as reference and analyze the same metrics). Please see Fig. 29 for the results.

As one can see in these metrics, for most properties the generated binders follow the same trend as the PDB reference set, with the general trend being that the generated interfaces and binders are slightly less hydrophobic and smaller, which is also reflected in the reduced interface shape complementarity. Overall, this indicates that our model generates realistic target-binder interfaces.

## J  BASELINE EVALUATIONS

**BindCraft** We used the code and checkpoints provided in the public BindCraft repository. For binder generation, we directly used the `default_4stage_multimer_hardtarget` configuration for the advanced setting and the default filter provided in the repository. Hotspot conditioning was set according to Table 4. The original full pipeline of BindCraft included re-designing the hallucinated

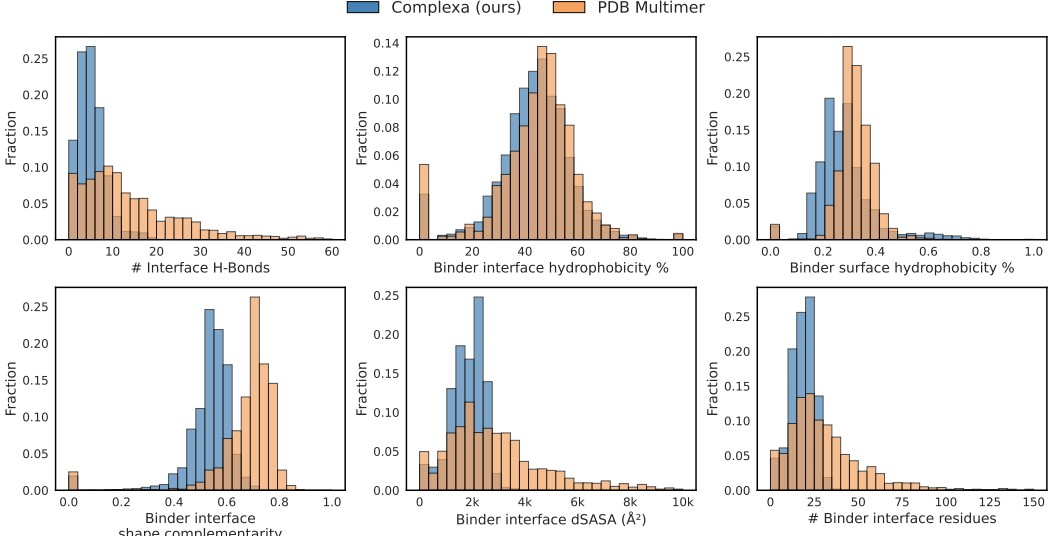

Figure 29: **Generated binder interfaces vs. PDB multimer inferfaces.** Comparison of bioinformatic metrics between the PDB Multimer reference distribution and successful binders generated by Complexa.

sequence with ProteinMPNN 20 times and filtering for the best. In our evaluation pipeline, we had a similar workflow of re-designing the sequence 8 times and filtered for the best. Therefore, we skipped the BindCraft built-in sequence re-designing stage after structure relaxation, so that the evaluation of BindCraft samples was consistent with our model and other baselines. The reported sampling time of BindCraft did not include the sequence re-designing stage.

**RFdiffusion** We used the code and checkpoint provided in the public RFdiffusion repository. We used the default setting in the repository for target-conditioned binder generation. Hotspot conditioning was turned on. We sampled with `noise_scale_ca` and `noise_scale_frame` set to both 0 and 1 and report the results for noise 0 since these consistently outperformed the noise 1 setting.

**ProtPardelle-1c** We used the code and checkpoints provided on the public Protpardelle-1c repository. Specifically, we used the `cc83_epoch2616` checkpoint for target-conditioned binder generation. The sampling parameters were default: `step_scale=1.2`, `schurns=200`, `crop_cond_starts=0.0`, and `translations=[0.0, 0.0, 0.0]`. Hotspot conditioning was set according to Table 4.

**APM** We used the code and checkpoints provided on the public APM repository. We set both the `direction_condition` and `direction_surface` to `null`, following the instruction from authors for binder generation.

**BoltzDesign1** We used the code and checkpoints provided in the public BoltzDesign1 repository. For protein targets, we set the flag `use_msa=true` while generating binders. We turned off the built-in LigandMPNN (Dauparas et al., 2025) sequence re-design and AlphaFold (Abramson et al., 2024) and rather used the identical fixed interface LigandMPNN re-design in our evaluation pipeline. We turned off the Rosetta (Alford et al., 2017) energy scoring because it was redundant in our evaluation.

**AlphaDesign** We used the AlphaDesign code released publicly on Zenodo. We used the default setting provided for the target-conditioned binder generation. We set the parameter `design_max_iter=200` as instructed. We skipped the AlphaDesign built-in sequence re-designing model and used the best-of-8 ProteinMPNN strategy in our evaluation pipeline to be consistent with other baselines. For the sampling of all baselines and our Complexa model, we set a computation budget of 4 NVIDIA A100 hours for each sample. AlphaDesign could exceed the budget for large targets. For example, only 1 of the 200 samples of target VEGFA finished within 4 hours. Those unfinished samples were unfortunately considered as fail cases during the benchmark of # unique successes with certain GPU hours budget (e.g. Table 7).

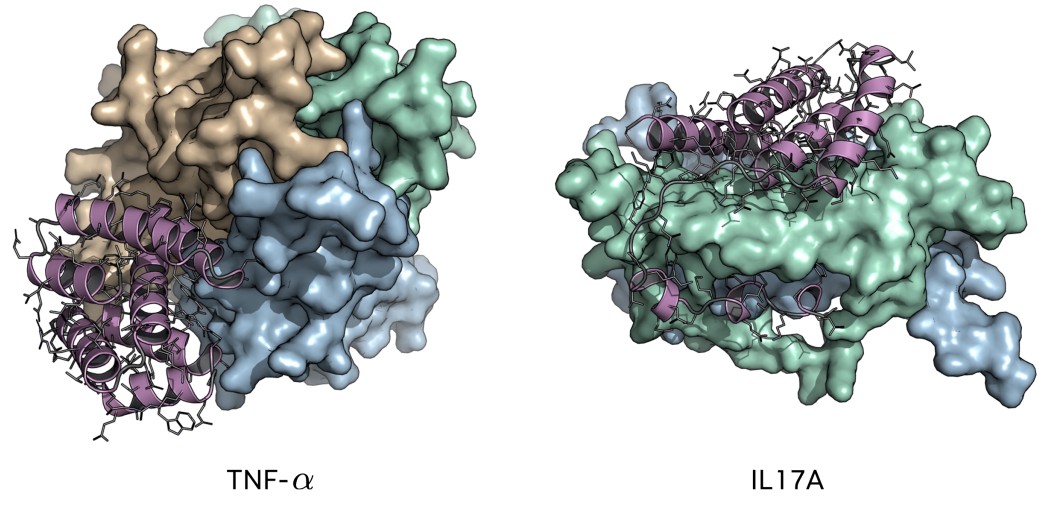

TNF-$\alpha$         IL17A

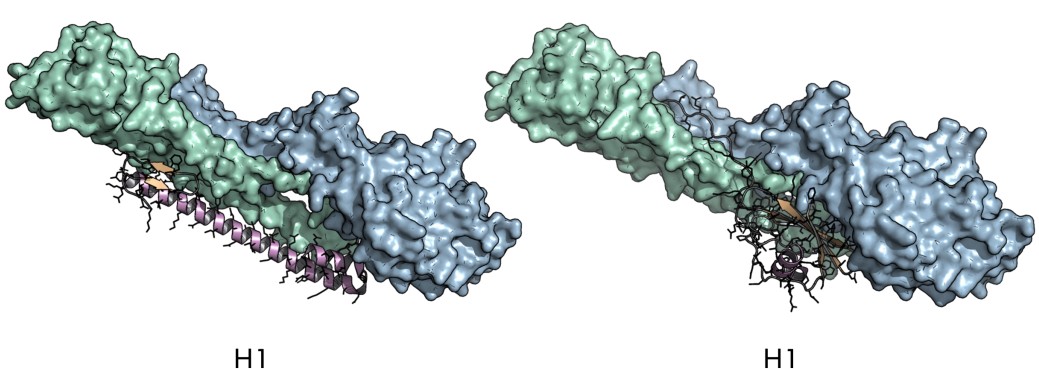

H1         H1

Figure 30: Binders generated by Complexa for the challenging large multi-chain targets TNF-$\alpha$ (three-chain target), IL17A (two-chain target), and H1 (two-chain target; two different binders visualized). All shown binders meet the in-silico success criteria. Binders are visualized in cartoon representation, including side chains, multi-chain targets in surface representation. See Sec. I.4.

## K  ADDITIONAL VISUALIZATIONS

In Fig. 30, we show successful de novo binders generated by Complexa for the challenging multi-chain targets TNF-$\alpha$ (three-chain target), IL17A (two-chain target), and H1 (two-chain target).

In Fig. 31, we show successful small molecule binders generated by Complexa for the four small molecules SAM, IAI, FAD and OQO.

In Fig. 32, we show fold class-conditioned binder generation with Complexa for five different targets, controlling secondary structure content.

## L  DECLARATION ON USAGE OF LARGE LANGUAGE MODELS

Large Language Models were used during the preparation of the manuscript exclusively to catch typographical and grammatical errors and to improve writing style. Beyond that, no LLMs were involved in the research or the project in any way.

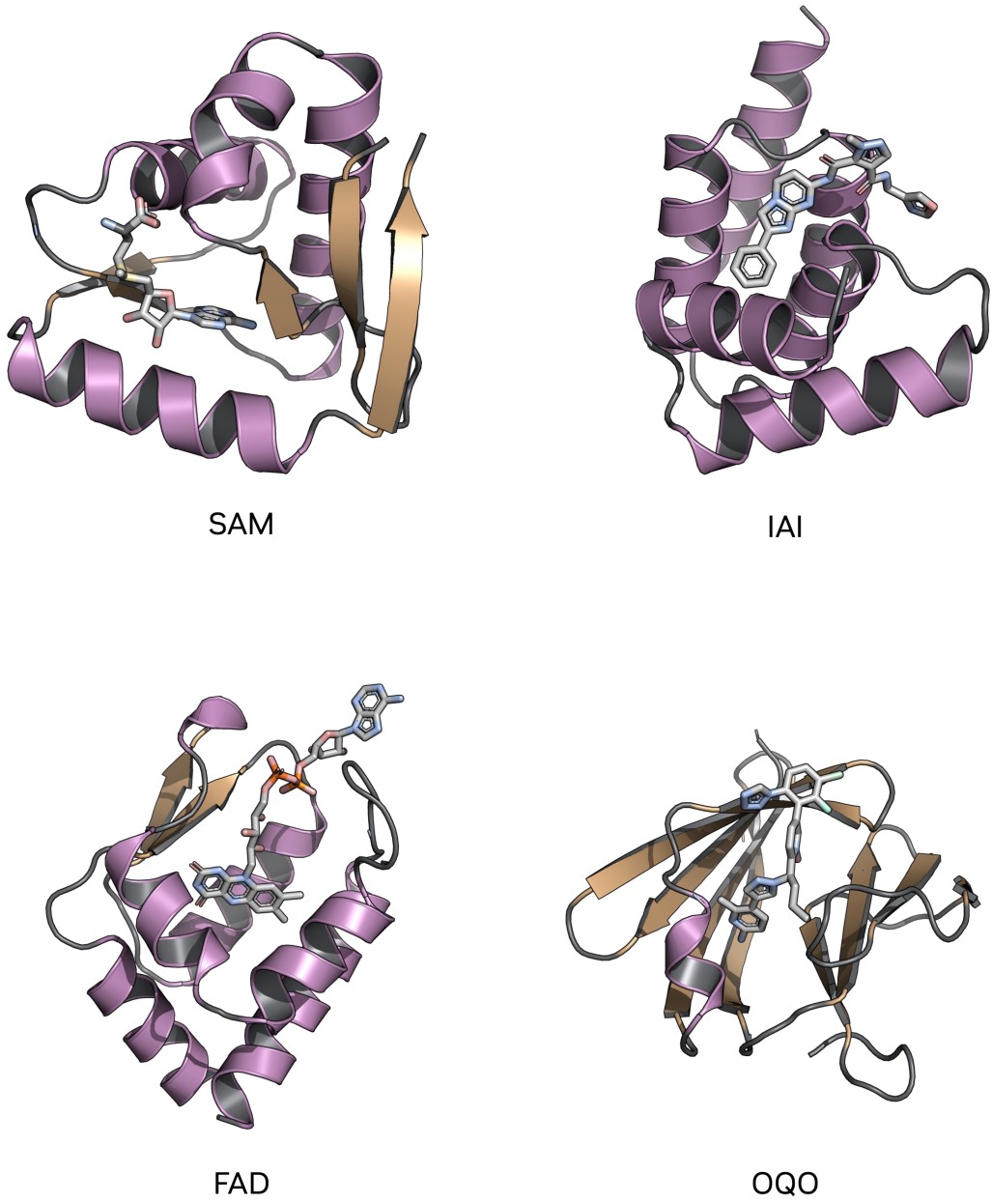

Figure 31: Binders generated by Complexa for small molecule targets SAM, IAI, FAD and OQO (see Sec. E). All shown binders meet the in-silico success criteria.

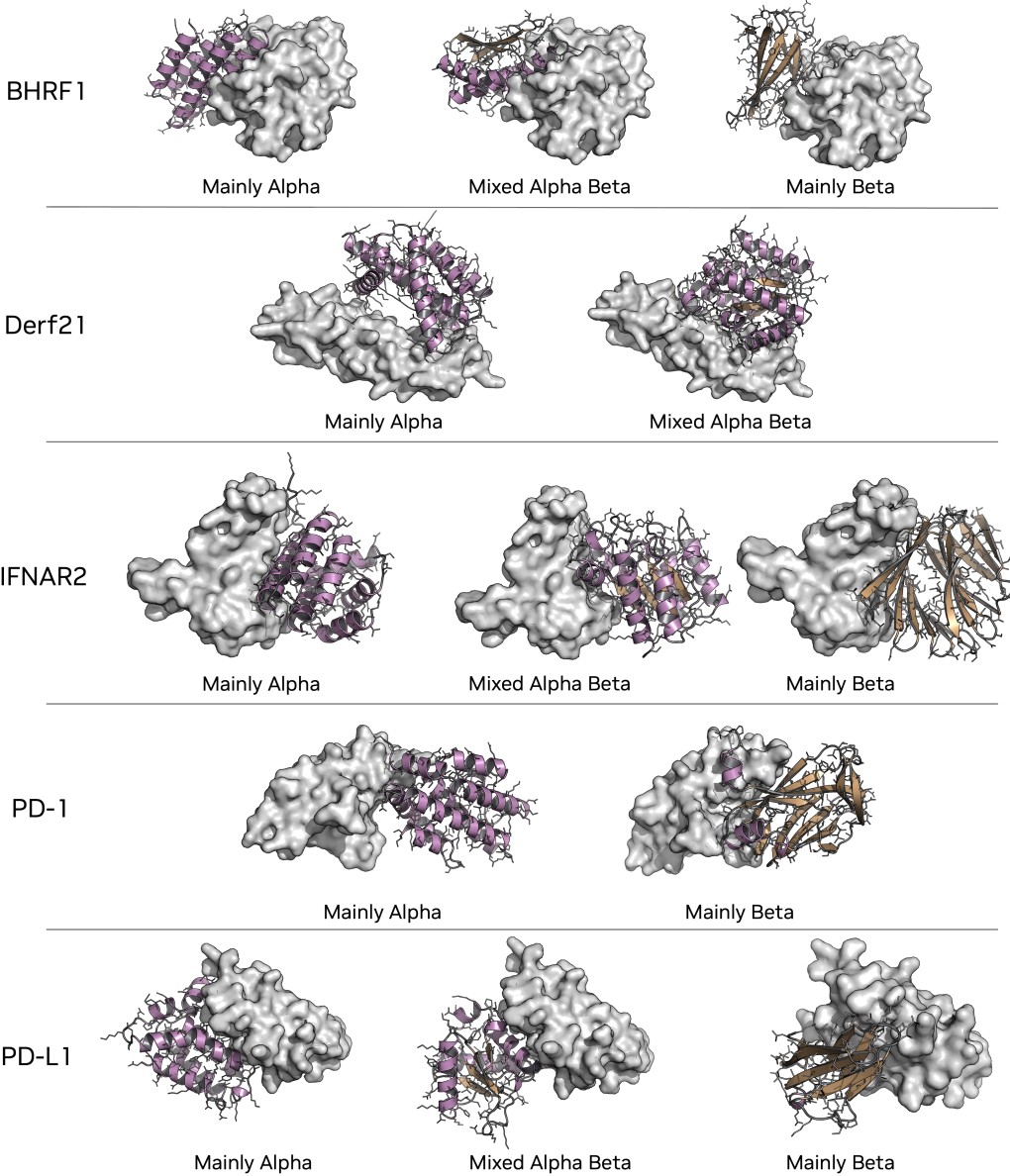

Figure 32: Fold class-conditioned binders generated by Complexa for five different targets. We control the secondary structure content through conditioning on C-level CAT labels "Mainly Alpha", "Mainly Beta" or "Mixed Alpha Beta" (Dawson et al., 2016; Geffner et al., 2025). We can observe that the generated binders follow the conditioning, exhibiting primarily alpha helices, beta sheets, or both. All shown binders meet the in-silico success criteria. Binders are visualized in cartoon representation, including side chains, targets in surface representation.

Table 14: Hyperparameters for Complexa model training. We denote two versions of Complexa that specialized in generating binders for protein targets and small molecule ligand target as **Protein-Protein** and **Ligand-Protein**, respectively. **Protein-Protein CAT** refers to the model that can generate binders to protein target conditioned on CAT labels.

| Hyperparameters | Complexa | | |
|---|---|---|---|
| | Protein-Protein | Protein-Protein CAT | Ligand-Protein |
| **Model Architecture** | | | |
| **Partially Latent Flow Matching** | | | |
| sequence repr dim | 768 | 768 | 768 |
| sequence cond dim | 256 | 256 | 256 |
| $t$ sinusoidal enc dim | 256 | 256 | 256 |
| idx. sinusoidal enc dim | 256 | 256 | 256 |
| fold class cond dim | 0 | 256 | 0 |
| pair repr dim | 256 | 256 | 256 |
| seq separation dim | 128 | 128 | 128 |
| pair distances dim ($\mathbf{x}_t$) | 30 | 30 | 30 |
| pair distances dim ($\hat{\mathbf{x}}(\mathbf{x}_t)$) | 30 | 30 | 30 |
| pair distances min (Å) | 1 | 1 | 1 |
| pair distances max (Å) | 30 | 30 | 30 |
| # attention heads | 12 | 12 | 12 |
| # transformer layers | 14 | 14 | 14 |
| # trainable parameters | 159 M | 159 M | 159 M |
| **VAE** | | | |
| enc/dec sequence repr dim | 768 | 768 | 768 |
| enc/dec # attention heads | 12 | 12 | 12 |
| enc/dec # transformer layers | 12 | 12 | 12 |
| enc/dec sequence cond dim | 128 | 128 | 128 |
| enc/dec idx. sinusoidal enc dim | 128 | 128 | 128 |
| enc/dec pair repr dim | 256 | 256 | 256 |
| latent dimension | 8 | 8 | 8 |
| # trainable parameters | 256 M | 256 M | 256 M |
| **Training Details** | | | |
| **VAE AFDB training** | | | |
| Dataset | AFDB monomer | AFDB monomer | AFDB monomer |
| max sequence length | 256 | 256 | 512 |
| # train steps | 500k | 500k | 140K |
| train batch size per GPU | 14 | 14 | 5 |
| learning rate | 1e-4 | 1e-4 | 1e-4 |
| optimizer | Adam | Adam | Adam |
| # GPUs | 16 | 16 | 32 |
| **VAE PDB fine-tuning** | | | |
| Dataset | PDB | PDB | N/A |
| # train steps | 40k | 40k | N/A |
| train batch size per GPU | 12 | 12 | N/A |
| learning rate | 1e-4 | 1e-4 | N/A |
| optimizer | Adam | Adam | N/A |
| # GPUs | 16 | 16 | N/A |
| **Partially Latent pre-training** | | | |
| Dataset | AFDB monomer | AFDB monomer | AFDB monomer |
| max sequence length | 256 | 256 | 512 |
| # train steps | 540K | 540K | 270K |
| train batch size per GPU | 12 | 12 | 5 |
| learning rate | 1e-4 | 1e-4 | 1e-4 |
| optimizer | Adam | Adam | Adam |
| # GPUs | 32 | 32 | 48 |
| **Partially Latent fine-tuning** | | | |
| Dataset | Teddymer+PDB | Teddymer | PLINDER+AFDB monomer |
| # train steps | 290K | 200K | 60K |
| target dropout | 0% | 0% | 50% |
| use LoRA | False | False | True |
| LoRA rank | N/A | N/A | 32 |
| LoRA $\alpha$ | N/A | N/A | 64 |
| train batch size per GPU | 6 | 6 | 5 |
| learning rate | 1e-4 | 1e-4 | 1e-4 |
| optimizer | Adam | Adam | Adam |
| # GPUs | 96 | 96 | 96 |

