# OpenReview forum: "Scaling Atomistic Protein Binder Design with Generative Pretraining and Test-Time Compute"
_ICLR.cc/2026/Conference — ICLR 2026 Oral_

### Official Review · Reviewer_7jnG · 2025-10-30

**Soundness:** 3
**Presentation:** 4
**Contribution:** 3
**Rating:** 6
**Confidence:** 3

**Summary:**

This paper introduces Complexa, a novel framework for fully atomistic protein binder design that aims to unify two dominant but previously separate paradigms: generative modeling (like RFDiffusion) and sequence "hallucination" (like BindCraft). The authors argue that this dichotomy is false and that performance can be maximized by combining a strong generative prior with scalable inference-time optimization.

To train a strong base model, the paper makes two key contributions. First, it introduces Teddymer, a new, large-scale dataset of synthetic binder-target pairs, cleverly constructed by splitting predicted multi-domain monomer structures from the AlphaFold Database into interacting domain pairs. Second, it extends a state-of-the-art flow-matching generative model (La-Proteína) to be conditional on a target's structure, pretraining it on monomers and the new Teddymer dataset.

The framework's main novelty is in its use of this trained generative model as a prior for inference-time optimization. The authors adapt multiple test-time search algorithms (e.g., Best-of-N, Beam Search, MCTS) to steer the generative process toward high-quality binders, using scores from external structure predictors (like AlphaFold2) as rewards.

Complexa achieves new state-of-the-art results on computational binder design benchmarks. Its base model outperforms prior generative methods, and its inference-time optimization significantly outperforms prior hallucination methods under normalized compute budgets. The authors also demonstrate the framework's flexibility by extending it to small molecule targets, optimizing for interface hydrogen bonds, and guiding generation by protein fold class.

**Strengths:**

1. Solves a Key Data Bottleneck: The Teddymer dataset is a very clever and practical solution to the lack of large-scale binder-target complex data. The ablation study proves it is essential to the model's success.

2. Extremely Strong Empirical Results: The paper demonstrates clear and significant superiority over existing SOTA methods (RFDiffusion, BindCraft, BoltzDesign) in fair, compute-matched comparisons.

3. Thoroughness and Versatility: The authors are not satisfied with just one result. They show the method works for both protein and small-molecule targets, can optimize for physics-based properties (H-bonds), and can be controlled (fold-class conditioning). This demonstrates the framework's power and flexibility.

**Weaknesses:**

1. System Complexity: This is an "everything but the kitchen sink" model. It involves a VAE, a flow-matching model, and complex test-time search algorithms that in turn rely on other large models (AF2, RF3) as reward oracles. This makes the system as a whole extraordinarily complex and computationally expensive to train and run, even if it is more efficient than baselines.

2. Reliance on External Oracles: The performance of the test-time optimization is fundamentally coupled to the quality of the external structure predictors (AF2/RF3) used as reward functions. If the oracle is wrong or has blind spots, the search will be steered to exploit those flaws, not necessarily to find biologically viable binders.

3. Teddymer Interface Quality: The central assumption is that the interfaces in Teddymer (from predicted monomer domains) are good proxies for real binder-target interfaces. While Fig. 3 is a nice qualitative example, the paper would be stronger with a quantitative analysis comparing the geometric and biophysical properties of Teddymer interfaces versus a ground-truth set from the PDB.

**Questions:**

1. The ablation study for "Generate & Hallucinate" (G&H) in Fig. 12 shows that this simpler approach is less effective than the fully integrated search methods (like Beam Search). Why do the authors think this is? Is it because the search methods can correct the generative trajectory early and often, whereas G&H can only fix a single complete, and potentially flawed, sample at the end?

2. The "translation noise" ablation (Table 7) shows it is critical for performance. The hypothesis is that this forces the model to learn global positioning. Is it also possible that without this noise, the model simply overfits to the "centered target" convention used in training, and the noise acts as a data augmentation/regularization? Could the authors comment on this alternative interpretation?

3. For the "Generate & Hallucinate" (G&H) experiments (Sec I.3), the paper ablates using BindCraft stages 2+3+4 vs. stage 4 only. Why does the simpler "stage 4 only" (discrete optimization) perform better on easy targets? One might expect the gradient-based logit optimization (stages 2+3) to be more powerful.

---

> ### Author Response · Authors · 2025-11-22
> **Reply by Authors (1)**
>
> We would like to thank the reviewer for their thorough review. We appreciate that they highlighted our novel Teddymer dataset, our extremely strong empirical results, as well as Complexa’s power and flexibility.
>
> For an overview over all added results and modifications to the updated paper, please see the message to all reviewers. Please note that we added many new experiments and analyses to the paper, which we believe further improves our contributions. The main changes in the updated manuscript are highlighted in orange color for your convenience.
>
> In the following, we will address the reviewer’s comments and questions:
>
> **System Complexity:** Indeed, Complexa relies on an autoencoder (VAE) with a partially latent flow matching model as well as test-time compute scaling algorithms. We would like to note that we use the same autoencoder in all experiments for both protein and small molecule targets, thereby simplifying the framework. Only the latent model is conditioned on the target. Overall, the framework can be seen as similar to modern generative models in the vision literature (image and video generation), which typically rely on an autoencoder with a latent diffusion model, and which can benefit from analogous inference-time scaling algorithms. Hence, we argue that Complexa’s overall framework is not more complex than modern generation systems in other domains.
>
> Moreover, as also mentioned by the reviewer, computationally our framework remains highly efficient. As shown in Tables 1 and 2, inference is very fast, which also translates to highly efficient inference-time search, outperforming all baselines (Figures 7, 8, 9). To make this more concrete, consider the case of generating a 100-residue binder to a 300-residue target on an NVIDIA A100-80GB GPU. In this setting, Complexa only takes around 5 seconds, whereas RFDiffusion takes 137 seconds, demonstrating the efficiency and scalability of our method and the fact that Complexa is not expensive but in fact very cheap to run.
>
> We commented on these aspects in the updated manuscript in a newly added paragraph at the end of Section 3.2.
>
> **Reliance on External Oracles:** Indeed, as correctly pointed out by the reviewer we are relying on the confidence scores by structure prediction models to guide our search (specifically ipAE). We would like to point out that this follows standard practice of the field [1,2,3] and that such folding model metrics have been shown to have strong correlation with binding success in the wet lab [4]. The reviewer is correct that these models can have blind spots. Nonetheless, the consensus of the literature is that such folding model scores currently represent the best computational metrics available for binder design, despite their limitations. Developing new in-silico predictive models and rewards for binder design would be a relevant avenue for future work, but has not been the focus in our work.
>
> **Teddymer Interface Quality:** We appreciate the reviewer’s suggestion to more thoroughly analyze the interfaces that are created in the Teddymer dataset. We followed this advice and now include a geometric and biophysical analysis of the Teddymer interfaces, and we quantitatively compare to experimental binder interfaces from the protein databank (PDB). The results are shown in Appendix C.1 in the updated manuscript. We measured the number of hydrogen bonds across the interface, hydrophobicity of the binder interface, hydrophobicity of the binder surface, shape complementarity of the binder interface, delta Solvent Accessible Surface Area (dSASA) of the binder interface, and the number of binder residues on the interface. Some of these biophysical metrics have also been successfully used in prior works to select binder candidates [1,3], which demonstrates their relevance for protein binder interface analyses. We find that the distributions of these metrics for Teddymer interfaces and for PDB interfaces significantly overlap. This means that Teddymer interfaces are indeed good proxies for real binder interactions, which is also validated by the significant performance boost that Complexa experiences after being trained on Teddymer. Please see extended discussion in C.1. We hope that this addresses the reviewer’s concern.
>
> *(continued below)*

---

> > ### Author Response · Authors · 2025-11-22
> > **Reply by Authors (2)**
> >
> > **Question 1: Inefficiency of Generate & Hallucinate:** There are two reasons for this. (1) As hypothesized by the reviewer, G&H indeed can only modify a potentially flawed sample at the end, whereas methods like Beam Search already steer the generation trajectory early on. (2) Moreover, standard hallucination is intrinsically slow: The early stages of hallucination rely on gradients that need to be backpropagated through the structure prediction model (see details in Appendix H.5). If one starts from a good generated candidate already (e.g. for G&H on easy targets), these first stages are not necessary and only slow down overall optimization (Fig. 12, left). The last stage of hallucination does not require backpropagation anymore, and instead only relies on random, unguided mutations, but also that can be less effective than the efficient search effectively guided by Complexa’s pre-trained generative prior in algorithms like Beam Search. Note that this is exactly a key point of our paper: Directly integrating generative pre-training with inference-time optimization for binder design enables enhanced and more scalable binder design compared to previous hallucination methods.
> >
> > **Question 2: Translation Noise:** We prefer the principled interpretation of our translation noise as described in 3.2: Global translations correspond to the lowest frequency of the data to be generated (i.e. the binder), and to enable enhanced refinement over the global positioning we add additional low frequency noise (i.e. the global translation noise) along the entire noising process of the flow matching objective. The model then learns to denoise and correct this noise throughout the entire generation process, which corresponds to continued refinement of the global positioning throughout generation. In standard flow matching, without this, the lowest frequency mode, i.e. the global positioning of the generated data, would be generated at the beginning of the process without further refinement – we avoid this.
> >
> > However, as pointed out by the reviewer, one can potentially also look at this differently: Without the translation noise, the model will essentially determine the global positioning of the generated binder in its first generation step, based on the initial random noise and the given target, without significant refinement. This can also be interpreted as a form of “overfitting”. Adding additional global noise during training as part of the noising schedule can then be seen as a form of regularization to address this, preventing the model from predicting the global position in the first step and forcing it to reason and refine throughout generation. We would not call this data augmentation in the strict sense, though, as we essentially augment the noise distribution used in flow matching, but not the clean data itself.
> >
> > We would also like to refer the reviewer to the concurrent work by Ahern et al. [5], who encounter a similar challenge in the context of motif scaffolding. Their solution, “stochastic centering”, which perturbs the data itself instead of changing the noising process, is different from our translation noise, though.
> >
> > *(continued below)*

---

> > > ### Author Response · Authors · 2025-11-22
> > > **Reply by Authors (3)**
> > >
> > > **Question 3: Optimization Stages in Generate & Hallucinate and Easy Targets:** This question was essentially already answered in our reply to Question 1: On easy targets, the initial binder candidate generated by the Complexa prior usually already represents a strong candidate. Therefore, the coarse and computationally slow (due to the folding model gradients) refinement of the early stages of BindCraft-like hallucination is not necessary anymore and does not improve the sequence. The compute spent is essentially wasted, leading to overall less efficient optimization compared to spending more compute on stage 4 updates. The approximations in the early stages (feeding continuous logits to the folding model; also, logit optimization corresponds to an approximate discrete sequence relaxation) can help steering an initial, random input towards reasonable sequence logits relatively quickly, as in standard BindCraft. But the situation is different when starting from a good candidate already, as in our “Generate & Hallucinate” applied to easy targets. In that case, it is better to directly switch to the final stage of more careful mutation-based refinement. For hard targets, the situation changes: The initial binder candidate by the generative model is often not strong and significant refinement is necessary, for which the early BindCraft stages that more radically update the sequence can be helpful as well. This behavior is captured in Figure 16 in the manuscript. We extended the discussion in Appendix I.3 in the updated paper to discuss these points more thoroughly.
> > >
> > > We would again like to thank the reviewer for their feedback and hope that we were able to address their concerns and questions. Please let us know if you have any further questions that we can discuss. Otherwise, we would like to kindly ask you to consider raising your score accordingly. Thank you very much.
> > >
> > > [1] Pacesa et al., One-shot design of functional protein binders with BindCraft, Nature, 2025.
> > >
> > > [2] Nori et al., BindEnergyCraft: Casting Protein Structure Predictors as Energy-Based Models for Binder Design, arXiv, 2025.
> > >
> > > [3] Cho et al., Boltzdesign1: Inverting All-Atom Structure Prediction Model for Generalized Biomolecular Binder Design, BioRxiv, 2025.
> > >
> > > [4] Overath et al., Predicting Experimental Success in De Novo Binder Design: A Meta-Analysis of 3,766 Experimentally Characterised Binders, BioRxiv, 2025.
> > >
> > > [5] Ahern et al., Atom level enzyme active site scaffolding using RFdiffusion2, BioRxiv, 2025.

---

### Official Review · Reviewer_79qv · 2025-10-31

**Soundness:** 2
**Presentation:** 2
**Contribution:** 2
**Rating:** 4
**Confidence:** 3

**Summary:**

This paper proposes Complexa, a generative framework for fully atomistic protein binder design that claims to unify generative and hallucination-based paradigms. The model builds upon La-Proteína (Anonymous, 2025) using a partially latent flow-matching architecture and introduces (1) a new synthetic binder–target dataset called Teddymer derived from domain–domain interactions in AlphaFold Database monomers, and (2) test-time optimization strategies such as beam search, Feynman–Kac steering, and Monte Carlo Tree Search. The authors argue that this unification improves binding success rates under normalized compute budgets.

**Strengths:**

The paper provides a comprehensive engineering system integrating generative pretraining, dataset synthesis, and inference-time optimization.

The construction of Teddymer—though synthetic—could be a useful large-scale dataset for structural learning.

The writing is technically detailed and clear, with ablation studies and code-release commitment.

The topic of scaling test-time compute for binder design is timely and of interest to the ICLR community.

**Weaknesses:**

Unclear contribution beyond La-Proteína.
The proposed architecture directly extends La-Proteína’s partially latent flow-matching model with an added conditioning token for target residues. While the authors emphasize “unifying generative and hallucination methods,” this mainly translates into using La-Proteína’s backbone plus inference-time sampling and optimization (beam search, MCTS) already well-known from diffusion models. The methodological novelty appears incremental, not conceptual. It is unclear what fundamentally distinguishes Complexa from La-Proteína aside from adding conditioning and test-time heuristics.

No rigorous comparison with latest state-of-the-art works.
Despite citing BoltzDesign (Cho et al., 2025) and BindCraft (Pacesa et al., 2025) in the related-work section, the experiments do not include direct quantitative or qualitative comparisons with these models under standardized benchmarks. These are the leading methods for atomistic binder design that already implement optimization over structure predictors and incorporate differentiable docking. Without such baselines, it is impossible to judge the claimed “state-of-the-art” performance.

Questionable novelty of “test-time scaling.”
The adaptation of test-time compute scaling (Best-of-N, Beam Search, MCTS) is straightforward and mirrors what is standard in diffusion and flow-based generative modeling in language, vision, and molecule generation. There is no specific algorithmic innovation tailored to protein structures, nor theoretical analysis of why such scaling improves binder discovery.

Marginal improvement and unclear biological impact.
Reported “unique success” metrics show numerical gains over older baselines such as RFDiffusion or APM, but do not exceed or even match the scale and biochemical relevance achieved by recent multimodal systems like AlphaFold 3, Boltz-2, or Chai-2. There is no wet-lab validation, no docking energy correlation, and no experimental evidence that the generated binders are meaningful beyond AlphaFold confidence metrics.

Ambiguous conceptual framing.
The paper repeatedly emphasizes “bridging generative and hallucination approaches,” yet the implementation merely combines a pretrained generator with structure-score-guided search. This framing risks overstating what is essentially a hybrid of established techniques. Moreover, the term “scaling atomistic protein binder design” is misleading—there is no clear demonstration of scaling laws, compute efficiency, or emergent capability analyses.

**Questions:**

N/A

---

> ### Author Response · Authors · 2025-11-22
> **Reply by Authors (1)**
>
> We would like to thank the reviewer for their review. We appreciate that they highlighted our novel Teddymer dataset, the clear writing, our extensive ablation studies, as well as our code-release commitment. We also appreciate that the reviewer considers our work timely and of interest to the ICLR community.
>
> For an overview over all added results and modifications to the updated paper, please see the message to all reviewers. Please note that we added many new experiments and analyses to the paper, which we believe further improves our contributions. The main changes in the updated manuscript are highlighted in orange color for your convenience.
>
> In the following, we will address the reviewer’s comments and questions:
>
> **1. Contributions beyond La-Proteina:** Complexa indeed builds on top of La-Proteina, but our work makes a number of novel contributions:
> - The original La-Proteina model can only generate monomers, while real-world protein design tasks often involve binder design. To this end, our new Complexa model extends the La-Proteina latent denoiser architecture with a novel latent target conditioning mechanism to condition on protein targets. Meanwhile, the autoencoder module only needs to model monomers, in our case the binder protein, thereby maintaining a simple, scalable and efficient generative model design. On top of that, to facilitate reasoning over the positioning of the generated binder, we additionally introduce a novel translation noise mechanism (see Section 3.2 and Figure 5). We believe that these innovations represent non-trivial conceptual extensions of La-Proteina.
> - The literature on binder design so far consists of either fully generative (e.g. RFDiffusion, APM, Protpardelle), or fully optimization-based hallucination methods (e.g. BindCraft, BoltzDesign, AlphaDesign). Complexa is the first method that combines both generation and optimization in a single framework, where the optimization directly leverages the generative prior. This is a conceptual novelty in the large field of protein design, and our results demonstrate the clear advantage of our method. In practice, this corresponds to applying search methods like Beam Search, MCTS, or Feynman-Kac Steering to our generative prior. To the best of our knowledge, we are the first to apply these methods in the context of binder design across the entire literature. Moreover, we actually needed to adjust some of these methods in novel ways to make them applicable to our setting. This applies in particular to Monte Carlo Tree Search, please see Appendix H.4 and discussion below in point 3.
> - To train a strong generative base model, we curate and introduce a novel interface dataset, Teddymer, which we will publicly release. This represents another valuable contribution, from which the community will benefit.
> - Our extensive in-silico evaluations dramatically improve over all publicly available baseline methods, establishing Complexa as the state-of-the-art binder design method for protein and small molecule targets (also see next point on this). Importantly, this is without the need for sequence re-design, in contrast to virtually all prior approaches.
> -  We demonstrate additional capabilities, such as fold class-guided binder design (Figure 6) and binder design with hydrogen bond optimization (Table 3, Figure 10), opening up the possibility to integrate physics-based methods in the design process. No previous binder design methods offer such capabilities.
>
> In conclusion, we are confident that our work makes several original and valuable contributions, as also recognized by the other reviewers.
>
> **2. Comparisons to previous state-of-the-art methods:** The reviewer writes *”Despite citing BoltzDesign (Cho et al., 2025) and BindCraft (Pacesa et al., 2025) in the related-work section, the experiments do not include direct quantitative or qualitative comparisons with these models under standardized benchmarks.”*. We would like to respectfully point out that this is factually incorrect. We are indeed extensively comparing Complexa to both BoltzDesign and BindCraft (and AlphaDesign): In Figure 7, we are comparing to both BoltzDesign and BindCraft, finding that under matched compute budgets Complexa generates many more unique successes, for both easy and hard protein targets (Best-of-N, BeamSearch, Monte Carlo Tree Search, Fyenman-Kac Steering, Generate & Hallucinate all correspond to different Complexa versions with difference inference-time optimization methods). In Figure 9, we are comparing Complexa to BoltzDesign for small molecule targets, and Complexa again wins, except when using MCTS (note that BindCraft and AlphaDesign do not allow binder design against small molecule targets). A case study for VEGFA is shown in Figure 8, where we again win. In the Appendix, we have results for all individual targets, see Figures 16, 17, 18. We would like to kindly ask the reviewer to revisit these results.
>
> *(continued below)*

---

> ### Author Response · Authors · 2025-11-22
> **Reply by Authors (2)**
>
> Summarizing, we indeed *extensively* benchmark Complexa against both BindCraft and BoltzDesign in our paper, in fair comparisons. Please note that this was recognized by the other reviewers. For instance, *reviewer 7jnG* explicitly writes *”Extremely Strong Empirical Results: The paper demonstrates clear and significant superiority over existing SOTA methods (RFDiffusion, BindCraft, BoltzDesign) in fair, compute-matched comparisons.”*
>
> Therefore, we stand by our claim that Complexa achieves state-of-the-art binder design performance against protein and small molecule targets when compared to publicly available baseline methods.
>
> **3. Novelty of test-time scaling:** We would like to clarify that our paper’s goal is not to fundamentally design new inference-time scaling algorithms and we make no such claims. Rather, Complexa is the first model to successfully adapt inference-time optimization methods from the flow and diffusion literature to protein binder design. This allows us to unify previous distinct generative and optimization-based binder design methods, and leads to state-of-the-art performance by a large margin. Furthermore, the reason why this scaling improves binder discovery under normalized compute budgets is that search within a pre-trained generative prior is more efficient than random search. Prior methods like BindCraft or BoltzDesign essentially perform a slow random sequence search, guided by a protein structure prediction model. Complexa instead relies on a pre-trained generative model to more efficiently steer the search, thereby finding successful binder candidates more quickly.
>
> We would also like to point out that we actually did modify the existing methods in a non-trivial manner. Specifically, traditional MCTS has a finite action space. Here, in contrast, we develop a new MCTS algorithm that can be efficiently run in a continuous and unbounded action space, corresponding to the different denoising paths, in a batched manner. We encourage the reviewer to read Section H.4 and algorithm 4 in the Appendix. In a nutshell, we merge MCTS’ selection and expansion into a single stochastic decision: with some probability epsilon a new child is expanded via denoising steps, otherwise the best existing child is selected. This replaces the fixed discrete branching factor with a probabilistic branching mechanism that naturally scales to continuous diffusion trajectories while preserving exploration-exploitation balance. Moreover, by using a shared epsilon across all nodes, the algorithm ensures batched parallel denoising, enabling efficient inference compatible with diffusion models’ synchronized time steps.
>
> **4. Quantitative improvements and biological impact:** The reviewer writes *”Reported “unique success” metrics show numerical gains over older baselines such as RFDiffusion or APM, but do not exceed or even match the scale and biochemical relevance achieved by recent multimodal systems like AlphaFold 3, Boltz-2, or Chai-2.”* This statement is unclear, because AlphaFold-3 and Boltz-2 are not de novo binder design models, but protein structure prediction models that leverage given, already-existing input sequences. They tackle a different task and are therefore not applicable as baselines for Complexa. We would like to kindly ask the reviewer to clarify. Also note that Chai-2 is an undisclosed, proprietary method and not publicly available, making comparisons impossible. Moreover, as discussed above (see 2.), we compare to all publicly available baselines, including strong methods such as BoltzDesign and BindCraft. Furthermore, we agree that experimental wet lab validation of the generated binder candidates would be valuable, but this is beyond the scope for a machine learning conference paper. In an effort to nonetheless provide additional biophysical evaluations orthogonal to the current folding model ipAE scores, we carried out two more analyses:
> - For the protein target binders, we analyzed the generated interfaces by calculating a set of biophysical interface metrics, and compared to reference interfaces from the Protein Databank. The metrics include: number of hydrogen bonds across the interface, hydrophobicity of the binder interface, hydrophobicity of the binder surface, shape complementarity of the binder interface, delta Solvent Accessible Surface Area (dSASA) of the binder interface, and the number of binder residues on the interface. For most properties the generated binders follow the same trend as the PDB reference set, indicating that our model generates realistic target-binder interfaces. See Appendix I.13 and figure 29 in the updated paper for details.
>
> *(continued below)*

---

> > ### Author Response · Authors · 2025-11-22
> > **Reply by Authors (3)**
> >
> > - For the small molecule target binders, we analyzed the binding affinity of the generated protein-ligand complexes using the FLOWR.root affinity prediction model [1]. The results indicate that our Complexa model generates physically plausible binders, with stronger or on-par binding affinity values compared to RFDiffusion-AllAtom. Please see Appendix I.12 and table 13 for details.
> >
> > **5. Conceptual framing:** We would like to kindly point out that existing hallucination-based methods, arguably currently the most popular binder design approach, essentially correspond to pure structure-guided sequence search. In our work, we are accelerating this search by optimizing *within a pre-trained generative model*, using different search algorithms applicable to optimization of the denoising trajectories of diffusion and flow models. We argue that this indeed corresponds to bridging generative and search-based (hallucination) approaches. We agree that this can also be seen as a hybrid of these approaches, but that is the message of our paper: Complexa is the first framework to unify generative model-based and optimization-based binder design within a joint, hybrid framework, harnessing the benefits of both frameworks and outperforming all previous pure generation-based and optimization-based methods.
> >
> > With regards to “scaling atomistic protein binder design”, we scale the training data by creation of a novel training dataset (Teddymer), and we scale inference-time compute and show clear scaling plots, where more in-silico successes are found with scaled compute (see figures 7, 8, 9, 21, 22, 23). This also demonstrates our method’s compute efficiency compared to baselines. We also perform a more extreme compute scaling case study, optimizing for over 100 GPU hours for very hard targets (TNF-alpha, H1, IL17A; please see Section 4.2 and scaling plots in figure 19 in Appendix I.4 in the updated manuscript) as well as SpCas9 (see Appendix I.10 and figure 27). Publicly available baselines were not able to produce binders for TNF-alpha, H1, IL17A, while Complexa succeeded when scaling compute and optimization sufficiently. This can be seen as a novel capability emerging from our model’s search and compute scaling framework. In conclusion, there are multiple aspects of data and compute scaling in our work and we even scale the number of targets for which we can create successful in-silico binders. For these reasons, we believe that “scaling atomistic protein binder design” is an appropriate description of our work.
> >
> >
> > We would again like to thank the reviewer for their feedback and hope that we were able to address their concerns. Please let us know if you have any further questions that we can discuss. Otherwise, we would like to kindly ask you to consider raising your score accordingly. Thank you.
> >
> > [1] Cremer et al., FLOWR.root: A flow matching based foundation model for joint multi-purpose structure-aware 3D ligand generation and affinity prediction, arXiv, 2025.

---

> > > ### Author Response · Authors · 2025-11-28
> > > **Thank you.**
> > >
> > > Dear reviewer 79qv, we noticed that you increased your overall paper score to 6 (on Nov. 25th), which we appreciate. Thank you.

---

### Official Review · Reviewer_FnDt · 2025-10-31

**Soundness:** 4
**Presentation:** 3
**Contribution:** 4
**Rating:** 8
**Confidence:** 4

**Summary:**

The paper proposes Complexa, a partially latent flow-matching framework for binder generation across protein and small-molecule targets. To enlarge the training dataset, this paper also introduces Teddymer, a newly constructed dimer dataset derived from TED with complete CATH annotations. The authors also design and analyze various test-time scaling strategies to improve generative performance. Experimental results on multiple datasets demonstrate the method’s strong capability and generalization in protein–ligand and protein–protein binding tasks.

**Strengths:**

1. This paper contributes a valuable dataset named Teddymer, which can serve as high-quality binder training data and benefit future studies.

2. The proposed Complexa framework leverages La-Proteína, a partially latent flow matching framework, for binder generation, extending the scope of the original model.

3. This paper thoroughly analyzes test-time scaling strategies, opening an advanced and promising direction.

4. The experimental evaluation is extensive, covering diverse scenarios with clear explanations and detailed ablations.

**Weaknesses:**

While the technical contributions are solid, this paper contains several typos or inconsistencies that slightly affect readability. Specific examples include:

- Line 292: interference-time -> inference-time
- Inconsistent alternates between "CATH annotation" and "CAT labels"
- Line 1633: MTCD -> MTCS

Moreover, as this paper introduces a relatively complicated framework, the reviewer recommend adding an overview paragraph to go through the entire pipeline at the beginning of Section 3, including the data construction, latent flow matching framrwork, and test-time scaling strategies.

**Questions:**

1. There is a recent work [A] attempting to solve both small-molecule and protein binder generation within a single model. Within Complexa’s partially latent formulation, could a single shared model cover both cases? A brief discussion of this possibility would strengthen the paper’s positioning.
2. For very hard cases, it would be informative to include scaling curves similar to Figure 13/14 to illustrate the generation difficulty and scaling behavior quantitatively.

[A] Kong, Xiangzhe, et al. "UniMoMo: Unified Generative Modeling of 3D Molecules for De Novo Binder Design." Forty-second International Conference on Machine Learning.

---

> ### Author Response · Authors · 2025-11-22
> **Reply by Authors**
>
> We would like to thank the reviewer for their positive feedback. We appreciate that they highlighted our novel Teddymer dataset as well as our extensive experimental evaluations. Furthermore, we thank the reviewer for highlighting our thorough analyses of test-time scaling strategies and the corresponding promising directions.
>
> For an overview over all added results and modifications to the updated paper, please see the message to all reviewers. Please note that we added many new experiments and analyses to the paper, which we believe further improves our contributions. The main changes in the updated manuscript are highlighted in orange color for your convenience.
>
> In the following, we will address the reviewer’s comments and questions:
>
> **Typographical Errors:** Thank you for pointing those out. We addressed them in the updated version of the manuscript.
>
> **Additional Overview Paragraph:** Thank you for this great suggestion. We added a corresponding paragraph at the beginning of Section 3 in the updated manuscript. Please have a look. Since the paper length requirement is changed to 10 pages for rebuttal and camera-ready, we could easily make this adjustment. If you have additional suggestions, we would be happy to further modify.
>
> **Question 1: A Single Model:** Thank you for pointing this out. Indeed, in the future a single model could possibly be trained for different design tasks, like mini-binder generation for both protein and small molecule targets as in our work. This would be conceptually relatively easy in the current partially latent flow matching framework, since in both cases it would be protein binders that are generated, which have the same partially latent representation. We could use the same autoencoder as in the current Complexa version, and we would only need to implement the ability to condition the same partially latent generative model on both small molecule and protein targets.
>
> However, more generally, it would be interesting to train a single, unified model that can both condition on and flexibly generate different molecular modalities, including proteins, peptides, small molecules, nucleic acids, antibodies and more. To this end, one could extend the autoencoder training of the partially latent flow matching framework to new modalities and predict corresponding latent representations. Joint training on diverse molecules, and potentially even a shared (partial) latent space may also boost overall performance due to transfer learning across the modalities. We leave these directions to future research, but we added a corresponding sentence in the conclusions section 5 in the updated manuscript, as well as an extended discussion in Appendix A. We now also cite the UniMoMo paper, which you pointed out and which follows such a paradigm (UniMoMo does not design mini-binders like we do, but small molecules, peptides, and antibodies).
>
> **Question 2: Scaling Behavior for Very Hard Cases:** We now included the scaling curves for the very hard binder design cases (targets TNF-alpha, H1, IL17A) in Appendix Section I.4 in Figure 19. Please have a look. Note that for IL17A we found only a single unique success early during the search process, whereas for the other two targets sufficient inference-time optimization can indeed produce further unique successes, thereby demonstrating the power of Complexa’s inference-time compute scaling framework.
>
> We would again like to thank the reviewer for their feedback. Please let us know if you have any further questions that we can address. Otherwise, we would like to kindly ask you to consider raising your score accordingly. Thank you very much.

---

> > ### Comment · Reviewer_FnDt · 2025-11-23
> >
> > Thanks for the authors' efforts! The reviewer's concerns have been fully addressed. Hence I have increased my confidence to 5. Nice work!

---

> ### Author Response · Authors · 2025-11-28
> **Thank you.**
>
> Dear reviewer FnDt, we are glad that we were able to fully address your concerns. Thank you for raising your confidence score to 5 (on Nov. 23rd) and for supporting our submission.

---

### Official Review · Reviewer_NBYf · 2025-11-01

**Soundness:** 4
**Presentation:** 4
**Contribution:** 4
**Rating:** 10
**Confidence:** 4

**Summary:**

This work introduces Complexa, which extends La-Proteina for binder design. Authors introduce the Teddymer dataset of synthetic protein dimer pairs: using annotations from TED, domain-domain interactions are retrieved from AFDB. For base model training, the partially latent La-Protein model is extended by representing targets as atom37 features, alongside sequence identity and binary hotspot tokens. A translation noise objective is used. At inference time, the model is steered using interface confidence scores from structure predictors or h-bond energies as rewards, with hotspots pre-specified. Authors experiment with different techniques (beam search, MCTS, etc.) Baselines are examined for both protein binders and small molecules.

**Strengths:**

Really nice exploration of inference-time steering strategies. Conditioning on both small molecule and protein binders is a great framing. Rewards are very sensibly chosen. The inference-time optimization analyses, complete with the inference-time compute scaling plots and compute normalized comparisons, will be a great addition to the literature. Not requiring explicit sequence re-design is elegant, and it’s great to set this new standard for the field. It’s great to see that the authors did not cut corners by choosing an easy task or weak baselines. Technically strong paper with strong presentation and very comprehensive appendix. This is a polished work and should be highlighted at the conference.

**Weaknesses:**

* The Teddymer dataset is quite sensible given the data-limited regime in biology. I do wonder if recapitulating the inter-domain distribution will be as useful for real-world uses such as mini-binder design. It also means that we are again massively overrelying on AlphaFold both for distilling the training data, post-training, and for evaluation. This could conflate evaluation numbers, especially for anything that still relies on AF2, and any biases stemming from AF2 would not be picked up until there are wet-lab (or other orthogonal) validations
* As with any optimization procedure, we’re beholden to the quality of the reward signal. ipAE is not a perfect signal, though I appreciate it’s one of the best solutions available.

**Questions:**

* Not a necessary additional experiment per se, but I’m curious if the inference search strategies are helping move us away from the pretraining distribution to something better, or if it’s helping us better interpolate within the pretraining distribution? Curious to hear the authors’ intuitions on this.
* It looks, from Figure 7, that the metrics continue to increase with GPU hours. What if we ran one of the methods for even longer? Do we expect to see an eventual plateau? Could this possibly imply reward hacking, given that we don’t see a plateau in the current plot?
* When optimizing for ipAE, I’m curious if authors saw any signs of adversarial optimization. What if we plotted orthogonal metrics (e.g. Rosetta energy or h-bond counts) on the x axis with ipAE on the y axis for all generations? Do these metrics correlate well?

---

> ### Author Response · Authors · 2025-11-22
> **Reply by Authors (1)**
>
> We would like to thank the reviewer for their very positive review and constructive feedback. We appreciate that the reviewer highlighted our inference-time compute scaling framework and our extensive baseline comparisons and considers our paper technically strong with a strong presentation. We also thank the reviewer for noting our comprehensive appendix and suggesting the paper as being highlighted at the conference.
>
> For an overview over all added results and modifications to the updated paper, please see the message to all reviewers (main changes are highlighted in orange color). In the following, we will address the reviewer’s comments and questions:
>
> **Teddymer dataset and reliance on AlphaFold2 and ipAE scores:** We agree with the reviewer’s comment that in-silico evaluation of protein binders currently relies heavily on folding models and their confidence and error estimates (ipAE). As mentioned by the reviewer, this is nonetheless the best available solution and standard practice in the field – we are simply following best practices, aligned with prior work. Wet lab validation would be helpful, but is beyond the scope of this submission to a machine learning conference, where our focus is on our novel joint generative and optimization framework.
>
> The value of the newly created Teddymer dataset is demonstrated in our ablation study (see Appendix I.1, Table 6 and Figure 13).  The reviewer correctly points out, though, that Teddymer originates from AF2-predicted AlphaFold Database structures, and that we are also using AF2 (ColabDesign AlphaFold2-Multimer implementation) to quantify model performance. To address this concern, we carried out an additional experiment in which we measured the performance in our ablation study over Teddymer data with two separate folding models, Boltz-2 [1] and RosettaFold-3 [2], using similar ipAE-based success criteria as before with AF2. Please see the results in Appendix I.1.1 in the updated paper.
>
> In a nutshell, we find that also when binder quality is measured via Boltz-2 or RosettaFold-3, Teddymer dramatically boosts performance. This confirms the value of the Teddymer data independent of using AF2 specifically as the final model for evaluation. Moreover, related to this question, we also analyzed the Teddymer interfaces in terms of their geometric and biophysical properties more closely and compared them to experimental interfaces from the protein databack (PDB), see newly added section C.1 in the Appendix in the updated paper. We find that the Teddymer data, despite some biases, generally mimics real structures very well, which is aligned with the positive results we observed when including Teddymer in the training of Complexa.
>
> **Question 1: Interpolation vs. Extrapolation during Inference-time Search:** This is an excellent question. Methods like Best-of-N Sampling, Beam Search, Feynman-Kac Steering and Monte Carlo Tree Search directly leverage the pre-trained generative model. Hence, in our implementation they primarily search *within* the distribution, which can be interpreted as *interpolation*. Directly leveraging the generative prior during search in that way resulted in our strong results when compared to baselines. However, this behaviour could be adjusted: For instance, we could modify the sampling algorithm (see B.4) used during inference-time search to increase the amount of random noise injection and scale down the effect of the generative prior (the velocity component). This would encourage random exploration and *extrapolation* at the cost of less efficiency and less guidance by the pre-trained model.
>
> Also note that the search stage of “Generate and Hallucinate” is pure sequence optimization (as exemplified by the prior work BindCraft) and also does not rely on the pre-trained generator. Here, only the initialization is based on the generator. Therefore, this can also be regarded as extrapolatory behavior. This is also leveraged when optimizing for the very hard targets (see I.4), where we first optimize with MCTS (interpolatory behavior), and then perform further sequence refinement (extrapolatory behavior).
>
> *(continued below)*

---

> > ### Author Response · Authors · 2025-11-22
> > **Reply by Authors (2)**
> >
> > **Question 2: Longer Inference-time Search and Plateaus:** To save compute resources, we did not run the inference-time search exhaustively until potential plateauing behavior in our main experiments. However, we appreciate the reviewer’s question, and in response we performed a case study and now ran one of our search methods (beam search) significantly longer for the SpCas9 target, for up to 1,100 GPU hours; please see the newly added section I.10 in the Appendix in the updated paper. Importantly, we analyze the generated unique successes while using different clustering thresholds to determine uniqueness. We find that while the total number of successful binders continues to rise, the discovery of structurally distinct clusters indeed slows down, aligned with the intuition of the reviewer. We would like to point out that this plateauing behavior is expected to be highly target-dependent. Due to the computational expense, here we focused the case study on a single target. We do not think that the observed results are a sign of adversarial optimization. It is expected that the overall success rate with fine clustering and little deduplication can increase for a long time, while the number of distinct successes plateaus, as observed.
> >
> > **Question 3: ipAE vs. Hydrogen Bond Counts:** As a response to this question, we analyzed ipAE vs. hydrogen bond count, as suggested by the reviewer; please see the newly added section I.11 in the updated paper’s Appendix and figure 28. We find that when optimizing binders for low ipAE scores, as in all our main inference-time optimization experiments, there is a weak correlation between interface hydrogen bonds and ipAE: lower ipAE, indicating a better interface, is correlated with more interface hydrogen bonds (Spearman correlation of -0.69), similarly indicating a better interface. This is the desired behavior, and we found no signs of adversarial optimization.
> >
> > We would again like to thank the reviewer for their constructive comments and positive evaluation. Please let us know if there are any further questions. Thank you!
> >
> > [1] Passaro et al., Boltz-2: Towards accurate and efficient binding affinity prediction, BioRxiv, 2025.
> >
> > [2] Corley et al., Accelerating Biomolecular Modeling with AtomWorks and RF3, BioRxiv, 2025.

---

### Author Response · Authors · 2025-11-22
**Additional Experiments during Rebuttal**

Dear Reviewers,

We would like to thank you for the extensive and overall very positive reviews. We are glad that our paper is received well. The reviewers highlighted our new inference-time compute scaling framework for protein binder design, our strong numerical results and extensive evaluations, our novel Teddymer dataset, as well as Complexa’s power and flexibility generally. Also, our comprehensive appendix and the paper’s presentation were positively pointed out.

Concerns and questions are addressed in individual replies to the reviewers. Here, we provide an overview over all additional experiments we carried out for the rebuttal. Given the increased page limit for rebuttal and camera-ready version, we extended the paper’s main text to 10 pages. Relevant changes in the updated manuscript are highlighted in orange color for your convenience (we will change this for the camera-ready version).

Additional results and experiments carried out during the rebuttal:
- **Teddymer vs. PDB.** We performed a detailed geometric and biophysical analysis of our novel synthetic Teddymer interface data and quantitatively compared the results to real experimental binder interfaces from the PDB. We find substantial overlap between the results, showing that the synthetic Teddymer interfaces closely resemble real protein-protein interfaces. This supports Teddymer as a valid proxy for true binder interactions valuable for training Complexa. **See Appendix C.1.**
- **Teddymer ablation study with different folding models.** We measured the performance in our ablation study over Teddymer data with two separate folding models, Boltz-2 [1] and RosettaFold-3 [2], using similar ipAE-based success criteria as before with AlphaFold2-Multimer. In a nutshell, we find that also when binder quality is measured via Boltz-2 or RosettaFold-3, Teddymer dramatically boosts performance. This confirms the value of the Teddymer data independent of using AlphaFold2 specifically as the final model for evaluation. **See Appendix I.1.1.**
- **Enzyme Design Benchmark.** We apply Complexa to enzyme design, evaluating it on the Atomic Motif Enzyme (AME) benchmark introduced by RFDiffusion2 [3]. To tackle this task, we extended our Complexa model for small molecule targets with atomistic motif reconstruction capabilities. We find that Complexa significantly outperforms RFDiffusion2 on almost all tasks, both with self-generated sequences and re-designed sequences using LigandMPNN. **See Section 4.3 and Appendix I.9.**
- **Long Inference-time Optimization Case Study.** We performed an extended inference-time search on the SpCas9 target to study how Complexa explores the binder design space over long runtimes. Using different structural clustering thresholds, we tracked the diversity of generated binders. The model continued to discover new solutions for a long time, revealing a rich design space, but the rate of finding new binders distinct from previous successes gradually slowed, indicating eventual saturation. **See Appendix I.10.**
- **ipAE and Interface Hydrogen Bond Correlations.** To study whether inference-time optimization of ipAE can lead to adversarial effects such as the degradation of relevant biophysical interface properties, we investigated both ipAE scores and the number of interface hydrogen bonds in Complexa samples generated by ipAE optimization. We found no sign of adversarial behavior, but the opposite: improved ipAE scores are correlated with enhanced hydrogen bond interactions between binder and target. **See Appendix I.11.**
- **Protein-Ligand Binding Affinities.** For the small molecule target binders, we analyzed the binding affinity of the generated protein-ligand complexes. The results indicate that our Complexa model generates physically plausible small molecule binders, with stronger or on-par affinity values compared to RFDiffusion-AllAtom. **See Appendix I.12**
- **Biophysical Interface Analyses for Protein Targets.** For the protein target binders, we analyzed the generated interfaces by calculating a set of biophysical interface metrics, and compared to reference interfaces from the Protein Databank. For most metrics the generated binders follow the same trend as the PDB reference set, indicating that our model generates realistic target-binder interfaces. **See Appendix I.13.**

We believe that these additional results further strengthen our paper significantly. Again, we thank the reviewers for their valuable suggestions leading to these investigations.

Thank you,

The authors of *“Scaling Atomistic Protein Binder Design with Generative Pretraining and Test-Time Compute”*

[1] Passaro et al., Boltz-2: Towards accurate and efficient binding affinity prediction, BioRxiv, 2025.

[2] Corley et al., Accelerating Biomolecular Modeling with AtomWorks and RF3, BioRxiv, 2025.

[3] Ahern et al., Atom level enzyme active site scaffolding using RFdiffusion2, BioRxiv, 2025.

---

### Meta-Review · Area_Chair_YaDF · 2026-01-05

**Summary:**

This paper introduces Complexa, a structure conditioned binder design framework that combines large scale generative pretraining with inference time compute scaling to improve success rates.
I recommend acceptance for three main reasons. First, the central idea of unifying a strong generative prior with systematic inference time search is timely and well supported by extensive experiments and compute matched comparisons.
Second, the Teddymer construction tackles the data bottleneck at scale and the paper provides additional evidence that its benefit is not merely an artifact of a single evaluator, including re evaluation with other structure predictors.

**Reviewer Concerns:**

Addressed in rebuttal or discussion: stronger evidence for Teddymer utility and realism, more detailed scaling analyses, and additional checks related to reward signals and evaluation dependence.
Still outstanding: reliance on external structure predictor based scoring at inference time remains a limitation, and the work does not include wet lab validation, though this is common for the venue and the paper is careful to frame results as computational evidence.

**Reviewer Scores:**

NBYf stays at 10.
FnDt stays at 8 and increased confidence after concerns were addressed.
7jnG stays at 6 with remaining practicality and oracle dependence concerns.
79qv increased from 4 to 6 as reflected in the discussion summary.

---

### Decision · Program_Chairs · 2026-01-26

Accept (Oral)